# MEGA-BENCH 🔮 : SCALING MULTIMODAL EVALUATION TO OVER 500 REAL-WORLD TASKS

**Jiacheng Chen**[*,†] **Tianhao Liang,**[*] **Sherman Siu, Zhengqing Wang, Kai Wang,**
**Yubo Wang, Yuansheng Ni, Wang Zhu, Ziyan Jiang, Bohan Lyu, Dongfu Jiang,**
**Xuan He, Yuan Liu, Hexiang Hu,**[‡] **Xiang Yue,**[‡] **Wenhu Chen**[*,†]

MEGA-Bench Team

https://tiger-ai-lab.github.io/MEGA-Bench/

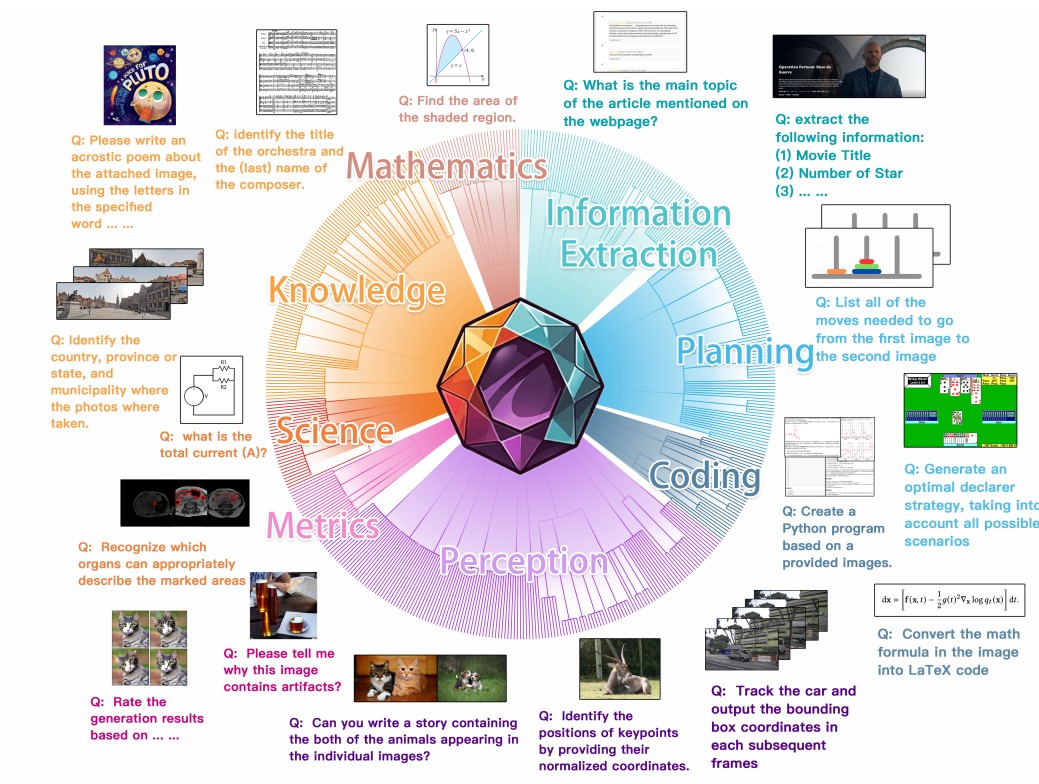

Figure 1: MEGA-BENCH contains 505 multimodal tasks with diverse data sources, input/output formats, and skill requirements. The taxonomy tree guides and calibrates the annotation process.

## ABSTRACT

We present MEGA-BENCH, an evaluation suite that scales multimodal evaluation to over 500 real-world tasks, to address the highly heterogeneous daily use cases of end users. Our objective is to optimize for a set of high-quality data samples that cover a highly diverse and rich set of multimodal tasks, while enabling cost-effective and accurate model evaluation. In particular, we collected 505 realistic tasks encompassing over 8,000 samples from 16 expert annotators to extensively cover the multimodal task space. Instead of unifying these problems into standard multi-choice questions (like MMMU, MMBench, and MMT-Bench), we embrace a wide range of output formats like numbers, phrases, code, LaTeX, coordinates, JSON, free-form, etc. To accommodate these formats, we developed over 40 metrics to evaluate these tasks. Unlike existing benchmarks, MEGA-BENCH offers a

---

[*] Core Contributors, [‡] Contributed equally. See the Author Contribution Statement for details.
[†] ✉ jca348@sfu.ca; wenhuchen@uwaterloo.ca

fine-grained capability report across multiple dimensions (e.g., application, input type, output format, skill), allowing users to interact with and visualize model capabilities in depth. We evaluate a wide variety of frontier vision-language models on MEGA-BENCH to understand their capabilities across these dimensions.

# 1 INTRODUCTION

Large foundation models (OpenAI, 2023; 2024a; Anthropic, 2024a; Google, 2023; Meta, 2024; Alibaba, 2024) have dramatically transformed the landscape of artificial intelligence by showcasing exceptional capabilities across various tasks and domains. Originating in the realm of natural language processing, these models have progressively expanded to perceive and interpret multimodal information, including single images, multiple images, and videos. Previously, multimodal models were mainly used for standardized tasks like image captioning (Lin et al., 2014), video captioning (Wang et al., 2019), and visual question answering (Antol et al., 2015; Goyal et al., 2017; Xiao et al., 2021). With the recent progress on multimodal alignment, these models have shown great potential to solve many diverse and complex tasks with well-designed prompts. As a result, people have applied them to assist with many realistic tasks like "web navigation" (Koh et al., 2024), "game playing" (Valevski et al., 2024), "travel planning" (Xie et al., 2024), "visual navigation" (Wang et al., 2023a), "sports analysis" (Xia et al., 2024), "visual entity recognition" (Hu et al., 2023), "visual quality assessment" (Ku et al., 2024), and more. These efforts have significantly increased the utility of multimodal models.

An important challenge is identifying how to accurately gauge the abilities of these vision-language models (VLMs) across a wide range of tasks. Most existing benchmarks are designed to cover only one or a few similar tasks, making them inadequate for evaluating the models' overall capabilities. The status quo is to evaluate the model on many existing benchmarks to showcase their all-round abilities. For example, Qwen2-VL[1] was evaluated on 27 image and video benchmarks in total. Although this massive evaluation effort provides valuable insights into how well these models handle specialized tasks, it also introduces a significant overhead and several challenges:

**- Limited Output Diversity**: The existing multi-task benchmarks like MMMU (Yue et al., 2024a), MMT-Bench (Ying et al., 2024) rely heavily on multiple-choice questions to lower the burden of evaluation. This fails to evaluate the generative abilities of these multimodal models.

**- Lack of Task Coverage**: The existing benchmarks are often sporadic and lack a systematic design to cover the multimodal task space. Certain abilities are not well covered in the current ecosystem. Consequently, even exhaustively testing all the available benchmarks would not be sufficient.

**- Expensive Inference Cost**: The full evaluation process is expensive regarding computation cost/time or API expense. Since many examples or tasks are similar in the capabilities they assess (e.g., DocVQA Mathew et al. (2021) alone has thousands of examples for examining doc understanding and OCR-related abilities), overly repetitive evaluation at a large scale leads to resource waste.

**- Unmanageable Setups**: Each benchmark has complexities when setting up the evaluation. For example, VQA (Goyal et al., 2017) alone has four splits (val, dev-test, std-test, and test). It is hard to track the exact setup of different baseline models to ensure a fair comparison.

To address these challenges, we advocate for a unified protocol that scales up multi-modal evaluation to *maximize the task coverage and the diversity in model outputs while optimizing the inference cost*. As an initial attempt, we propose MEGA-BENCH, which is designed to provide a comprehensive and systematic assessment of multimodal foundation models.

To build MEGA-BENCH, we first construct a *task taxonomy tree* that organizes different multimodal tasks based on the application type (Figure 1), with significant effort spent adjusting and refining the taxonomy tree to ensure sufficient coverage and diversity. The task taxonomy tree then serves as the guiding principle to ensure all relevant tasks and skills are covered and appropriately balanced. To help the annotators create their tasks, we build an annotation GUI to simplify the process of creating the task JSON files and a web tool to visualize the results of the VLM's responses alongside the ground truth. We also review each task contribution when it is first submitted, after evaluating the models on the new tasks, and periodically throughout the annotation process to ensure that all of the tasks are novel and high-quality. This collaborative effort resulted in the compilation of 505 realistic

---

[1] https://github.com/QwenLM/Qwen2-VL

tasks, effectively covering (almost) the entire multimodal capability space at a manageable inference cost. To facilitate nuanced and precise evaluation, we also developed 45 *highly-customized metrics* tailored to these tasks during the annotation process.

Unlike existing benchmarks that often provide a single score, MEGA-BENCH offers a fine-grained capability report based on multiple dimensions such as the input type, input format, output format, and required skills. This interactive and visualizable report enables users to identify the models' performance across several orthogonal dimensions, uncovering strengths and weaknesses that might be obscured in aggregate scores. Such detailed analysis is invaluable for researchers and developers aiming to enhance foundation models and optimize them for specific downstream applications.

Using MEGA-BENCH, we conducted comprehensive studies of popular flagship and efficiency models (with both open-source software and proprietary APIs) and identified some findings below:

**1.** Among flagship models, Claude 3.5 Sonnet (1022) and GPT-4o (0513) currently lead in performance across a wide range of multimodal tasks, with less than a 0.1% difference in their overall scores. Our detailed breakdown shows that Claude 3.5 Sonnet excels in planning and math with its latest upgrade bringing clear boosts in processing UI/Infographics inputs, while GPT-4o leads in information extraction and knowledge-intensive tasks.

**2.** Among open-sourced models, Qwen2-VL performs the best, with its performance near the top close-sourced flagship models, and outperforms the second best open-source model by ≈10%.

**3.** Among efficiency models, Gemini 1.5 Flash is the strongest model overall, except for the tasks related to handling User Interfaces and Documents.

**4.** Proprietary models can effectively leverage Chain-of-Thought (CoT) prompting to improve their performance, while open-source models hardly produce helpful reasoning processes. In our evaluation results, 10 of 13 open-source models get worse results with CoT prompting.

## 2 RELATED WORK

**Multimodal benchmarks.** Benchmarking in vision-language models has been a long-standing research problem. Before the era of large multimodal models, most benchmarks were designed for specific tasks or skills. Some benchmarks like VQA (Antol et al., 2015), GQA (Hudson & Manning, 2019), and ViswizVQA (Gurari et al., 2018) focus on photograph or natural images. ChartQA (Masry et al., 2022), InfoVQA (Mathew et al., 2022), DocVQA (Mathew et al., 2021), and OCR-VQA (Mishra et al., 2019) focus more on documents, infographics, and other similar media. Later on, there was a trend to build more well-rounded benchmarks to cover a wider range of skills or topics, such as ScienceQA (Lu et al., 2022), MMBench (Liu et al., 2023b), MMMU (Yue et al., 2024a;b), MMT-Bench (Ying et al., 2024), and more. However, due to the diversity of these different tasks, most benchmarks use multiple-choice questions for all problems. Therefore, these benchmarks cannot fully reflect the generational abilities of multimodal models. Complementary to this, LMsys arena (Chiang et al., 2024) and WildVision arena (Lu et al., 2024) have proposed to use user voting and Elo-ranking to benchmark multimodal models. Our benchmark is the first to scale up the tasks by a significant magnitude. Furthermore, our benchmark provides a breakdown report to analyze multimodal models across multiple dimensions.

**Sensitivity of large model leaderboards to input format.** Creating reliable leaderboards poses a substantial challenge for evaluating large models. Previous studies have noted that LLMs exhibit sensitivity to minor input modifications, including prompts and in-context examples in few-shot settings (Sclar et al., 2024; Chang & Jia, 2023). To mitigate input sensitivities, researchers have developed specialized prompt design and prompting-based training approaches (Liu et al., 2023a; Jain et al., 2024b). Nonetheless, for benchmarks that only allow a multiple-choice format (Wang et al., 2024d), studies by Zheng et al. (2024) and Robinson et al. (2023) find the option sequencing can significantly alter model rankings on the leaderboard. Recently, Alzahrani et al. (2024) explores the advantage of a hybrid scoring method to stabilize models' leaderboard rankings over input format. Though MEGA-BENCH does not include hybrid scoring for each individual task, the overall use of diverse and hybrid scoring methods and output formats across more than 500 tasks demonstrates *the robustness of the benchmark.*

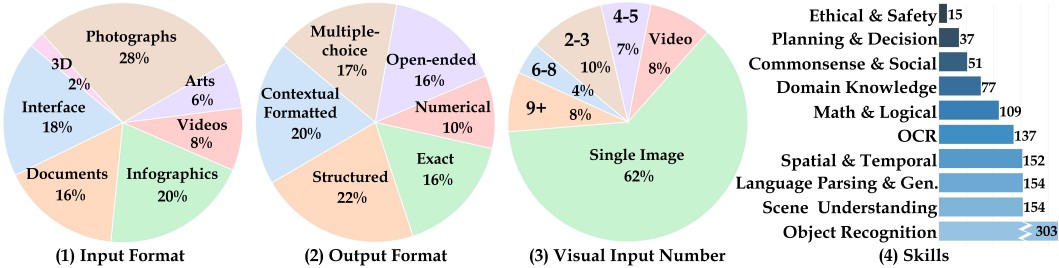

Figure 2: MEGA-BENCH's four keyword dimensions and the task-level statistics. The diversity along various dimensions enables fine-grained capability analysis.

## 3 MEGA-BENCH

MEGA-BENCH is a comprehensive multimodal benchmark that spans 7 input formats, 6 output formats, 10 different types of skills, and a varying number of visual inputs, whether single-image, multi-image or from video, as shown in Figure 2. Our benchmark covers 8 distinct subject areas in a hierarchical taxonomy to evaluate VLMs' ability to tackle various tasks.

### 3.1 BENCHMARK CONSTRUCTION PROCESS

**Preparation.** Figure 3 illustrates our annotation process. In the conceptualization stage, we propose a "draft" task taxonomy tree with the top two levels of Figure 1 by getting inspirations from existing multi-task or multi-discipline LLM/VLM benchmarks (Srivastava et al., 2022; Liu et al., 2023b; Yue et al., 2024a). The first level consists of general applications like "perception", "planning", "reasoning", etc., while the second level has more concrete meta-tasks like "document understanding", "app function understanding", "logic reasoning", etc. We host a brainstorming session to add exemplars under each second-level node and write descriptions about the number and quality of the tasks we expect. Based on our empirical observations of how general users use VLMs in real-world scenarios, we assign more task budgets to perception, knowledge, and information extraction than other first-level nodes while strictly monitoring the application-level distribution balance in the annotation process. We then distribute the second-level nodes in the "draft" tree to the annotators. This top-down framework minimizes overlaps between annotators and facilitates overall organization.

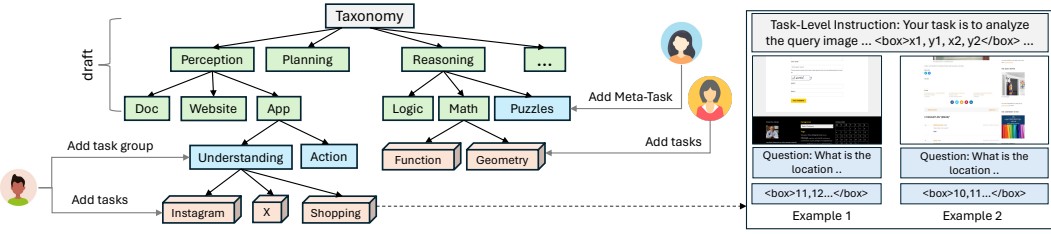

Figure 3: The annotation process of MEGA-BENCH. We propose a "draft" taxonomy tree and then distribute the second-level nodes to annotators. We allow the annotators to gradually refine the tree structure as they add new tasks. Each task has many examples and a shared task-level instruction. Each example has a question and a ground truth answer.

To ensure reliable commitment and annotation quality, we call up 16 designated expert annotators with rich LLM experience and computer science backgrounds. All annotators are graduate students or above with majors in computer science, electronics/communications engineering, bio-statistics, or finance, and 12 of them served as annotators or authors of LLM/VLM benchmarks published at top conferences. The annotators can 1) refine the "draft" taxonomy by adding/deleting nodes, 2) add "task group" nodes and then add a series of tasks under that, and 3) directly add tasks under an existing high-level node. We develop tools to facilitate the annotation process, including 1) an interactive annotation tool that defines the annotation format and automatically unifies all annotations into JSON files, 2) a GitHub repository to coordinate the task submission, reviewing, and discussion process, which was inspired by BIG-bench (Srivastava et al., 2022), and 3) a visualization tool that

allows annotators to browse the existing tasks and the evaluation results of representative vision language models (VLMs). We coordinate all the annotators to ensure they understand our expectations and continuously improve our tools. Please see §B for complete details of annotation protocols.

**Task annotation.** The annotation process contains two rounds. The annotators submit tasks to the benchmark by creating pull requests (PR) to the main branch of our GitHub repository. In the first round of the annotation process, we ask the annotators to contribute 20 tasks following the principles below to ensure the quality of the task:

• *Data source and output format:* Creative tasks with diverse data sources and output formats are encouraged. If the data was collected from existing datasets, we ask annotators to adapt the original annotation into more specific questions and design more diverse answer formats.

• *Number of examples:* Each task should have at least 15 examples. Exceptions are allowed for some complicated tasks where the data are scarce.

• *Documentation:* Each task should be accompanied by documentation that indicates the source of the data, the capabilities the task tries to evaluate, and the evaluation metric to be used.

Our core contributors review each PRs carefully to provide feedback, and the accepted PRs are merged into the main branch. We periodically run the evaluation with commercial VLMs (e.g., GPT-4o) and update the results of existing tasks on our visualization page, which allows the annotators to better understand the difficulty of their tasks and catch potential glitches in the annotation. We found that this helps significantly improve the annotation quality.

Before the second round of annotations, the core contributors review all tasks in the taxonomy tree and investigate the biases in the task distribution. We then host another annotator session to propose new meta-tasks to balance the distribution and maximize the coverage. We then distribute the updated tree nodes to annotators and employ the same guidelines to finish the second-round annotations. After this round, each annotator contributes at least another 30 tasks.

**Quality control and refinement.** We leverage commercial VLMs to examine the task quality. Concretely, we gather the results of GPT-4o, Claude 3.5 Sonnet, and Gemini 1.5 Pro and compute an average score on each task. Tasks with almost 1.0 scores often have trivial questions (based on manual inspection) and can hardly distinguish the ability of different models. We ask the corresponding annotators to investigate and augment those tasks. For tasks with almost zero scores, the task reviewers audit them carefully and remove them if the zero score comes from incorrect annotations or insufficient instruction contexts. Finally, the benchmark contains a total of 505 tasks with roughly 8,200 examples, which is large enough to minimize the sample variance within each high-level taxonomy node. Please refer to §4.3 for an analysis of the number of examples per task.

## 3.2 METRICS FOR ANSWERS IN DIVERSE OUTPUT FORMATS

To properly evaluate the tasks with different output formats, we develop a set of *highly-customized evaluation metrics* in parallel with the benchmark construction process (§3.1). Figure 4 shows several examples of the model outputs along with the task's associated metrics. When new tasks are submitted to our GitHub repository, we implement any new metrics specified by the task authors. We use two types of metrics: rule-based metrics and LLM-assisted metrics. All metrics are normalized into [0, 1], with 1.0 being the full mark.

**Rule-based metrics.** When there is a unique answer under the question context or the correctness of the answer can be verified by rules (e.g., if the generated story/poetry meets the desired formats or if the generated code can pass test cases), we implement *rule-based* metrics for evaluation. To satisfy the needs of all tasks submitted by annotators, we end up with a suite of over 40 rule-based metrics. Robust string parsing is also implemented to extract the answer from the model's response. We conduct a sanity check to ensure the correct implementation of rule-based metrics. Specifically, we create an "oracle" model that always returns the ground truth, then compute its score over all tasks evaluated by rule-based metrics. The sanity check is passed when the "oracle" model gets a full 1.0 score. See §D.4 for details.

**LLM-assisted metrics.** For open-ended tasks that do not have a unique answer, we instead employ an *LLM-assisted* metric (Zheng et al., 2023; Li et al., 2023a). We design a per-task evaluation prompt template and fill in the tailored evaluation criteria for each task. The LLM is instructed to compare

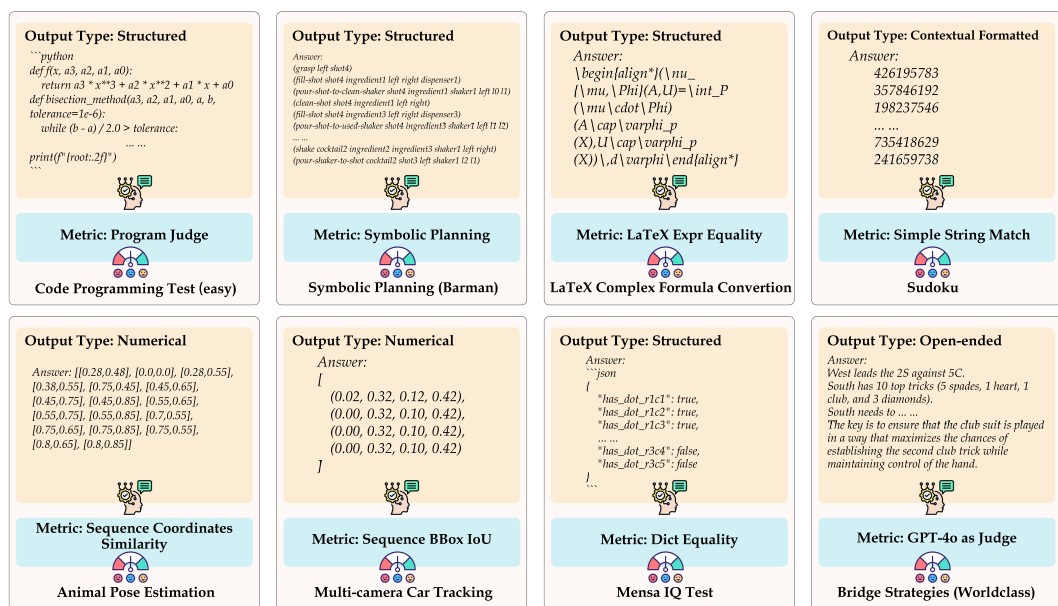

Figure 4: Representative examples of MEGA-BENCH's diverse output formats and customized metrics (input queries are omitted). The outputs are extracted from *real responses* of GPT-4o (0513). We implement robust parsing to extract the final answer from raw model responses.

the model response with the reference answer and assign a score from 1 to 10. The score is then normalized into [0, 1] to be consistent with the other metrics. See §D.3 for details.

We divide the tasks into two subsets based on the different evaluation processes. The *Core Set* is evaluated with rule-based metrics to make the evaluation fast and cost-free. The *Open-Ended Set* is evaluated with metrics that use an LLM-as-a-judge, where the evaluation pipeline calls a proprietary LLM over an API. Specifically, we use GPT-4o-0806 (OpenAI, 2024a) as the judge LLM while maintaining an extensible implementation for using other judge models. The Core and Open-Ended sets contain 440 and 65 tasks, respectively.

## 3.3 MULTI-DIMENSIONAL KEYWORDS FOR FINE-GRAINED ANALYSIS

Existing multi-task multimodal benchmarks analyze models according to dimensions like the image type and academic discipline (Yue et al., 2024a), ability (Liu et al., 2023b), or meta-task (Ying et al., 2024). MEGA-BENCH offers a broad and diverse range of coverage across all these dimensions, and extends even further beyond them. As explained in §3.1, the taxonomy tree divides the tasks into general application scenarios, the most manageable dimension for distributing the annotation efforts to different annotators. After we collected all tasks and finished the quality control process, we grouped all tasks based on four extra dimensions: input visual type, input visual number, output format, and required skills (Figure 2). Each dimension has 6 to 10 keywords, enabling fine-grained analysis and comparison. Interactive visualization tools can then be developed based on our evaluation results, which allows model developers to delve deep into different aspects of a model and compare different models comprehensively.

## 3.4 DATASET STATISTICS AND COMPARISON WITH OTHER BENCHMARKS

MEGA-BENCH contains 505 real-world tasks with 8,186 manually annotated or repurposed samples. Even for repurposed data, considerable effort is needed to convert the original annotations into specific task descriptions, diverse output formats, and additional instructions to include auxiliary information about formatting. Figure 2 shows the task distribution of all five dimensions, and the detailed task taxonomy tree and statistics of each dimension are in Appendix C.

Table 1 compares MEGA-BENCH to existing multimodal benchmarks. The key feature of our benchmark is the diversity across all aspects, driven by our high-level designs of diverse task appli-

Table 1: A comparison between MEGA-BENCH and existing works. MEGA-BENCH has a greater diversity in data sources, input/output format, the number of metrics, and the number of tasks.

| Dataset | Annotation | Source | Input | Output | #Metrics | #Tasks |
|---|---|---|---|---|---|---|
| VQA-v2 (Antol et al., 2015) | New | Photo | 1 Image | Phrase/Bool/Num | 1 | 1 |
| VizwizVQA (Gurari et al., 2018) | New | Photo | 1 Image | Phrase/Bool/Num | 1 | 1 |
| ChartQA (Masry et al., 2022) | New | Chart | 1 Image | Bool/Num | 1 | 1 |
| AI2D (Kembhavi et al., 2016) | New | Diagram | 1 Image | Multi-choice (MC) | 1 | 1 |
| GeoQA (Chen et al., 2021) | New | Geometry | 1 Image | Multi-choice (MC) | 1 | 1 |
| NLVR$^2$ (Suhr & Artzi, 2019) | New | Photo | 2 Images | Bool | 1 | 1 |
| InfoVQA (Mathew et al., 2022) | New | Infographics | 1 Image | Phrase/Bool/Num | 1 | 1 |
| DocVQA (Mathew et al., 2021) | New | Document | 1 Image | Phrase/Bool/Num | 1 | 1 |
| OCR-VQA (Mishra et al., 2019) | New | Book covers | 1 Image | Phrase | 1 | 1 |
| ScienceQA (Lu et al., 2022) | New | K12 Books | ≤1 Image | Multi-choice (MC) | 1 | 26 |
| MathVista (Lu et al., 2023) | Repurposed | Diverse | 1 Image | MC / Num | 1 | 5 |
| MMBench (Liu et al., 2023b) | Hybrid | Diverse | 1 Image | Multi-choice (MC) | 1 | 20 |
| MME (Yin et al., 2023) | Repurposed | Existing | 1 Image | Multi-choice (MC) | 1 | 14 |
| Seed-Bench (Li et al., 2024c) | New | Existing | Image/Video | Multi-choice (MC) | 1 | 12 |
| VisIT-Bench (Bitton et al., 2023) | Hybrid | Diverse | 1/2 Images | Free-form (FF) | 1 | 70 |
| MMStar (Chen et al., 2024a) | Repurposed | Existing | 1 Image | Multi-choice (MC) | 1 | 18 |
| MM-Vet (Yu et al., 2024b) | Repurposed | Existing | 1 Image | Free-form (FF) | 1 | 16 |
| MMMU (Yue et al., 2024a) | New | Diverse | ≥1 Image | MC / FF | 1 | 30 |
| MUIRBench (Wang et al., 2024a) | Hybrid | Existing | >1 Image | Multi-choice (MC) | 1 | 12 |
| MileBench (Song et al., 2024) | Repurposed | Existing | >1 Image | MC / FF | 2 | 12 |
| VideoMME (Fu et al., 2024a) | New | Youtube | Video | Multi-choice (MC) | 1 | 30 |
| MVBench (Li et al., 2024e) | Repurposed | Existing | Video | Multi-choice (MC) | 1 | 20 |
| MMT-Bench (Ying et al., 2024) | Repurposed | Existing | ≥1 Image/Video | Multi-choice (MC) | 1 | 162 |
| MEGA-BENCH | New | Diverse | ≥1 Image/Video | Unrestricted | 45 | 505 |

cations and output formats. MMMU (Yue et al., 2024a;b) focuses on college-level exam questions with various discipline and image formats. All questions are single-image and answered in multiple-choice format. MMT-Bench (Ying et al., 2024) covers 162 concrete sub-tasks, enabling in-depth analysis based on their "taskonomy" and diverse input forms. However, all of the tasks MMT-Bench are from existing datasets, mostly under the "Perception" sub-tree in our taxonomy, and all outputs are in multiple-choice form like MMMU. To maximize task coverage and the diversity in model outputs with cost-effective inference, MEGA-BENCH includes a much broader range of task types and output formats, while having fewer total samples compared to existing benchmarks.

## 4 EXPERIMENTS

We evaluate 19 VLMs with multi-image support on MEGA-BENCH. §4.1 describes the evaluated models and the evaluation pipeline. §4.2 presents the evaluation results with a fine-grained analytical breakdown. §4.3 provides analyses on the number of examples per task and error types.

### 4.1 EVALUATION SETTINGS

**Evaluated models.** We evaluate a diverse range of large multimodal models. The proprietary models assessed include GPT-4o (0513) and GPT-4o mini (OpenAI, 2024a), Claude-3.5-Sonnet (0620 and 1022) (Anthropic, 2024a;b), Gemini-1.5-Pro (002) and Gemini-1.5-Flash (002) (Google, 2024a). For open-source models, we mainly focus on large flagship (>70B parameters) and small-to-medium efficiency models. The large models include Qwen2-VL-72B (Alibaba, 2024), InternVL2-Llama3-76B (Chen et al., 2024d), LLaVA-OneVision-72B (Li et al., 2024a), and NVLM (Dai et al., 2024). The medium-scale models comprise Qwen2-VL-7B (Alibaba, 2024), Pixtral 12B (Mistral, 2024), Aria (Li et al., 2024d), InternVL2-8B (Chen et al., 2024d), Phi-3.5-Vision (Abdin et al., 2024), MiniCPM-V2.6 (Yao et al., 2024), LLaVA-OneVision-7B (Li et al., 2024a), Llama-3.2-11B Meta (2024), and Idefics3-8B-Llama3 (Laurençon et al., 2024).

**Evaluation pipeline.** MEGA-BENCH has diverse and flexible formats. To ensure the models have clear instructions on the output format, we provide all evaluated VLMs with a one-shot in-context example. For each query, we fill in a pre-defined prompt template with the task instructions written by the task annotators, the 1-shot example, and the concrete query question. Since this one-shot

Table 2: The main results of different models on the Core and Open-ended subset of MEGA-BENCH, with 440 and 65 tasks, respectively. We report the macro mean scores across all tasks in each set. The overall score is the weighted average of the Core and Open-ended scores. When computing the overall score, we use the higher Core score from 'w/o CoT' and 'w/ CoT'.

| Model | Eval Tier | Open Source | Core (rule eval) | | Open-ended (GPT eval) | Overall |
|---|---|---|---|---|---|---|
| | | | w/o CoT | w/ CoT | | |
| Claude-3.5-Sonnet (1022) (Anthropic, 2024b) | Flagship | No | 49.20 | 52.59 | **65.63** | **54.27** |
| GPT-4o (0513) (OpenAI, 2024a) | Flagship | No | **52.03** | **52.65** | 64.78 | 54.21 |
| Claude-3.5-Sonnet (0620) (Anthropic, 2024a) | Flagship | No | 48.80 | 50.41 | 63.74 | 52.13 |
| Gemini-1.5-Pro-002 (Google, 2024b) | Flagship | No | 46.99 | 48.22 | 58.58 | 49.55 |
| Gemini-1.5-Flash-002 (Google, 2024b) | Efficiency | No | **41.90** | **41.89** | 56.91 | **43.82** |
| GPT-4o mini (OpenAI, 2024b) | Efficiency | No | 39.85 | 40.77 | **58.65** | 43.07 |
| Qwen2-VL-72B (Alibaba, 2024) | Flagship | Yes | **46.41** | 45.42 | 56.40 | **47.70** |
| InternVL2-Llama3-76B (Chen et al., 2024d) | Flagship | Yes | 35.02 | 35.63 | 51.93 | 37.73 |
| LLaVA-OneVision-72B (Li et al., 2024a) | Flagship | Yes | 31.99 | 29.74 | 45.99 | 33.79 |
| NVLM-72B (Dai et al., 2024) | Flagship | Yes | 24.21 | 21.59 | 34.78 | 25.57 |
| Qwen2-VL-7B (Alibaba, 2024) | Efficiency | Yes | **34.80** | 32.93 | 43.96 | **35.98** |
| Pixtral-12B (Mistral, 2024) | Efficiency | Yes | 31.91 | 31.36 | 45.66 | 33.68 |
| Aria-MoE-25B (Li et al., 2024d) | Efficiency | Yes | 30.49 | 28.90 | **51.03** | 33.13 |
| InternVL2-8B (Chen et al., 2024d) | Efficiency | Yes | 25.96 | 24.09 | 39.79 | 27.74 |
| Phi-3.5-Vision-4B (Abdin et al., 2024) | Efficiency | Yes | 23.27 | 23.00 | 39.48 | 25.36 |
| MiniCPM-V2.6-8B (Yao et al., 2024) | Efficiency | Yes | 22.88 | 22.96 | 41.73 | 25.38 |
| LLaVA-OneVision-7B (Li et al., 2024a) | Efficiency | Yes | 22.41 | 21.36 | 33.98 | 23.90 |
| Llama-3.2-11B (Meta, 2024) | Efficiency | Yes | 10.04 | 16.00 | 31.73 | 18.02 |
| Idefics3-8B-Llama3 (Laurençon et al., 2024) | Efficiency | Yes | 11.12 | 8.96 | 32.11 | 13.82 |

example's primary purpose is to illustrate the output format, we allocate it a tiny portion of the total image budget. For each model, we conduct experiments with and without Chain-of-Thought (CoT) prompting (Wei et al., 2022) for the Core tasks. Full evaluation details are in §D. Our default evaluation pipeline focuses on models with multi-image support. To properly evaluate models trained mainly for single-image use cases, we create a single-image setting using the single-image tasks of MEGA-Bench. See §A for the detailed results and analyses of the single-image setting.

## 4.2 MAIN RESULTS WITH BREAKDOWN ANALYSIS

Table 2 presents the main evaluation results, while Figure 5 and Figure 6 are the fine-grained breakdowns enabled by MEGA-BENCH's multi-dimensional diversity. This section discusses some important findings, and the full breakdown results are in §E. We organize the evaluated models into two tiers: (1) The *Flagship Model Tier* compares the strongest performing models from each model's organization, (believed) with #params ≥ 70B. (2) The *Efficiency Model Tier* compares efficiency models from each model's organization, (believed) with #params ≤ 20B.

**Flagship models.** Unlike the results on recent benchmarks like MMMU-Pro (Yue et al., 2024b) where GPT-4o (0513) and Claude-3.5-Sonnet (0620) get close scores, GPT-4o (0513) outperforms Claude-3.5-Sonnet (0620) with a clear margin on MEGA-BENCH (> 2%). Investigating the breakdown results, we observe that GPT-4o (0513) wins in most applications/skills except for coding, math, and planning-related tasks, where the answers are typically in a "structured" output format ( Figure 5). The recent update of Claude-3.5-Sonnet (1022) makes improvements across almost all dimensions, especially in planning tasks and those with infographics/UI/photographs inputs, and slightly surpasses GPT-4o in the overall score (< 0.1%). The "planning" application keyword contains tasks like symbolic planning (Zhu et al., 2024), navigation (Ku et al., 2020), chess games (Fu et al., 2024b), puzzle games (e.g., maze, Sudoku), etc., and even the best models get low scores.

One typical observation of Claude-3.5-Sonnet models is that they tend to be meticulous and refuse to answer routine knowledge or commonsense questions, such as the name and nationality of famous actors. The bottom radar maps show that they fall behind in knowledge, information extraction, and commonsense reasoning compared to GPT-4o, partially because of this refusal behavior.

The evaluation results suggest that Qwen2-VL performs particularly well amongst open-source models of similar parameter sizes. In Figure 5, Qwen2-VL-72B gets a similar score to closed-source

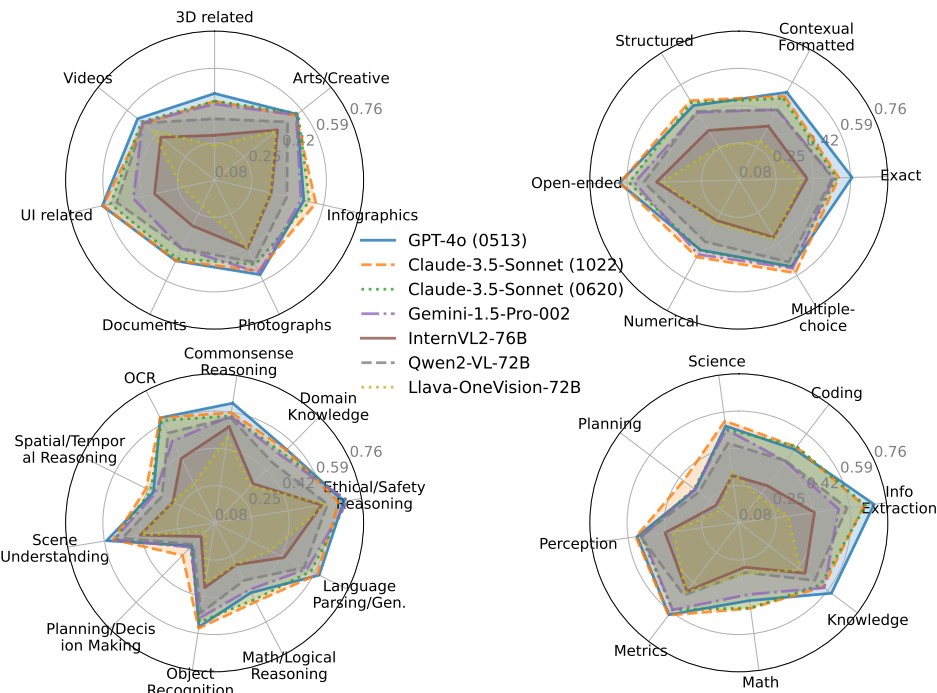

Figure 5: Fine-grained breakdown analysis of flagship models on four dimensions. From top-left to bottom-right: input format, output format, skills, and application.

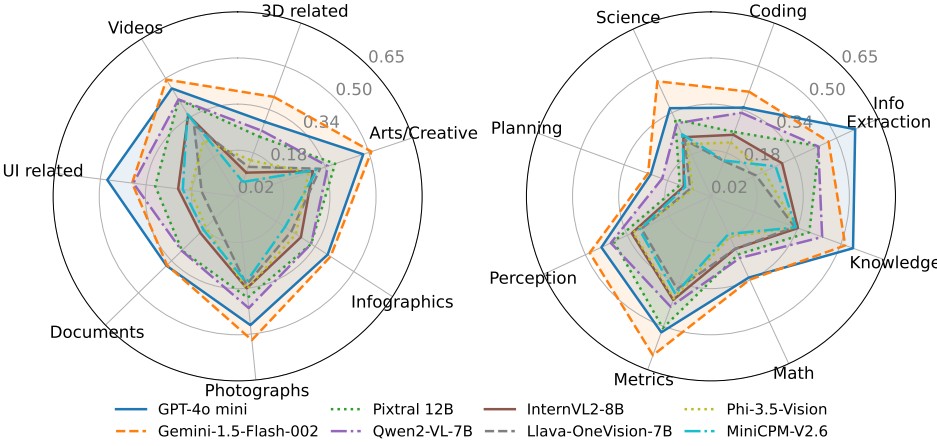

Figure 6: Fine-grained analysis of efficiency models on input format (left) and application (right).

models in the general perception category and outperforms Gemini-1.5-Pro-002 on information extraction tasks. Llava-OneVision-72B scores very low when the visual inputs are in "UI related" and "Document" formats while performing well on video inputs. This suggests a lack of OCR and language parsing abilities, which can be confirmed with its skills radar plot.

**Efficiency models.** Figure 6 analyzes the results on efficiency models. In general, Gemini-1.5-flash-002 has the best performance with exceptional scores in Science and Metrics applications. The Metrics keyword contains tasks such as rating the quality of GenAI results (He et al., 2024; Jiang et al., 2024b) and requires deep multimodal reasoning and commonsense. However, its performance on UI-related inputs and information extraction tasks falls behind GPT-4o mini.

**Chain-of-Thought.** An interesting finding is that the CoT prompt (See §D) effectively guides all proprietary models to generate a detailed reasoning process, and flagship-tier proprietary models all obtain better performance on the Core set. However, it has almost no effect on most open-source models. For example, the Qwen2-VL, InternVL2, and LLaVA-OneVision models rarely

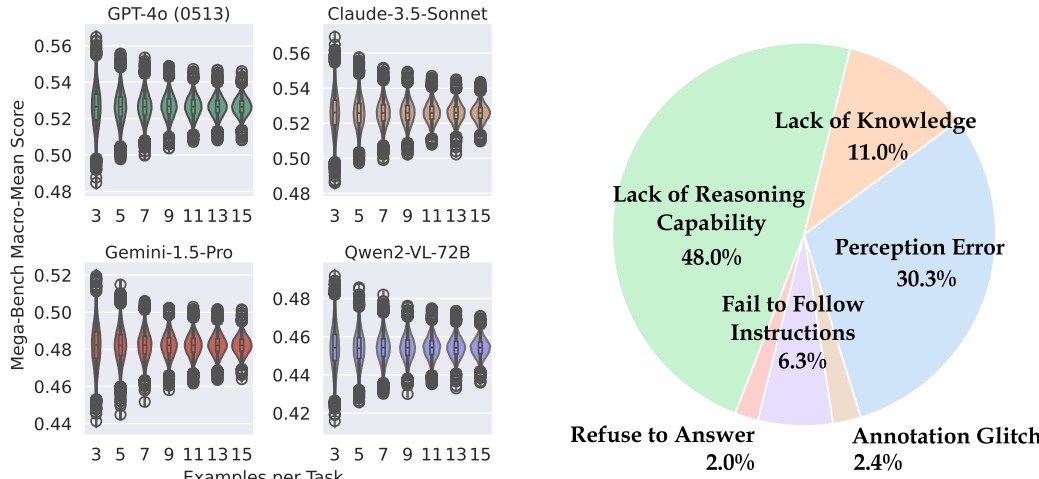

Figure 7: (Left) The bootstrap distribution of benchmark scores (w/ CoT prompting) with a gradually increased bootstrap sample size of the number of per-task examples. (Right) The task-wise error distribution of GPT-4o (0513) over a subset of 255 Core tasks.

produce reasoning with a CoT instruction, and sometimes get confused about the required format after generating the reasoning process, leading to a lower score on the Core set.

Some open-source models get comparatively low scores for their parameter count (e.g., Llama-3.2-11B, Idefics3). One typical difficulty is leveraging the one-shot example to understand the output format, which is alleviated with CoT because the prompt provides extra instructions on the output format beyond the one-shot example. The lack of multi-image ability is another bottleneck of these models, and §A presents a single-image setting of MEGA-BENCH with further analyses.

### 4.3 MORE ANALYSIS

**Number of samples per task.** As discussed in §1, one of MEGA-BENCH's goals is optimizing the inference cost while producing comprehensive breakdown analyses, making us prioritize the task diversity over examples per task in the benchmark construction process. To understand the robustness of the benchmark score with 15 examples per task, we obtained bootstrap distributions (Efron & Tibshirani, 1994; Hesterberg, 2011) of the model scores for our Core set with CoT. We did this by taking a random subset of the model's responses of size $n$ ($n = 3, 5, \ldots, 15$) with replacement for each task and calculating the task-level macro-mean scores. To ensure the bootstrap distribution was numerically stable, we ran 10,000 Monte Carlo simulations. Figure 7 (left) shows that the variance in model scores rapidly narrows as the number of examples per task increases. As the number of examples per task increases beyond 7, the marginal return in variance reduction diminishes.

**Error analysis.** To understand the limitations of state-of-the-art VLMs, we analyze the GPT-4o (0513) results by manually identifying the error types over a subset of 255 tasks from the Core set. We use the CoT setting since the reasoning process helps determine the error type. Figure 7 (right) presents the error distribution. For GPT-4o, the lack of various reasoning capabilities (e.g., symbolic reasoning for planning/coding tasks, spatial or temporal reasoning for complex perception tasks, etc.) is the dominating failure mode on MEGA-BENCH. Please refer to §F for the full definition of error types and detailed example-wise inspection results with different models.

## 5 CONCLUSION

This paper presents MEGA-BENCH, a comprehensive benchmark that scales multimodal evaluation to over 500 real-world tasks but at a manageable inference cost. By systematically organizing tasks across dimensions like skill, output format, and input type, we enable fine-grained analysis of multimodal models. Our evaluation of state-of-the-art VLMs revealed significant performance variations between models that previously seemed similar. MEGA-BENCH provides a new standard for multimodal evaluation, offering a robust analysis tool for model development.

ACKNOWLEDGMENT

We would like to thank Vector Institute for their support throughout this research. We also express our gratitude to Tung Vu from GreenNode for providing access to GPUs, which were instrumental in running some of the evaluation experiments. Additionally, we appreciate the contributions of all open-source language model providers, whose efforts have significantly propelled the advancement of research in this field.

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

# Table of Contents in Appendix

## A  SINGLE-IMAGE SETTING: RESULTS AND ANALYSES

Table 2 in the main paper focuses on models with multi-image support. However, some open-source models are only trained with single images. To provide a feasible evaluation setting for these models, we create a single-image (SI) setting using the single-image tasks in MEGA-BENCH, containing 273 and 42 tasks from the Core and Open-ended sets, respectively.

**Evaluation setup.** The Chain-of-Thought (CoT) prompting is used for Core SI tasks. To make the entire query contain only one image as needed by some single-image models, we drop the image input of the 1-shot demonstration example ("✗ demo im" column in the table). In this case, the 1-shot example only demonstrates the output format, which is necessary for inferring the correct answer. For those models already evaluated in Table 2, we calculate the task-level average scores on single-image tasks to obtain the "✓ demo im" results. Compared to Table 2, 3 single-image models are evaluated and added: Molmo-72B-0924 (Deitke et al., 2024), Molmo-7B-D-0924 (Deitke et al., 2024), and POINTS-Qwen2.5-7B (Liu et al., 2024b).

**Evaluation results.** Table 3 presents the evaluation results of the SI setting. The Core and Open-ended scores of the standard setting (with CoT prompting) are also in the table for reference. Some observations from the table are listed below:

• Single-image tasks are easier than multi-image tasks in general, and all models get higher scores in the SI setting than in the standard setting.

• GPT-4o has the best overall SI score, slightly higher than Claude 3.5 Sonnet (1022). Interestingly, GPT-4o mini overtakes Gemini-1.5-Flash-002 under the SI setting, suggesting that Gemini-1.5-Flash has pretty stable performance across different numbers of image inputs.

• NVLM-72B (Dai et al., 2024) has much better scores in the SI setting than in the standard setting, suggesting its training data might only contain single or a few images.

• Comparing the "✓ demo im" and "✗ demo im" results of open-source models, the image input in the 1-shot demonstration example is not well utilized by the models to better understand the task logic. Including the image input in the demonstration example makes the results much worse for models like Llama-3.2-11B.

More detailed breakdown results are available on our project page and the leaderboard (hosted with Hugging Face Spaces)

Table 3: The single-image (SI) setting results of MEGA-BENCH. The Core set evaluation uses Chain-of-Thought (CoT) prompting. The "demo img" means the image input of the 1-shot demonstration example. The "✓ demo im" directly takes the single-image subset average from the full results in Table 2. The "✗ demo im" means the 1-shot demonstration example only demonstrates the output format, and the entire query has a single image. We report "✓ demo im" alone for the proprietary models because they have good multi-image support. For open-source models, we do additional evaluations with the "✗ demo im" setting and use it to compute the overall score.

| Model | Core | Core SI | | Open | Open SI | | Overall SI |
| | | ✓ demo im | ✗ demo im | | ✓ demo im | ✗ demo im | |
|---|---|---|---|---|---|---|---|
| GPT-4o (0513) (OpenAI, 2024a) | 52.65 | **55.30** | - | 64.78 | 66.00 | - | **56.73** |
| Claude-3.5-Sonnet (1022) (Anthropic, 2024b) | 52.59 | 54.63 | - | 65.63 | **67.64** | - | 56.36 |
| Claude-3.5-Sonnet (0620) (Anthropic, 2024a) | 50.41 | 52.03 | - | 63.74 | 64.80 | - | 53.73 |
| Gemini-1.5-Pro-002 (Google, 2024b) | 48.22 | 49.14 | - | 58.58 | 58.15 | - | 50.34 |
| GPT-4o mini (OpenAI, 2024b) | 40.77 | **44.31** | - | 58.65 | **59.56** | - | **46.32** |
| Gemini-1.5-Flash-002 (Google, 2024b) | 41.89 | 43.48 | - | 56.91 | 57.87 | - | 45.40 |
| Qwen2-VL-72B (Alibaba, 2024) | 45.42 | **47.31** | 47.31 | 56.40 | 58.50 | 55.10 | **48.34** |
| InternVL2-Llama3-76B (Chen et al., 2024d) | 35.63 | 39.32 | 39.99 | 51.93 | 55.33 | **55.47** | 42.05 |
| Molmo-72B-0924 (Deitke et al., 2024) | - | - | 36.48 | - | - | 44.66 | 37.58 |
| NVLM-72B (Dai et al., 2024) | 21.59 | 31.19 | 32.99 | 34.78 | 48.67 | 44.69 | 34.55 |
| LLaVA-OneVision-72B (Li et al., 2024a) | 29.74 | 31.77 | 31.26 | 45.99 | 46.12 | 44.26 | 32.99 |
| Qwen2-VL-7B (Alibaba, 2024) | 32.93 | **35.04** | 35.39 | 43.96 | 45.87 | 45.17 | **36.69** |
| Pixtral-12B (Mistral, 2024) | 31.36 | 34.87 | 34.37 | 45.66 | 44.03 | 44.17 | 35.68 |
| Aria-MoE-25B (Li et al., 2024d) | 28.90 | 31.67 | 31.79 | 51.03 | **50.92** | 51.37 | 34.40 |
| InternVL2-8B (Chen et al., 2024d) | 24.09 | 27.19 | 27.65 | 39.79 | 40.94 | 39.39 | 29.21 |
| Phi-3.5-Vision-4B (Abdin et al., 2024) | 23.00 | 25.72 | 25.61 | 39.48 | 44.61 | 42.72 | 27.89 |
| POINTS-Qwen2.5-7B (Liu et al., 2024b) | - | - | 25.51 | - | - | 30.32 | 26.15 |
| MiniCPM-V2.6-8B (Yao et al., 2024) | 22.96 | 23.82 | 23.23 | 41.73 | 42.54 | 43.61 | 25.95 |
| LLaVA-OneVision-7B (Li et al., 2024a) | 21.36 | 22.70 | 23.68 | 33.98 | 36.44 | 38.71 | 25.69 |
| Qwen2-VL-2B (Alibaba, 2024) | 20.88 | 24.16 | 22.78 | 31.54 | 30.59 | 35.09 | 24.43 |
| Molmo-7B-D (Deitke et al., 2024) | - | - | 20.98 | - | - | 35.70 | 22.95 |
| Llama-3.2-11B (Meta, 2024) | 16.00 | 17.34 | 20.79 | 31.73 | 34.29 | 38.61 | 23.17 |
| Aquila-VL-2B-llava-qwen (Gu et al., 2024) | 16.00 | 16.98 | 20.77 | 24.57 | 24.58 | 31.47 | 22.20 |
| InternVL2-2B (Chen et al., 2024d) | 13.14 | 13.83 | 12.07 | 23.86 | 24.28 | 28.52 | 14.26 |
| Idefics3-8B-Llama3 (Laurençon et al., 2024) | 8.96 | 9.13 | 8.94 | 32.11 | 33.25 | 32.31 | 12.06 |

## B    DETAILS OF ANNOTATION PROTOCOLS

This section presents additional details of our task annotation pipeline and protocols, providing complete details for §3.1 of the main paper.

### B.1    THE UNIFIED ANNOTATION FORMAT

Figure 8 presents the annotation format designed and used in our annotation process. All annotated tasks share this unified structure, including task instruction, *optional* global media to provide context to all the questions (typically used in retrieval-related tasks). Additionally, each specific example contains distinct media path(s), a concrete question, and an answer with a single or multiple answer fields. Multi-field answers are organized as JSON structures.

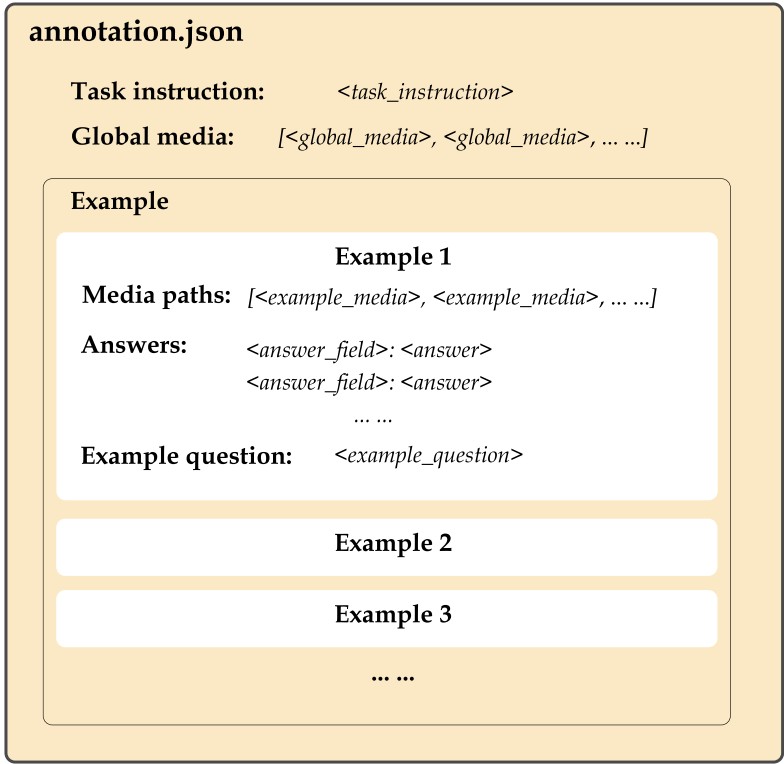

Figure 8: The structure of our task annotation format, which helps coordinate all task annotators and standardize the annotation format.

Our evaluation pipeline follows this format to convert the task information into concrete queries and feed them to the evaluated model. Based on this format, we establish an interactive annotation tool to ensure the tasks submitted by all annotators have the correct and unified format. Figure 9 demonstrates the GUI of the annotation tool.

### B.2    GENERAL TASK COLLECTION AND CREATION GUIDELINES

This subsection provides more detailed annotation guidelines for our annotators, complementing the descriptions in §3.1.

**Data source of each task.** There is no restriction on the data source as long as the annotator follows the copyright and license requirements of the original data. Below are three typical task types and their data sources:

(1). The task is designed entirely by the annotator, and the annotator looks for the image or video resources from the Internet or even using code/simulator;

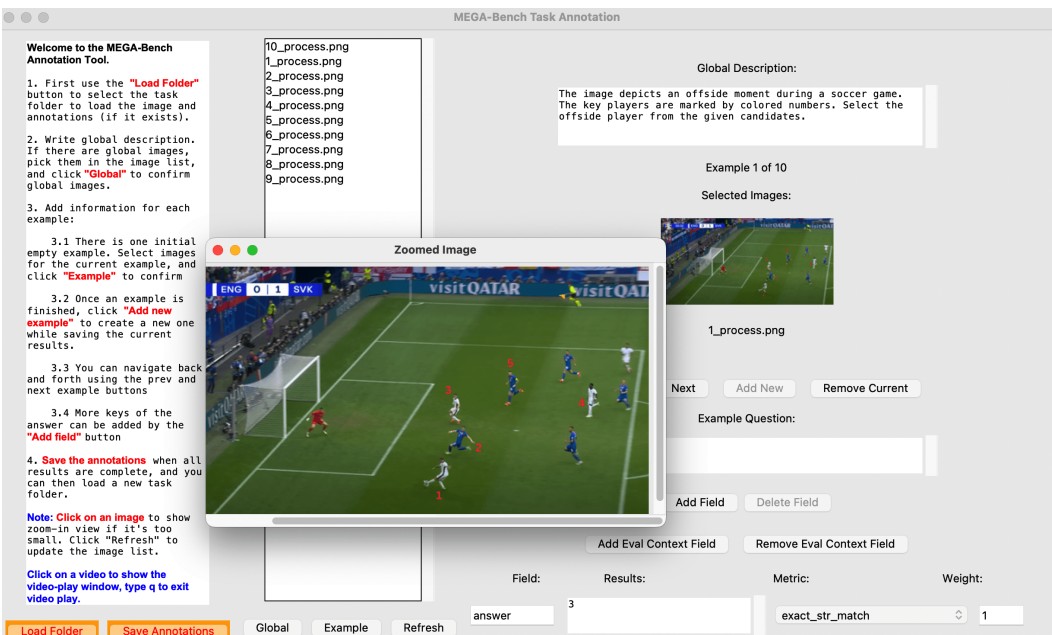

Figure 9: A screenshot of our GUI annotation tool.

(2). The task is inspired by existing benchmarks or datasets. The annotator collects the raw image/video data from existing datasets but does not use the original annotation. The annotator redesigns/repurposes the data by writing concrete task descriptions and creating new questions and answers, or using scripts to re-process the data for the designed task.

(3). The task is directly converted from existing benchmarks or datasets. The annotator randomly samples a subset from the existing benchmark, directly using its image/video and the annotation without redesign.

In our annotation process, the first two task types are encouraged. The task reviewers strictly control the number of the third type and reject the task if an annotator submits many tasks of the third type. Table 18 shows the detailed data source of all tasks in MEGA-BENCH.

**Output format and answer uniqueness.** We aim to cover diverse output formats in MEGA-BENCH. Therefore, we always require the task annotators to consider adapting the original dataset's answer format, especially avoiding unnecessary multiple-choice questions (many MCQs are unnatural and mainly for evaluation convenience). Notably, the annotator must provide sufficient context in the task description and per-example question so that the range of the correct answer is manageable and the task can be evaluated with a clearly defined metric.

**Metric specification.** When creating a task, the annotator must specify the corresponding evaluation metric. Since the metric implementation is in parallel to the task construction process, as described in §3.2, our GUI annotation tool (Figure 9) allows annotators to choose from existing metrics for each answer field of the task and assigns different weights to each field. When the desired metric is unavailable, the annotator chooses an "unsupported" metric type and writes down detailed metric specifications in the pull request. Our core contributors periodically check the needs of new metrics and implement them.

**Documentation.** When submitting the pull request, the annotator must write README documentation for each task. If the desired metric has not been implemented, the documentation should contain the specification described in the last point. Furthermore, the doc should record the data source (e.g., the Web, an existing dataset, etc.) and brief descriptions of the task. These descriptions are instrumental in helping the core contributors assign various keywords to the task and creating Table 18 to show the details of all tasks.

### B.3 TOOLS FOR COORDINATING ANNOTATION AND QUALITY CONTROL

As described in §3.1, we have two additional tools for coordinating the annotation process and maintaining the data. We present the details in this subsection.

**The GitHub repository for task organization.** We created a private GitHub repository for constructing MEGA-BENCH. The repository's main branch is protected, and all task submissions must go through pull requests (PRs). The core contributors serve as the task reviewers and discuss with task annotators in the pull request forum to ensure the task conforms to our data collection guidelines (§B.2). The code of our evaluation pipeline, including the model query and score computation, is maintained in the same repository. The core contributors submit pull requests to support different VLMs and add new evaluation metrics, and these PRs are cross-reviewed by other core contributors.

We also actively use the repository's Issues forum to report bugs in annotation or metric implementation so the corresponding contributors can get notified and work on the fix. At the end of the annotation process, our repository has 685 pull requests and 40 issues. 277 out of the 685 PRs are for task submission, indicating that many annotators submit task groups with more than one task in each PR. Other PRs are mainly for the evaluation pipeline and bug fixing.

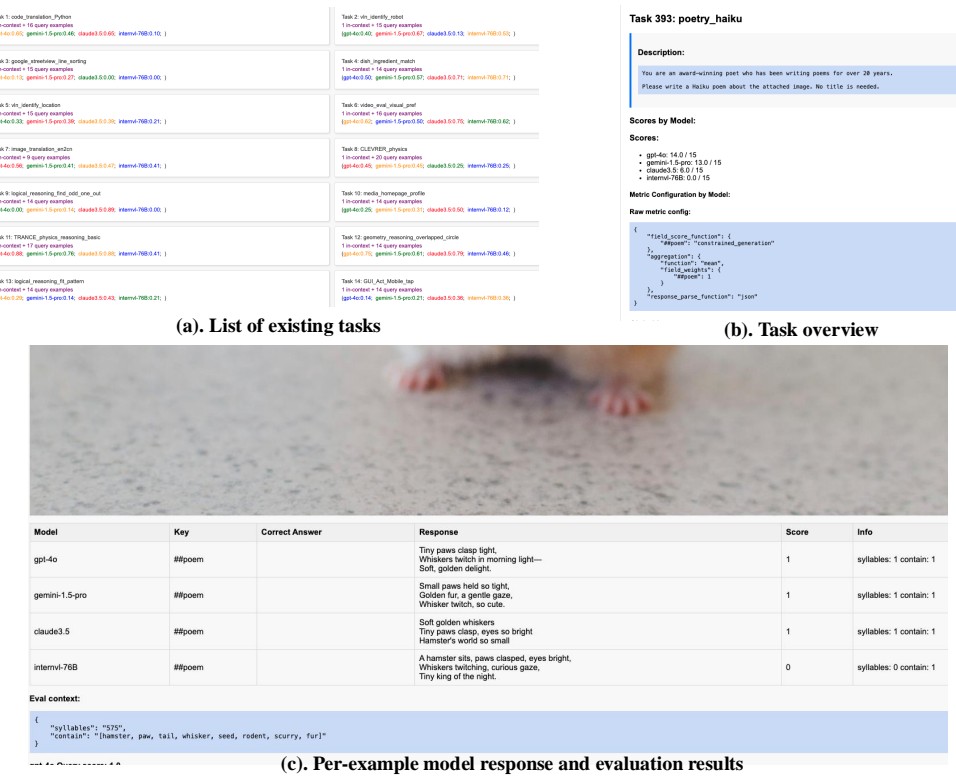

Figure 10: Illustrations of our task visualization page.

**Task visualization web page.** We developed a simple visualization web page and periodically synchronized the evaluation results of existing tasks on the page. The page provides several benefits: 1) it allows the core contributors to keep track of the overall annotation process, 2) it helps the annotators understand the capability of state-of-the-art VLMs, so that they can adjust the task difficulty accordingly, and 3) it facilitates the checking of the potential annotation glitches or metric bugs, significantly improving the overall quality of MEGA-BENCH. Figure 10 shows screenshots of the visualization page taken during the benchmark construction process. Note that the task names in the figure might not align with the final names in the paper. In our project page, we will provide a similar visualization page for users to interactively inspect the behaviors of different VLMs.

## C  TAXONOMY TREE AND MULTI-DIMENSIONAL KEYWORDS

This section presents the full details of our application-based taxonomy tree and the multi-dimensional keywords.

### C.1  DETAILS OF THE TAXONOMY STRUCTURE

Table 4 shows the detailed structure of our application-driven task taxonomy. The first level defines the broad scope of use cases. At the second level, tasks are categorized into more specific domains. These first two levels guide the annotation process of our benchmark and are gradually updated/refined in the annotation process. The third level lists the concrete names of tasks or task groups. If the third-level node is a task group, the number of concrete tasks under this group is shown in the parenthesis.

Table 4: Details of taxonomy of MEGA-BENCH.

| Level-2 Tasks | Leaf Tasks (at Level-3 or deeper) | # Tasks |
|---|---|---|
| **Coding** | | |
| Code Debugging | Stackoverflow Debug Qa, Code Error Line Identification | 2 |
| Code Generation | Document Conversion (8 tasks), Programming Problems (4 tasks), Visualization With Code | 13 |
| Code Translation | Code Translation Easy, Code Translation Python, Code Translation Hard, Code Translation Advanced | 4 |
| Code Understanding | Symbolic Graphics Programming (2 tasks), Webpage Code Understanding, Code Add Tag, Code Match (5 tasks), Code Output (3 tasks) | 12 |
| **Information Extraction** | | |
| App Function Understanding | App Layout Understanding Leetcode, App Layout Understanding Youtube, App Layout Understanding Amazon, App Layout Understanding Word, App Layout Understanding Notes, App Layout Understanding Ppt, App Layout Understanding Alipay, App Layout Understanding Instagram, App Layout Understanding Zoom, App Layout Understanding Excel, App Layout Understanding Iphone Settings, App Layout Understanding Tiktok, App Layout Understanding Twitter | 13 |
| Compound Search and Calculate | Cheapest Flight Identification, Weather Info Retrieval, Stock Info Retrieval, Game Platform Support Identification, Top Rated Hotel Identification, Movie Info Retrieval, Top Video Creator Identification, Highest Discount Game Price Identification, Newspaper Page Parse And Count, Remaining Playback Time Calculation | 10 |
| Detailed Manual Understanding | Multi Lingual Manual Explanation Scooter Spanish, Multi Lingual Manual Explanation Scooter Arabic, Multi Lingual Manual Explanation Scooter French, Multi Lingual Manual Explanation Scooter Chinese, Multi Lingual Manual Explanation Scooter Russian | 5 |
| Multimodal QA | Multilingual News Qa, Product Ocr Qa, Large Image (3 tasks), Gui Chat (2 tasks), Realworld Qa En2cn, Star Object Interaction Video, Video Qa (7 tasks) | 16 |

Table 4 – continued from previous page

| Level-2 Tasks | Leaf Tasks (at Level-3 or deeper) | # Tasks |
|---|---|---|
| Search by Attribute wo Calculate | Coco Ood Global Image Retrieval By Query Property, Places365 Similar Scene Retrieval, Booking Web Recommendation, Game Info Retrieval, Media Homepage Profile, Movie Retrieval By Actor, Music Info Retrieval, Tv Show Retrieval By Character | 8 |
| Structured Parsing | Multilingual Movie Info Parsing, Movie Info Parsing, Stock Info Parsing, Music Info Parsing, Multilingual Game Info Parsing, Ocr Article Authors, Youtube Video Info Parsing, Tv Show Info Parsing, Ocr Resume School Plain, Image Translation En2cn, Booking Web Rating, Weather Info Parsing, Game Info Parsing, Weather Map Climate Type Temperature Parsing, Hotel Booking Confirmation Parsing, Entertainment Web Game Style | 16 |
| Summarization | Video Summary, Video Short Title, Video2notes, Video Content Reasoning | 4 |
| **Knowledge** | | |
| Arts | Poetry Generation (7 tasks), Ascii Art 30 | 8 |
| Fact Checking | Background Change, Out Of Context, Text Entity Replace, Text Style, Face Attribute Edit, Face Swap, Interpret Force Perspective Illusion, Clip Stable Diffusion Generate, Unusual Images, Forensic Detection Of Different Images, Veracity, Distinguish Ai Generated Image | 12 |
| Human and Culture | Cultural Vqa, Human Relationship Reasoning, Sign Language, Ishihara Test, Safety And Norm (13 tasks), Video Content Follow Up, Emotion And Intent Understanding (9 tasks), Theory Of Minds (2 tasks), Hashtag Recommendation | 30 |
| World Knowledge | Dish Ingredient Match, Music (6 tasks), Insect Order Classification, Signage Navigation, Song Title Identification From Lyrics, Logo And Sign (3 tasks), Chinese Idiom Recognition, Ruozhiba (6 tasks), Font Recognition, Traffic Accident Analysis, Multiple State Identification (4 tasks), Worldle, Location Vqa, Daily (2 tasks), Ancient Map Understanding, Rocks Samples Compare, Painting (2 tasks), Memorization (4 tasks), Soccer Offside, Deciphering Oracle Bone, Actor Character And Famous People (3 tasks), Landmark And Buliding (3 tasks), Defeasible Reasoning | 47 |
| **Mathematics** | | |
| Algebra | Algebra | 1 |
| Calculus | Scibench Calculus Wo Solution | 1 |
| Functions | Math Parity, Math Breakpoint, Math Convexity Value Estimation | 3 |
| General | Math Exams V, Theoremqa, Math | 3 |

Table 4 – continued from previous page

| Level-2 Tasks | Leaf Tasks (at Level-3 or deeper) | # Tasks |
|---|---|---|
| Geometry | Geometry Reasoning Count Line Intersections, Geometry Length, Geometry Reasoning Nested Squares, Geometry Transformation, Geometry Reasoning Overlapped Circle, Geometry Area, Geometry Reasoning Grid, Polygon Interior Angles, Geometry Solid, Geometry Analytic, Geometry Descriptive | 11 |
| Graph Theory | Graph Shortest Path Kamada Kawai, Graph Shortest Path Planar, Graph Connectivity, Graph Theory, Graph Isomorphism, Graph Hamiltonian Cycle, Graph Hamiltonian Path, Graph Chordless Cycle, Topological Sort, Graph Maxflow | 10 |
| Number Theory | Counterfactual Arithmetic | 1 |
| Numeric Reasoning | Clevr Arithmetic, Iconqa Count And Reasoning, Number Comparison | 3 |
| **Metrics** | | |
| Generated Image Eval | Autorater Artifact, Autorater Control, Autorater Artifact Reason, Autorater Aesthetics, Autorater Unmask, Autorater Subject, Autorater 3d Model Texturing, Autorater Semantics, Autorater Motion Guided Editing, Autorater Mask | 10 |
| Generated Video Eval | Video Eval Visual Pref, Generated Video Artifacts, Video Eval Factual Pref, Video Eval Dynamic Pref | 4 |
| Paper Review | Paper Review Writing, Paper Review Rating, Paper Review Acceptance | 3 |
| Quality Assessment | Vizwiz Quality Accessment For Blind | 1 |
| Reward Models | Reward Models T2i Reward, Reward Models I2t Reward | 2 |
| **Perception** | | |
| 3D understanding | Adapted Cvbench Depth, Relative Depth Of Different Points, Visual Prediction Rater Depth Estimation, Visual Prediction Rater Novel View Synthesis, Pokemon 3d Recognition, Av View Identification, Multiview Reasoning Camera Moving, 3d Indoor Scene Text Bbox Prediction, Google Streetview Circle Reasoning, Google Streetview Direction Understanding, Video Motion Matching Real 3d, Video Motion Matching 3d Real, Visual Prediction Rater 3d Assembled Quality Understanding, Visual Prediction Rater Surface Normal Estimation, Visual Prediction Rater Plane Segmentation, 3d Indoor Scene Text Bbox Selection, Google Streetview Circle Sorting | 17 |
| Counting | Ad Count Detection, Adapted Cvbench Count, Av Vehicle Multiview Counting, Counting Multi Image, Av Human Multiview Counting, Shape Composition Shapes, Counting Single Image, Clevrer Video Moving Object Count, Shape Composition Colours | 9 |
| Diagram and Document Understanding | Diagram (23 tasks), Document (9 tasks), Table Qa (6 tasks) | 38 |

Table 4 – continued from previous page

| Level-2 Tasks | Leaf Tasks (at Level-3 or deeper) | # Tasks |
|---|---|---|
| Image Segmentation | Visual Prediction Rater Openable Part Segmentation, Visual Prediction Rater Panoptic Segmentation, Visual Prediction Rater Semantic Segmentation | 3 |
| Multimodal Captioning | Video Detail Description, Guess Image Generation Prompt, Docci Image Description Long, Tweets Captioning, Image Captioning With Additional Requirements | 5 |
| Multimodal Constrained Captioning | Contain Contain Images, Contain Repeat Length, Multi Contain Repeat Position Only Length, Contain Length, Contain Position Images, Contain Position Length, Xor Images, Multi Contain Repeat, Contain Contain Length, Multi Contain Position Only | 10 |
| Object and Scene Understanding | Autonomous Driving Scene Analysis, Super Clevr Scene Understanding, Functionality Matching In Different Objects, Visual Dialog Image Guessing, Nlvr2 Two Image Compare Qa, Egocentric Analysis Single Image, Clevrer Object Existence Video, Snli Ve Visual Entailment, Ocr Open Ended Qa, Semantic Matching Of Two Images | 10 |
| Physical Understanding | Physical Reasoning (8 tasks), Lighting And Shading (2 tasks) | 10 |
| Spatial Understanding | Adapted Cvbench Relation, Visual Correspondance In Two Images, 2d Image Jigsaw Puzzle Easy, Geometry Plot Position Relationship, Adapted Cvbench Distance, Video Grounding Spatial, Egocentric Spatial Reasoning | 7 |
| Temporal Understanding | Video To Camera Trajectory Retrieval, Sceneqa Scene Transition Video, Video Segments Reordering, Video Action Recognition, Action Sequence Understanding, Google Streetview Line Sorting, Next Action Prediction, Perception Test Video Action Count, Google Streetview Line Reasoning, Video Camera Motion Description, Video Grounding Temporal, Web Action Prediction, Cam Traj To Video Selection, Sta Action Localization Video | 14 |
| Visual Recognition | Face Identity Matching, Rocks Samples Identify, Animal Pose Estimation, License Plate Recognition, Image Style Recognition, Long String Letter Recognition, Coco Object Detection By Query Property, Widerface Face Count And Event Classification, Handwritten Math Expression Extraction, Geometry Reasoning Circled Letter, Av Multicamera Tracking Predict Bbox, Ascii Art Understanding, Face Keypoint Detection, Extract Webpage Headline, Waldo, Geographic Remote Sensing Land Cover, Signboard Identification, Long String Number Recognition, Waybill Number Sequence Extraction, Single Person Pose Estimation, Coco Person Detection, Places365 Scene Type Classification | 22 |
| **Planning** | | |

Table 4 – continued from previous page

| Level-2 Tasks | Leaf Tasks (at Level-3 or deeper) | # Tasks |
|---|---|---|
| Agents and Planning | Wikihow Complex Task Completion, Navigation (6 tasks), Gui Operation (18 tasks), Calendar Schedule Suggestion, Symbolic Planning (13 tasks) | 39 |
| Puzzles and Games | Logical Reasoning Find Odd One Out, Logical Reasoning Fit Pattern, Perception Test Object Shuffle Video, Board Games (12 tasks), Bongard Problem, Number Puzzle Kakuro 5x5, Mensa Iq Test, Arc Agi, Mnist Pattern, Number Puzzle Sudoku, Move Pos To Pos Hanoi 4 Pole, Pictionary (5 tasks), Annoying Word Search, Logical Reasoning 2d Views Of 3d Shapes, Maze 2d 8x8, Crossword Mini 5x5, Rebus, Icon Arithmetic Puzzle, Iq Test Open Ended, Ball Cup Swap 3, Logical Reasoning 2d Folding | 36 |
| Reordering | Perception Test Video Character Order, Comic Page Ordering, Recipe Image Ordering | 3 |
| **Science** | | |
| Chemistry | Chemistry Exams V, Science Molecule Chemistry | 2 |
| Life Sciences | Biology Exams V, Medical (15 tasks) | 16 |
| Physics | Circuit Diagram Understanding, Mmmu Physics Chemistry Selected, Science Basic Physics, Physics Exams V | 4 |
| STEM | Mmmu Pro Exam Screenshot, Scibench W Solution Open Ended, Arxiv Vqa, Tqa Textbook Qa, Question Solution Solving, Quizlet Question Solving, Scibench Fundamental Wo Solution | 7 |

## C.2 STATISTICS OF EACH KEYWORD DIMENSION

Figure 2 of the main paper presented the overall keyword distribution. As a complement, Table 5 provides more detailed statistics. Each of the five dimensions contains multiple keywords, and for each keyword, we explicitly show the number of related tasks and the total number of samples.

Table 5: Number of tasks and samples across the five dimensions, with detailed breakdown into each keyword.

| Dimension | Keywords (number of tasks, num of samples) |
|---|---|
| Skills | Object Recognition (303, 4755), OCR (137, 2239), Language Parsing & Gen. (154, 2509), Scene & Event Understanding (154, 2467), Math & Logical Reasoning (109, 1910), Commonsense & Social Reasoning (51, 855), Ethical & Safety Reasoning (15, 245), Domain-Specific Knowledge/Skills (77, 1387), Spatial & Temporal Reasoning (152, 2437), Planning & Decision Making (37, 577) |
| Input Format | User Interface (93, 1517), Text-rich Image & Doc (82, 1294), Diagrams & Visualizations (101, 1718), Videos (43, 698), Artistic & Creative (32, 542), Photographs (143, 2248), 3D Related (11, 169) |
| Output Format | Contextual Formatted (98, 1514), Structured (110, 1714), Exact (83, 1279), Numerical (49, 862), Open-ended (80, 1454), Multiple Choice (85, 1363) |
| Input Number | 6-8 images (21, 314), 9-image+ (41, 623), 1-image (315, 5228), Video (43, 698), 4-5 images (34, 520), 2-3 images (51, 802) |
| Application | Information_Extraction (72, 1124), Planning (78, 1239), Coding (31, 474), Perception (145, 2313), Metrics (20, 309), Science (29, 574), Knowledge (97, 1605), Mathematics (33, 547) |

# D    EVALUATION DETAILS

This section details our evaluation settings, including the prompt template design, model query details, and evaluation metrics.

Figure 11: The prompt template structure without Chain-of-Thought (CoT).

## D.1    PROMPT TEMPLATE

We provide the concrete prompt template in Figure 11 and Figure 12. All the information organized by the prompt template is serialized by our evaluation pipeline before sending queries to the evaluated model.

The non-CoT prompt instructs the VLM to strictly follow the one-shot example, directly producing the answer without additional text. In contrast, the CoT prompt instructs the VLM to output step-by-step reasoning before providing the final answer, and the model must strictly separate the reasoning process from the final answer.

Note that our prompt sets different formats for single-field and multi-field outputs. Single-field answers must be explicitly indicated by the "Answer: ..." format so that our output parser can robustly locate and extract the model's answer. Multi-field answers are in JSON format, and our JSON parser can robustly extract the JSON-style answer from the entire response without the "Answer: ..." format.

## D.2    MODEL QUERY DETAILS

Since the evaluated VLMs have different context windows, we must tailor the number of query images or video frames for each model. We implement an image/video pre-processing pipeline that follows the settings listed in Table 6 to sub-sample the input images and videos. We allocate different budgets for in-context examples and the query. Since the in-context examples (we use a

**Prompt Template**

*<task_instruction>*
*<global_media>,<global_media>, ... ...*
**Demonstration example(s) of the task:**

**Example 1:** *<example_media>, <example_media>, ...*
**Example Question:** *<example_question>*
**Example Response:** *[PLEASE OUTPUT YOUR REASONING]*

{
  *<answer_field>: <answer>*          *Answer: <answer>*
  *<answer_field>: <answer>*          *// if single-field*
  *... ...*
} *// if multi-field*

*... ...*

**Example n:**          *// if n-shot > 1*

**Answer the new question below. The last part of your response should be of the following format: "Answer: <YOUR ANSWER>" (without angle brackets) where YOUR ANSWER is your answer, following the same task logic and output format of the demonstration example(s). For your answer, do not output additional contents that violate the specified format. Think step by step before answering.**

*<question_media>,<question_media>, ... ...*

**Question:** *<question>*

Figure 12: The prompt template structure for the Chain-of-Thought (CoT) setting

one-shot example) mainly help models understand the task logic and the output format, we reserve most of the image budget for the query. Images or video frames surpassing the budget are discarded. To make sure the open-source models can run smoothly, we implement a fallback strategy, which reduces the image budget to decrease the number of input tokens if the model's maximum context length is exceeded.

For images or video frames with a longer side larger than 1000 pixels, we resize the longer side to 1000 without changing the aspect ratio before sending them to the evaluated model. Each

### D.3 LLM-ASSISTED METRICS

The LLM-assisted metric instructs a multimodal LLM to evaluate VLM's response by providing a detailed evaluation prompt. When submitting a task with open-ended answers that cannot be evaluated by rule-based metrics, the annotator is asked to write down a detailed evaluation prompt for the LLM judge following the prompt format in Figure 13.

Concretely, the task annotator decides if the LLM judge should consider the question's visual input when evaluating the model's response. If yes, then the query media (images or videos) will be passed to the LLM as well (we use GPT-4o-0806 as a multimodal judge model). For most tasks, the LLM judge can do a proper evaluation by comparing the model's response with the reference answer, and the visual media is not needed. The task annotator also writes a thorough evaluation criteria, explaining to the judge model the meaning of each score range, which is important to get reliable evaluation results.

Table 6: The maximum number of images and the budget for the in-context example per model.

| Model | Max # of images | In-context example budget |
|---|---|---|
| GPT-4o (0513) (OpenAI, 2024a) | 64 | 8 |
| Claude-3.5-Sonnet (1022) (Anthropic, 2024a) | 64 | 8 |
| Claude-3.5-Sonnet (0620) (Anthropic, 2024a) | 64 | 8 |
| Gemini-1.5-Pro-002 (Google, 2024b) | 128 | 16 |
| Gemini-1.5-Flash-002 (Google, 2024b) | 128 | 16 |
| GPT-4o Mini (OpenAI, 2024b) | 64 | 8 |
| Qwen2-VL-72B (Alibaba, 2024) | 24 | 2 |
| InternVL2-Llama3-76B (Chen et al., 2024d) | 24 | 4 |
| NVLM-72B Dai et al. (2024) | 32 | 4 |
| Molmo-72B-0924 (Deitke et al., 2024) | 1 | 0 |
| LLaVA-OneVision-72B (Li et al., 2024a) | 28 | 4 |
| Qwen2-VL-7B (Alibaba, 2024) | 18 | 2 |
| Pixtral-12B (Mistral, 2024) | 48 | 6 |
| Aria-MoE-25B (Li et al., 2024d) | 32 | 4 |
| POINTS-Qwen2.5-7B (Liu et al., 2024b) | 1 | 0 |
| InternVL2-8B (Chen et al., 2024d) | 18 | 2 |
| Phi-3.5-Vision (Abdin et al., 2024) | 16 | 2 |
| MiniCPM-V2.6 (Yao et al., 2024) | 64 | 8 |
| Molmo-7B-D (Deitke et al., 2024) | 1 | 0 |
| LLaVA-OneVision-7B (Li et al., 2024a) | 20 | 4 |
| Llama-3.2-11B (Meta, 2024) | 32 | 4 |
| Idefics3-8B-Llama3 (Laurençon et al., 2024) | 20 | 2 |
| Qwen2-VL-2B (Alibaba, 2024) | 16 | 2 |
| InternVL2-2B (Chen et al., 2024d) | 18 | 2 |
| Aquila-VL-2B-llava-qwen (Gu et al., 2024) | 8 | 1 |

---

**LLM-Assisted Metrics Prompt Template**

*<media>,<media>, ... ...    // if judge with image*
*<evaluation_criteria>       // defined by task annotator*
**Reference:**         *<reference_answer>*
**Model Response:**    *<model_response>*
**(Optional):**        *<per_example_label>*
                       *//  some tasks require per-example criteria*
*Please output your score in the following format:*
***Score**: <single_number>,*
***Score explanation**: <detailed_explanations>*

Figure 13: The prompt template structure for LLM-Assisted Metrics

At the end of the prompt, a pre-defined scoring format instruction is attached, ensuring the judge model outputs a score between 1 and 10 and an explanation for the score.

## D.4 RULE-BASED METRICS

We have over 40 highly customized rule-based metrics to evaluate the Core set of MEGA-BENCH. Basic metrics like "extract string match" and "simple string match" (which ignores punctuation and special characters) are first added to the supported metric set. New metrics are implemented when our task annotators submit new tasks requiring uncovered metrics. In the end, we get 45 customized tasks, as shown in Table 7. The usage distribution is long-tail because many metric implementations are triggered by a single novel task.

Table 7: All metrics used in MEGA-BENCH.

| Metric Name | Usage Count (# tasks) |
| --- | --- |
| Exact String Match | 198 |
| GPT-4o as Judge | 64 |
| Simple String Match | 61 |
| Multi Reference Phrase Evaluation | 25 |
| Constrained Generation | 18 |
| Set Equality | 15 |
| Sequence Equality | 15 |
| General Single Numerical Match | 14 |
| Exact String Match Case Insensitive | 14 |
| Sequence Accuracy Case Insensitive | 13 |
| Symbolic Planning Test | 13 |
| String Set Equality Comma | 9 |
| Normalized RMSE | 8 |
| Program Judge | 8 |
| Set Precision | 5 |
| Dictionary Equality | 4 |
| String Set Equality Line Break | 4 |
| Sequence Coordinates Similarity | 3 |
| LaTeX Expression Equality | 3 |
| Jaccard Index Case Insensitive | 3 |
| Jaccard Index | 3 |
| Normalized Bounding Box IOU Tuple | 2 |
| Number Relative Difference Ratio | 2 |
| XML Bounding Box IOU | 2 |
| Dictionary Exact String Match Aggregate Recall | 2 |
| Boxed Single Numerical Match | 2 |
| Positive Integer Match | 2 |
| Chess Move List Jaccard Index | 2 |
| Code Result Exact String Match | 1 |
| Normalized Bounding Box IOU Single | 1 |
| Normalized Bounding Box IOU Sequence | 1 |
| Normalized Similarity Damerau-Levenshtein | 1 |
| Near String Match | 1 |
| XML Normalized Point Distance | 1 |
| Dictionary Precision | 1 |
| Text with LaTeX Expression Equality | 1 |
| Angle Sequence Float RMSE | 1 |
| XML Normalized Point in Bounding Box | 1 |
| Longest Common List Prefix Ratio | 1 |
| Sequence Equality Case Insensitive | 1 |
| Set Equality Case Insensitive | 1 |
| GLEU (Chinese) | 1 |
| ASCII Art GPT-4O Judge | 1 |
| Dictionary Jaccard Aggregate Jaccard | 1 |
| Dictionary Normalized Bounding Box IOU Tuple Aggregate Jaccard | 1 |

## D.5 ANSWER EXTRACTION FROM MODEL RESPONSE

For Core tasks, our rule-based evaluation metrics compare the model's answer with a ground-truth answer or some ground-truth constraints. Therefore, an answer extraction step is necessary to separate the final answer from the reasoning process and other irrelevant texts. We implement robust extraction logic for different types of outputs based on the format specified in the prompt template:

**Single-field answer.** We first reduce the answer by the "Answer: ..." pattern. If this pattern does not exist, we take the entire response. Since many VLMs do not strictly follow the format instructions, we have specific and extra processing for different output formats to improve robustness. Some typical examples are: 1) For multiple-choice outputs, we locate the exact letter or index choice using sophisticated regular expressions, which excludes any potential parenthesis or accompanying texts; 2) For code outputs, we extract the code from the potential code blocks; 3) For structured

outputs, we parse the structural data into the proper Python data structures (list, set, dictionary, etc.), with tolerance on minor syntax errors (e.g., we automatically fix wrong quotes).

**Multi-field answer.** Since the prompt requires the model to output the final answer in JSON format, we implement a robust JSON parser to locate the JSON structure in the raw response and convert the JSON structure into the corresponding Python data structure.

If our comprehensive answer extraction fails to obtain any meaningful final answer from the model response, we consider the model as "fail to follow instructions".

# E   COMPLETE MULTI-DIMENSIONAL BREAKDOWN RESULTS

This section provides the full breakdown results over the five dimensions of MEGA-BENCH, complementing section 4 of the main paper.

## E.1   BREAKDOWN RESULTS ON THE SKILL DIMENSION

Table 8: Average scores for each model on the *skill* dimension. The best-performing model in each category is **in-bold**, and the second best is underlined.

| Model | CASR | DKAS | EASR | LUAG | MALR | ORAC | PADM | SAEU | SATR | TR |
|---|---|---|---|---|---|---|---|---|---|---|
| Claude-3.5-Sonnet (1022) (Anthropic, 2024b) | 59.1 | 54.9 | 65.7 | 60.8 | **48.9** | 56.9 | **29.1** | 55.1 | **43.2** | 62.2 |
| GPT-4o (0513) (OpenAI, 2024a) | **63.5** | **55.1** | 68.0 | **61.6** | 44.2 | 56.3 | 22.9 | **58.2** | 39.4 | 62.2 |
| Claude-3.5-Sonnet (0620) (Anthropic, 2024a) | 57.6 | 52.8 | 69.7 | 57.5 | 47.7 | 54.1 | 23.8 | 54.5 | 40.8 | 60.8 |
| Gemini-1.5-Pro-002 (Google, 2024b) | 57.5 | 51.4 | **69.8** | 55.3 | 42.6 | 52.0 | 23.9 | 54.7 | 38.5 | 50.2 |
| Gemini-1.5-Flash-002 (Google, 2024b) | 55.9 | 44.8 | 63.8 | 49.9 | 34.4 | 46.3 | 19.0 | 51.0 | 34.5 | 43.4 |
| GPT-4o mini (OpenAI, 2024b) | 55.7 | 41.9 | 69.0 | 51.7 | 34.1 | 44.9 | 19.4 | 46.7 | 29.4 | 49.0 |
| Qwen2-VL-72B (Alibaba, 2024) | 56.8 | 46.3 | 60.5 | 53.9 | 37.8 | 49.8 | 22.0 | 50.9 | 35.1 | 54.4 |
| InternVL2-Llama3-76B (Chen et al., 2024d) | 52.6 | 33.3 | 57.8 | 43.7 | 29.8 | 38.2 | 17.0 | 42.7 | 29.5 | 41.3 |
| LLaVA-OneVision-72B (Li et al., 2024a) | 47.8 | 31.7 | 60.1 | 36.7 | 29.5 | 36.2 | 13.9 | 42.1 | 29.6 | 28.3 |
| NVLM-72B (Dai et al., 2024) | 40.9 | 25.8 | 45.6 | 29.4 | 26.4 | 24.0 | 6.7 | 22.8 | 15.7 | 32.2 |
| Qwen2-VL-7B (Alibaba, 2024) | 49.4 | 33.3 | 52.2 | 40.3 | 28.2 | 37.1 | 14.7 | 41.1 | 27.6 | 40.2 |
| Pixtral 12B (Mistral, 2024) | 41.9 | 32.8 | 56.9 | 38.3 | 28.3 | 34.6 | 10.6 | 37.8 | 26.8 | 37.8 |
| Aria-MoE-25B (Li et al., 2024d) | 49.4 | 32.8 | 58.1 | 40.0 | 27.6 | 32.6 | 11.9 | 37.8 | 24.8 | 35.7 |
| InternVL2-8B (Chen et al., 2024d) | 39.7 | 27.1 | 47.0 | 32.0 | 24.1 | 28.2 | 8.3 | 32.6 | 23.2 | 28.1 |
| Phi-3.5-Vision (Abdin et al., 2024) | 36.8 | 24.1 | 46.7 | 28.7 | 21.7 | 25.5 | 8.9 | 30.5 | 21.5 | 24.8 |
| MiniCPM-V2.6 (Yao et al., 2024) | 40.7 | 23.7 | 48.8 | 30.0 | 18.3 | 26.0 | 8.7 | 31.8 | 19.7 | 25.0 |
| LLaVA-OneVision-7B (Li et al., 2024a) | 36.8 | 24.5 | 45.0 | 25.6 | 19.0 | 25.2 | 6.7 | 30.0 | 21.8 | 19.1 |
| Qwen2-VL-2B (Alibaba, 2024) | 31.3 | 20.8 | 41.4 | 25.7 | 17.6 | 22.2 | 6.2 | 26.5 | 17.3 | 23.7 |
| Llama-3.2-11B (Meta, 2024) | 32.3 | 17.7 | 42.6 | 19.6 | 13.3 | 19.1 | 6.6 | 22.4 | 15.4 | 14.3 |
| Aquila-VL-2B-llava-qwen (Gu et al., 2024) | 26.6 | 18.6 | 35.2 | 17.9 | 16.8 | 18.4 | 4.5 | 22.0 | 16.2 | 12.4 |
| InternVL2-2B (Chen et al., 2024d) | 24.0 | 14.8 | 34.2 | 16.9 | 13.9 | 14.5 | 1.7 | 18.5 | 13.0 | 12.1 |
| Idefics3-8B-Llama3 (Laurençon et al., 2024) | 19.2 | 17.9 | 28.6 | 17.3 | 13.3 | 14.5 | 4.2 | 14.7 | 10.2 | 11.6 |

The abbreviations used in the table above are explained in the following table:

Table 9: Abbreviation list of the keywords in the *skill* dimension.

| Abbreviation | Skill |
|---|---|
| CASR | Commonsense and Social Reasoning |
| DKAS | Domain-Specific Knowledge and Skills |
| EASR | Ethical and Safety Reasoning |
| LUAG | Language Understanding and Generation |
| MALR | Mathematical and Logical Reasoning |
| ORAC | Object Recognition and Classification |
| PADM | Planning and Decision Making |
| SAEU | Scene and Event Understanding |
| SATR | Spatial and Temporal Reasoning |
| TR | Text Recognition (OCR) |

## E.2 BREAKDOWN RESULTS ON THE INPUT FORMAT DIMENSION

Table 10: Average scores for each model on the *input format* dimension. The best-performing model in each category is **in-bold**, and the second best is underlined.

| Model | 3MAAI | AACC | DADV | P | TIAD | UIS | V |
|---|---|---|---|---|---|---|---|
| Claude-3.5-Sonnet (1022) (Anthropic, 2024b) | 44.2 | 55.6 | **55.6** | 54.3 | 48.9 | 60.5 | 49.5 |
| GPT-4o (0513) (OpenAI, 2024a) | **47.8** | 56.4 | 50.0 | **56.1** | **49.1** | **60.8** | **53.2** |
| Claude-3.5-Sonnet (0620) (Anthropic, 2024a) | 44.3 | **57.0** | 52.6 | 51.0 | 48.0 | 56.9 | 50.9 |
| Gemini-1.5-Pro-002 (Google, 2024b) | 42.9 | 55.8 | 48.7 | 55.0 | 42.9 | 46.3 | 50.3 |
| Gemini-1.5-Flash-002 (Google, 2024b) | 38.5 | 50.5 | 40.1 | 51.7 | 36.0 | 38.7 | 49.0 |
| GPT-4o mini (OpenAI, 2024b) | 29.4 | 47.6 | 38.9 | 46.5 | 36.2 | 47.2 | 45.5 |
| Qwen2-VL-72B (Alibaba, 2024) | 36.2 | 50.8 | 42.1 | 49.8 | 42.9 | 54.0 | 49.9 |
| InternVL2-Llama3-76B (Chen et al., 2024d) | 28.7 | 45.0 | 34.7 | 42.9 | 31.4 | 36.3 | 39.6 |
| LLaVA-OneVision-72B (Li et al., 2024a) | 23.9 | 44.0 | 34.6 | 42.5 | 21.3 | 23.4 | 44.5 |
| NVLM-72B (Dai et al., 2024) | 5.7 | 34.7 | 30.3 | 32.6 | 21.7 | 23.9 | 0.0 |
| Qwen2-VL-7B (Alibaba, 2024) | 26.2 | 34.8 | 32.2 | 40.7 | 29.0 | 38.2 | 41.1 |
| Pixtral 12B (Mistral, 2024) | 24.0 | 37.5 | 32.2 | 37.1 | 28.8 | 30.7 | 41.0 |
| Aria-MoE-25B (Li et al., 2024d) | 19.6 | 36.1 | 32.4 | 37.3 | 27.8 | 28.3 | 42.9 |
| InternVL2-8B (Chen et al., 2024d) | 10.9 | 29.4 | 28.0 | 33.9 | 20.1 | 22.8 | 34.8 |
| Phi-3.5-Vision (Abdin et al., 2024) | 15.4 | 27.9 | 26.1 | 34.1 | 17.5 | 18.7 | 24.7 |
| MiniCPM-V2.6 (Yao et al., 2024) | 7.6 | 31.0 | 21.6 | 31.8 | 18.6 | 21.2 | 35.3 |
| LLaVA-OneVision-7B (Li et al., 2024a) | 13.0 | 32.0 | 24.2 | 32.6 | 13.3 | 14.7 | 31.0 |
| Qwen2-VL-2B (Alibaba, 2024) | 13.4 | 24.9 | 19.6 | 28.8 | 16.3 | 19.1 | 25.2 |
| Llama-3.2-11B (Meta, 2024) | 6.4 | 25.2 | 16.9 | 24.9 | 11.5 | 11.9 | 21.2 |
| Aquila-VL-2B-llava-qwen (Gu et al., 2024) | 10.1 | 19.7 | 19.4 | 24.6 | 11.4 | 7.5 | 21.4 |
| InternVL2-2B (Chen et al., 2024d) | 11.9 | 14.9 | 16.3 | 20.1 | 10.5 | 5.7 | 19.0 |
| Idefics3-8B-Llama3 (Laurençon et al., 2024) | 4.0 | 18.4 | 16.2 | 14.9 | 11.4 | 10.1 | 16.2 |

The abbreviations used in the table above are explained in the following table:

Table 11: Abbreviation list of the keywords in the *input formats* dimension.

| Abbreviation | Input Format |
|---|---|
| 3MAAI | 3D Models and Aerial Imagery |
| AACC | Artistic and Creative Content |
| DADV | Diagrams and Data Visualizations |
| P | Photographs |
| TIAD | Text-Based Images and Documents |
| UIS | User Interface Screenshots |
| V | Videos |

### E.3 BREAKDOWN RESULTS ON THE OUTPUT FORMAT DIMENSION

Table 12: Average scores for each model on the *output format* dimension. The best-performing model in each category is **in-bold**, and the second best is underlined.

| Model | C | E | M | N | O | S |
|---|---|---|---|---|---|---|
| Claude-3.5-Sonnet (1022) (Anthropic, 2024b) | 51.9 | 53.9 | **57.8** | **48.2** | 62.4 | **50.7** |
| GPT-4o (0513) (OpenAI, 2024a) | **53.9** | **59.9** | 54.5 | 44.6 | **62.7** | 48.0 |
| Claude-3.5-Sonnet (0620) (Anthropic, 2024a) | 50.7 | 52.8 | 54.6 | 44.9 | 58.4 | 49.7 |
| Gemini-1.5-Pro-002 (Google, 2024b) | 44.9 | 51.5 | 55.4 | 46.9 | 55.8 | 44.4 |
| Gemini-1.5-Flash-002 (Google, 2024b) | 38.7 | 44.8 | 47.8 | 37.0 | 54.5 | 39.9 |
| GPT-4o mini (OpenAI, 2024b) | 41.2 | 44.2 | 39.9 | 36.3 | 57.1 | 39.1 |
| Qwen2-VL-72B (Alibaba, 2024) | 44.7 | 51.0 | 52.0 | 40.3 | 51.6 | 45.0 |
| InternVL2-Llama3-76B (Chen et al., 2024d) | 36.3 | 39.4 | 38.8 | 29.2 | 45.8 | 34.8 |
| LLaVA-OneVision-72B (Li et al., 2024a) | 28.7 | 37.1 | 39.9 | 30.7 | 42.9 | 25.9 |
| NVLM-72B (Dai et al., 2024) | 22.9 | 27.9 | 18.5 | 23.3 | 32.2 | 27.9 |
| Qwen2-VL-7B (Alibaba, 2024) | 34.3 | 35.2 | 39.9 | 32.7 | 39.1 | 34.3 |
| Pixtral 12B (Mistral, 2024) | 30.8 | 36.4 | 30.1 | 32.1 | 41.7 | 31.9 |
| Aria-MoE-25B (Li et al., 2024d) | 30.9 | 29.3 | 32.8 | 30.9 | 45.2 | 30.4 |
| InternVL2-8B (Chen et al., 2024d) | 25.1 | 27.4 | 30.3 | 22.4 | 35.4 | 25.2 |
| Phi-3.5-Vision (Abdin et al., 2024) | 21.8 | 25.7 | 26.0 | 21.4 | 36.5 | 21.4 |
| MiniCPM-V2.6 (Yao et al., 2024) | 23.5 | 25.5 | 29.3 | 20.8 | 36.5 | 17.8 |
| LLaVA-OneVision-7B (Li et al., 2024a) | 20.3 | 25.4 | 28.0 | 22.0 | 31.3 | 18.3 |
| Qwen2-VL-2B (Alibaba, 2024) | 16.2 | 20.0 | 25.7 | 22.0 | 30.2 | 21.0 |
| Llama-3.2-11B (Meta, 2024) | 12.4 | 15.8 | 19.3 | 15.0 | 30.0 | 16.4 |
| Aquila-VL-2B-llava-qwen (Gu et al., 2024) | 11.9 | 18.5 | 22.1 | 19.9 | 23.3 | 12.3 |
| InternVL2-2B (Chen et al., 2024d) | 11.3 | 15.5 | 21.3 | 16.0 | 21.4 | 5.7 |
| Idefics3-8B-Llama3 (Laurençon et al., 2024) | 14.0 | 7.1 | 11.6 | 9.8 | 29.9 | 10.6 |

The abbreviations used in the table above are explained in the following table:

Table 13: Abbreviation list of keywords in the *output formats* dimension.

| Abbreviation | Output Format |
|---|---|
| C | Contextual Formatted Text |
| E | Exact Text |
| M | Multiple Choice |
| N | Numerical Data |
| O | Open-ended Output |
| S | Structured Output |

### E.4 BREAKDOWN RESULTS ON THE APPLICATION DIMENSION

Table 14: Average scores for each model on the *application* dimension. The best-performing model in each category is **in-bold**, and the second best is underlined.

| Model | C | I | K | M | M2 | P | P2 | S |
|---|---|---|---|---|---|---|---|---|
| Claude-3.5-Sonnet (1022) (Anthropic, 2024b) | 51.7 | 65.9 | 56.6 | **47.6** | **61.2** | **55.6** | **39.9** | **55.1** |
| GPT-4o (0513) (OpenAI, 2024a) | 50.3 | **70.6** | **61.4** | 44.0 | 61.0 | 55.1 | 33.2 | 52.8 |
| Claude-3.5-Sonnet (0620) (Anthropic, 2024a) | **51.9** | 66.6 | 55.1 | 47.5 | 58.1 | 53.2 | 33.8 | 51.3 |
| Gemini-1.5-Pro-002 (Google, 2024b) | 43.5 | 54.2 | 57.2 | 41.2 | 58.2 | 52.5 | 33.4 | 51.2 |
| Gemini-1.5-Flash-002 (Google, 2024b) | 40.4 | 46.6 | 51.2 | 33.7 | 60.1 | 48.0 | 25.2 | 45.7 |
| GPT-4o mini (OpenAI, 2024b) | 34.6 | 56.7 | 54.0 | 32.9 | 51.8 | 43.6 | 24.2 | 35.5 |
| Qwen2-VL-72B (Alibaba, 2024) | 43.7 | 58.1 | 51.7 | 31.2 | 49.7 | 53.6 | 31.2 | 44.9 |
| InternVL2-Llama3-76B (Chen et al., 2024d) | 29.5 | 43.1 | 46.3 | 28.7 | 47.4 | 42.2 | 21.3 | 30.0 |
| LLaVA-OneVision-72B (Li et al., 2024a) | 23.2 | 30.8 | 43.6 | 31.6 | 48.1 | 38.4 | 18.2 | 31.7 |
| NVLM-72B (Dai et al., 2024) | 23.9 | 22.8 | 37.2 | 24.5 | 18.9 | 30.2 | 8.0 | 24.9 |
| Qwen2-VL-7B (Alibaba, 2024) | 32.7 | 42.7 | 42.8 | 25.6 | 42.5 | 40.0 | 20.0 | 29.9 |
| Pixtral 12B (Mistral, 2024) | 25.7 | 43.0 | 38.1 | 24.2 | 50.2 | 38.9 | 13.6 | 31.3 |
| Aria-MoE-25B (Li et al., 2024d) | 28.5 | 38.3 | 41.0 | 26.2 | 39.7 | 37.8 | 14.3 | 29.7 |
| InternVL2-8B (Chen et al., 2024d) | 24.7 | 29.1 | 33.9 | 22.1 | 40.0 | 32.1 | 12.2 | 24.6 |
| Phi-3.5-Vision (Abdin et al., 2024) | 21.9 | 22.4 | 33.3 | 17.6 | 39.5 | 31.6 | 8.9 | 21.9 |
| MiniCPM-V2.6 (Yao et al., 2024) | 15.3 | 26.7 | 33.2 | 16.5 | 37.8 | 29.2 | 11.7 | 25.7 |
| LLaVA-OneVision-7B (Li et al., 2024a) | 15.2 | 19.3 | 32.7 | 22.1 | 36.0 | 28.5 | 9.8 | 23.7 |
| Qwen2-VL-2B (Alibaba, 2024) | 17.0 | 25.2 | 26.6 | 16.4 | 31.0 | 27.6 | 7.0 | 21.1 |
| Llama-3.2-11B (Meta, 2024) | 5.8 | 17.3 | 28.1 | 13.9 | 25.4 | 19.9 | 8.1 | 16.3 |
| Aquila-VL-2B-llava-qwen (Gu et al., 2024) | 13.3 | 9.5 | 24.1 | 20.7 | 29.3 | 20.7 | 5.9 | 21.1 |
| InternVL2-2B (Chen et al., 2024d) | 11.3 | 8.7 | 21.2 | 11.0 | 33.3 | 17.0 | 4.1 | 16.9 |
| Idefics3-8B-Llama3 (Laurençon et al., 2024) | 9.1 | 14.7 | 17.6 | 13.2 | 14.6 | 14.6 | 5.4 | 22.7 |

The abbreviations used in the table above are explained in the following table:

Table 15: Abbreviation list of keywords in the *applications* dimension .

| Abbreviation | Application |
|---|---|
| C | Coding |
| I | Information-Extraction |
| K | Knowledge |
| M | Mathematics |
| M2 | Metrics |
| P | Perception |
| P2 | Planning |
| S | Science |

## E.5 BREAKDOWN RESULTS ON THE VISUAL INPUT NUMBER DIMENSION

Table 16: Average scores for each model on the *visual input number* dimension. The best-performing model in each category is **in-bold**, and the second best is underlined.

| Model | 1 | 2I | 4I | 6I | 9OM | V |
|---|---|---|---|---|---|---|
| Claude-3.5-Sonnet (1022) (Anthropic, 2024b) | 56.4 | 48.8 | 48.3 | 46.3 | **59.1** | 49.5 |
| GPT-4o (0513) (OpenAI, 2024a) | **56.7** | 49.1 | 45.0 | **47.5** | 53.4 | **53.2** |
| Claude-3.5-Sonnet (0620) (Anthropic, 2024a) | 53.7 | **49.3** | 44.2 | 46.3 | 54.1 | 50.9 |
| Gemini-1.5-Pro-002 (Google, 2024b) | 50.3 | 45.5 | **48.9** | 39.1 | 53.7 | 50.3 |
| Gemini-1.5-Flash-002 (Google, 2024b) | 44.3 | 42.0 | 42.3 | 33.7 | 43.7 | 49.0 |
| GPT-4o mini (OpenAI, 2024b) | 46.3 | 37.0 | 24.7 | 33.6 | 43.1 | 45.5 |
| Qwen2-VL-72B (Alibaba, 2024) | 49.2 | 45.2 | 36.7 | 31.0 | 54.7 | 49.9 |
| InternVL2-Llama3-76B (Chen et al., 2024d) | 41.5 | 31.5 | 24.4 | 20.3 | 34.8 | 39.6 |
| LLaVA-OneVision-72B (Li et al., 2024a) | 34.8 | 34.2 | 25.0 | 20.7 | 28.1 | 44.5 |
| NVLM-72B (Dai et al., 2024) | 36.8 | 23.3 | 3.8 | 0.0 | 0.0 | 0.0 |
| Qwen2-VL-7B (Alibaba, 2024) | 37.7 | 33.0 | 26.4 | 19.4 | 37.5 | 41.1 |
| Pixtral 12B (Mistral, 2024) | 37.1 | 31.0 | 25.8 | 19.7 | 16.6 | 41.0 |
| Aria-MoE-25B (Li et al., 2024d) | 35.8 | 27.3 | 19.8 | 21.1 | 27.1 | 42.9 |
| InternVL2-8B (Chen et al., 2024d) | 30.1 | 25.3 | 17.7 | 15.4 | 19.9 | 34.8 |
| Phi-3.5-Vision (Abdin et al., 2024) | 27.8 | 28.5 | 20.2 | 12.5 | 14.3 | 24.7 |
| MiniCPM-V2.6 (Yao et al., 2024) | 26.3 | 22.3 | 17.9 | 14.0 | 23.6 | 35.3 |
| LLaVA-OneVision-7B (Li et al., 2024a) | 25.5 | 24.1 | 17.8 | 14.8 | 13.8 | 31.0 |
| Qwen2-VL-2B (Alibaba, 2024) | 25.0 | 21.3 | 17.4 | 7.7 | 10.5 | 25.2 |
| Llama-3.2-11B (Meta, 2024) | 19.6 | 18.6 | 13.5 | 14.6 | 7.3 | 21.2 |
| Aquila-VL-2B-llava-qwen (Gu et al., 2024) | 18.2 | 23.3 | 19.0 | 11.1 | 1.2 | 21.4 |
| InternVL2-2B (Chen et al., 2024d) | 15.2 | 15.8 | 17.7 | 3.7 | 5.8 | 19.0 |
| Idefics3-8B-Llama3 (Laurençon et al., 2024) | 14.8 | 12.3 | 12.2 | 10.1 | 9.3 | 16.2 |

The abbreviations used in the table above are explained in the following table:

Table 17: Abbreviation list of keywords in the *visual input number* dimension.

| Abbreviation | Input Number |
|---|---|
| 1 | 1-image |
| 2I | 2-3 images |
| 4I | 4-5 images |
| 6I | 6-8 images |
| 9OM | 9-image or more |
| V | video |

## F    DETAILED INSPECTION OF MODEL BEHAVIOURS ON MEGA-BENCH

To complement §4.3 of the main paper, this section presents a case study analysis of the error types of different models on different tasks in MEGA-BENCH. We use similar error categories as in MMMU (Yue et al., 2024a) and MMT-Bench (Ying et al., 2024):

• **Perception Error**: VLMs fail to recognize or perceive the content of interest in the query image(s). Perception errors indicate the

• **Lack of Knowledge**: VLMs lack the domain-specific knowledge to answer specialized questions, such as identifying the taxonomic order of an insect.

• **Lack of (Reasoning) Capability**: VLMs lack the necessary capabilities to solve the task, mainly related to various reasoning abilities, such as logical reasoning, counting, spatial or temporal reasoning, symbolic reasoning for code or various programs, and so on. This is a broad type that covers many errors. One typical case for this error type is that the models can accurately follow instructions and perceive the visual inputs but struggle with the required reasoning process, leading to incorrect answers.

• **Refuse to Answer**: VLMs refuse to answer questions that they believe to involve sensitive content.

• **Fail to Follow Instructions**: VLMs fail to correctly understand instructions and provide wrong answers. The tasks in MEGA-BENCH usually have more instructions on the answer format compared to previous benchmarks. A typical error pattern is not comprehending the required format, thus providing answers with incorrect formats or generating irrelevant responses. This error type is much more common in open-source models.

Figure 14 to Figure 33 shows the case study for samples from different tasks. We use distinct colors to highlight the tags in each task sample. We borrow the error case analysis template from MMMU (Yue et al., 2024a) while adding the keywords information of MEGA-BENCH. We mainly focus on the flagship proprietary models. The Claude-3.5 in these figures refers to the Claude-3.5-Sonnet (0620) model.

# List of Figures Depicting Model Behaviors

---

**Coding: Code Debugging**

Task Name: Code Error Line Identification

Given a piece of program code, identify the most obvious error. Output the line number on the left where the error occurs. Attention: Do not invoke any external compilers or related programs.
**Question:** Where the error occurs?

Keywords: **Domain-Specific Knowledge and Skills, Text-Based Images and Documents, Mathematical and Logical Reasoning, Multiple Choice**

```
1    void Topsort(int a[NUM][NUM], int TopNum[NUM])
2    {   int S[NUM],  Indegree[NUM];
3        int  Counter = 0, top, n, i, j;
4        int V;
5        top= -1;
6        n=NUM;
7        for (j=0; j<n; j++) {
8            Indegree[j]=0;
9            for (i=0; i<n; i++)
10               if (a[i][j]!=0) Indegree[j]++;
11           if ( Indegree[j] == 0 )   S[++top]=j;
12       }
13       while (top>=0) {
14           V = S[top--];
15           TopNum[ V ] = ++ Counter;
16           for (j=0; j<n; j++)
17               if ( a[V][j]!=0)
18                   if (Indegree[--j] == 0 )  S[++top]=j;
19       }
20       if ( Counter!=n )  printf( "Graph has a cycle" );
21   }
```

**Claude-3.5:** 1. The function declaration and variable initializations seem correct. 2. The first nested loop (lines 7-12) appears to b e properly structured. 3. The while loop (lines 13-19) also seems structurally correct. 4. However, within this while loop, there's an issue on line 18: 5. This error is the most obvious and significant in the code, as it will likely cause runtime errors or incorrect results. Therefore, the most obvious error occurs on line 18.
**Answer: 18** (score: 1.0)

**Error Category: Lack of Capability**
**GPT-4o:** The most obvious error in the code is the use of `printf` instead of `printf` in line 20. The correct function name should be `printf`.
**Answer: 20** (score: 0.0)

**Correct Answer: 18**

Figure 14: A sample error case of Coding (subfield: Code Debugging). Source:Web
Back to List of Figures

---

### Coding: Code Understanding

**Task Name:** Symbolic Graphics Programs Computer Aided Design

Given a symbolic graphics program, the task is to answer a related question by understanding the program.
**Question:** How many visible screws or screw holes are there on the CAD object?

**Keywords:** **Mathematical and Logical Reasoning, Contextual Formatted Text, Domain-Specific Knowledge and Skills, Text-Based Images and Documents**

```
SOL;
Arc:(130,126,64,1);Line:(221,126);Arc:(223,128,64,1);
Line:(223,176);Arc:(221,178,64,1);Line:(130,178);
Arc:(128,176,64,1);Line:(128,128);
SOL;
Circle:(139,138,3);
SOL;
Circle:(139,161,3);
SOL;
Circle:(176,161,10);
SOL;
Circle:(212,138,3);
SOL;
Circle:(212,161,3);
Ext:(192,64,192,34,128,62,189,207,128,Newbody,One-sided);
SOL;
Arc:(131,125,64,1);Line:(220,125);Arc:(223,128,64,1);
Line:(223,217);Arc:(220,220,64,1);Line:(131,220);
Arc:(128,217,64,1);Line:(128,128);
SOL;
Circle:(176,173,18);
Ext:(192,64,192,74,128,78,107,224,128,Join,One-sided);EOS
```

**Claude-3.5: 1. In CAD programs, screws or screw holes are often represented by circles.**
**2. Looking at the program, we can see several Circle commands: …**
**Answer: 4** (score: 1.0)

**Error Category: Lack of Capability**
**GPT-4o:** To determine the number of visible screws or screw holes in the CAD object, we need to identify the instances of circles in the program, as circles typically represent screws or screw holes in CAD drawings …
**Answer: 6** (score: 0.0)

**Error Category: Lack of Capability**
**Gemini-1.5-pro:** The code shows six circles that could be screw holes and two "Ext" commands that specify extrusions. Extrusions can be used to create screw holes.
**Answer: 8** (score: 0.0)

**Correct Answer: 4**

Figure 15: A sample error case of Coding (subfield: Code Understanding).
Source:Web

**Coding: Code_Understanding**

Task Name: Code Output Result

Given several images of program code, output the program's execution result. Attention: Do not invoke any external compilers or related programs. Place the code output in a Markdown result code block.
**Question:** What's the output of the program?

Keywords:

**OCR, Mathematical and Logical Reasoning
Domain-Specific Knowledge, Text-Based Images and Documents**

```rust
use std::sync::{Arc, Mutex};
use std::thread;

fn main() {
    let data = Arc::new(Mutex::new(vec![1, 2, 3]));
    let mut handles = vec![];

    for i in 0..3 {
        let data_clone = Arc::clone(&data);
        let handle = thread::spawn(move || {
            let mut vec = data_clone.lock().unwrap();
            vec[i] += i;
        });
        handles.push(handle);
    }

    for handle in handles {
        handle.join().unwrap();
    }

    println!("Final data: {:?}", *data.lock().unwrap());
}
```

**Error Category: Lack of Capability**
**Claude 3.5 :** Let's analyze this Rust code step by step:
1. We start with a vector [1, 2, 3] wrapped in an Arc<Mutex<...>>.
2. We create an empty vector `handles` to store thread handles.
3. We spawn 3 threads (for i in 0..3):
- Each thread gets a clone of the Arc.
- Each thread locks the mutex and increments the first element of the vector.
4. After spawning, we join all threads, ensuring they complete.
5. Finally, we print the contents of the vector.
Let's trace the changes:
- Initial state: [1, 2, 3]
- Thread 1: [2, 2, 3]
- Thread 2: [3, 2, 3]
- Thread 3: [4, 2, 3]
The final state of the vector will be [4, 2, 3]. The println! macro will output this vector.
**Answer: Final data: [4, 2, 3]** (score: 0.0)

**Correct Answer: Final data: [1, 3, 5]**

Figure 16: A sample error case of Coding (subfield: Code Understanding).
Source:Web
Back to List of Figures

**Information Extraction: Multimodel QA**

Task Name: Object Interaction Video

The task involves identifying objects a person interacts with in various videos based on the provided questions. You need to output the object name.
**Question:** Which object was taken by the person?

Keywords:       **Contextual Formatted Text, Scene Understanding, Scene and Event Understanding, Videos**

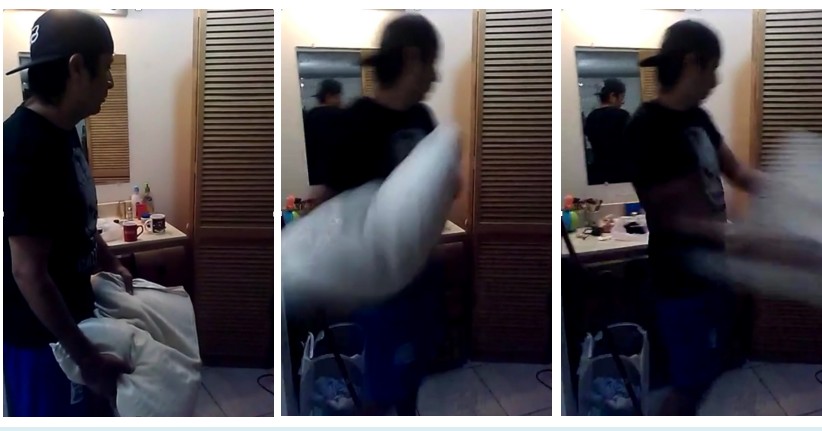

**GPT-4o: Answer: pillow** (score: 1.0)

**Error Category: Perception Error**
**Claude-3.5:** After carefully analyzing the sequence of images, I can see that the person in the video is interacting with a hat. In the earlier frames, the person is holding what appears to be a white or light-colored object, which is likely laundry or bedding. However, as the sequence progresses, we see the person in front of a mirror, and they appear to be adjusting or handling a hat. In the final frames, there's a motion that suggests the person is tossing or throwing the hat upwards.
**Answer: hat** (score: 0.0)

**Correct Answer: pillow**

Figure 17: A sample error case of Information Extraction (subfield: Multimodel QA).
Source: MVBench (Li et al., 2024e) and STAR (Wu et al., 2024)
Back to List of Figures

---

**Information Extraction : Detailed Manual Understanding**

Task Name: Multi-lingual Manual Explanation Scooter (Chinese)

根据给定的用户手册图片内容，使用中文回答问题。涉及操作上的问题，给出解决方法。涉及人身安全上的问题给出合理建议。

**Question:** 我骑这个车送孩子上学方便吗？

Keywords:                   **Document, Open-ended**

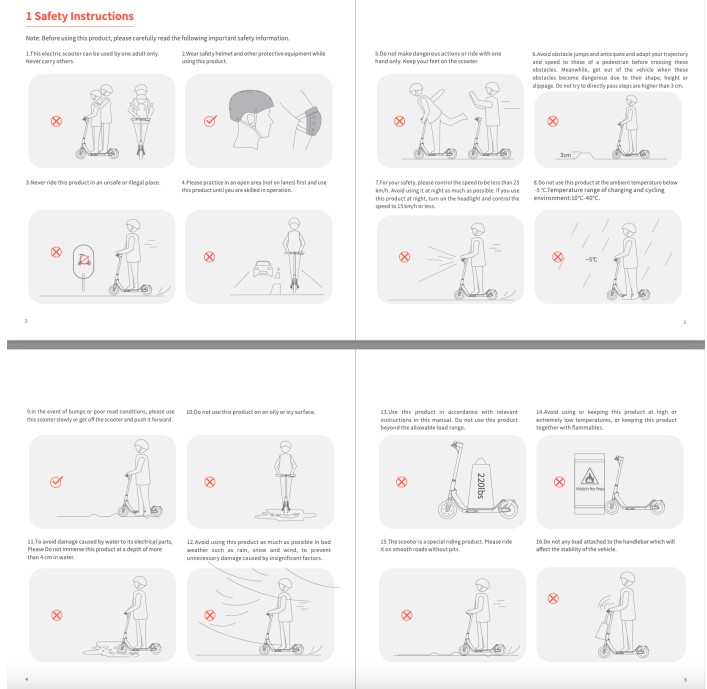

**GPT-4o : Answer:** 不建议使用这款滑板车送孩子上学。根据用户手册的安全说明，这款电动滑板车只能由一名成人使用，且不允许载人。此外，手册中明确指出不要进行危险动作或单手骑行，这样会增加安全风险。因此，为了您和孩子的安全，建议选择其他更为安全的交通工具。
(score: 0.9)

**Error Category: Perception Error, Lack of Knowledge**
**Idefics3 :**
**Answer:** 是的，滑板车可以方便地送孩子上学。您可以在孩子上学时使用滑板车，减少交通拥堵和堵车的时间。(score: 0.0)

**Reference Answer: 电动滑板车严禁搭载乘客，其设计和承重能力仅适合单人使用。擅自搭载他人，特别是带孩子一起乘坐，不仅违反使用规范，还可能带来严重的安全隐患。出于对自身和孩子安全的考虑，不建议使用滑板车送孩子上学。**

Figure 18: A sample error case of Information Extraction (subfield: Detailed Manual Understanding).Source:Web

**Information Extraction:**
**Search by Attribute without Calculate**

Task Name: Game Info Retrieval

Given a set of screenshots from a game information page and a specific query, retrieve the names of the games that match the query. If there are multiple matching games, provide a list of game names separated by commas.
**Question:** Find the game available on 'MacOS'

Keywords: **Object Recognition and Classification, OCR, Language Understanding and Generation, User Interface Screenshots**

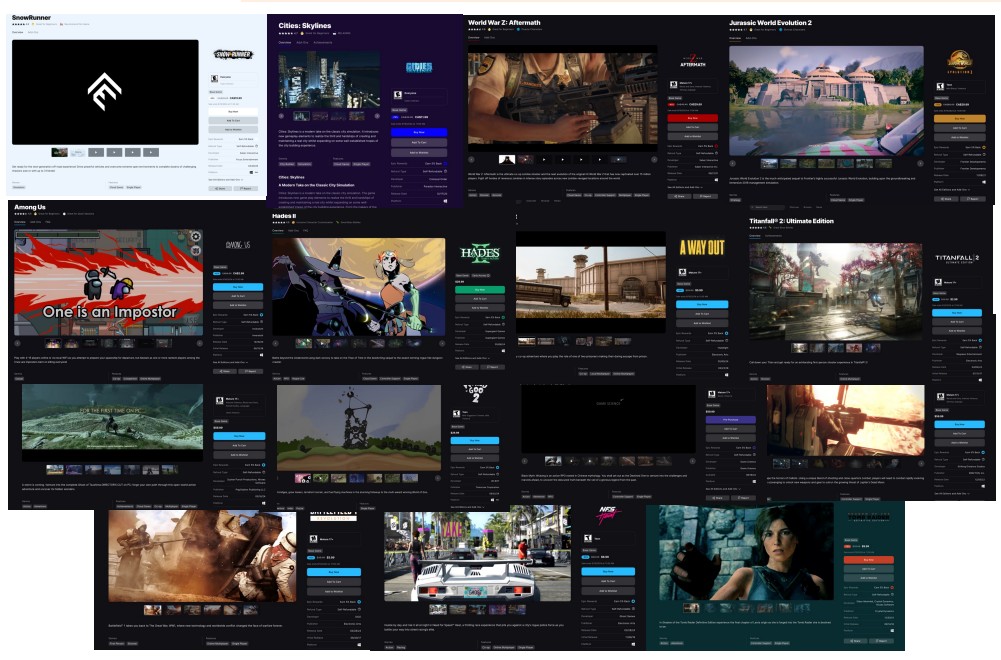

**Claude-3.5 :** Let's go through this step-by-step:
1. We need to look for a game that specifically mentions Mac or MacOS as a platform. …
**Answer: World of Goo 2, SnowRunner** (score: 1.0)

**Error Category: Perception Error**
**GPT-4o:** To find the game available on 'MacOS', I will examine each screenshot for the platform information.
…
6. World of Goo 2: No mention of MacOS.
…
**Answer: SnowRunner**(score: 0.0)

**Correct Answer: World of Goo 2, SnowRunner**

Figure 19: A sample error case of Information Extraction (subfield: Search by Attribute without Calculate).Source:Web

**Knowledge: World Knowledge**

Task Name: Insect Order Classification

This task involves the classification of insects into their respective taxonomic orders based on visual input. The goal is to accurately identify and categorize an insect from an image into one of the pre-defined orders. Each order is represented by a specific label corresponding to the insect's taxonomic classification, including orders such as Diptera, Hymenoptera, Coleoptera, Hemiptera, Lepidoptera, Psocodea, Thysanoptera, Trichoptera, Orthoptera, Blattodea, Neuroptera, Ephemeroptera, Dermaptera, Archaeognatha, Plecoptera, or Embioptera.

Keywords: **Object Recognition and Classification, Photographs, Domain-Specific Knowledge and Skills, Contextual Formatted**

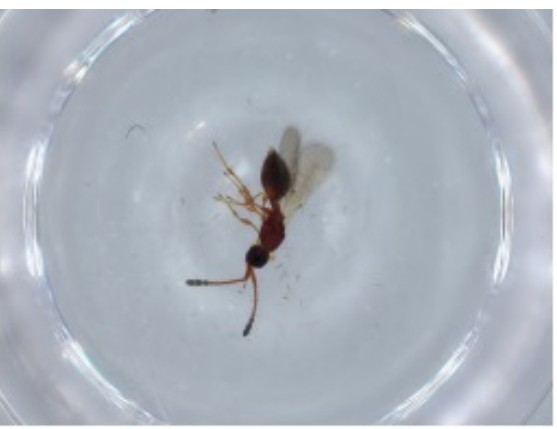

**Error Category: Lack of Knowledge**
**GPT-4o:** 1. The insect has a pair of wings. 2. The antennae are feathery, which is a characteristic feature of some insects. 3. The body structure is relatively small and delicate …
**Answer: Trichoptera** (score: 0.0)

**Error Category: Lack of Knowledge**
**Claude-3.5:** 1. The insect is small and appears to be in a circular container, likely for observation purposes. 2. The insect has an elongated body shape …
**Answer: Lepidoptera** (score: 0.0)

**Error Category: Lack of Knowledge**
**Gemini-1.5-pro:** The insect has long, slender antennae and its wings are held roof-like over its body. These are characteristic of the order Trichoptera, or caddisflies.
**Answer: Trichoptera** (score: 0.0)

**Correct Answer: Hymenoptera**

Figure 20: A sample error case of Knowledge (subfield: World Knowledge).
Source: BIOSCAN-1M (Gharaee et al., 2024)

**Knowledge: World Knowledge**

Task Name: Memorization Chinese Celebrity

Please answer the name and nationality of the celebrity. For the name, please put last name in the front.

Keywords: **Object Recognition and Classification, Photographs, Language Understanding and Generation, Structured**

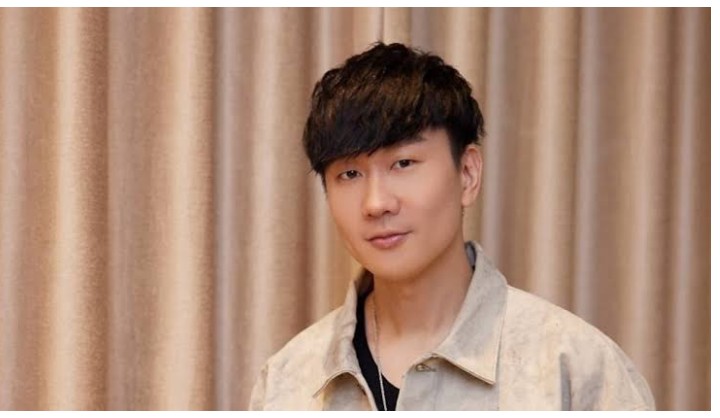

**Error Category: Lack of Knowledge**
**GPT-4o:** I don't know who they are.
**Answer: null**(score: 0.0)

**Error Category: Refuse to Answer**
**Claude-3.5:** I apologize, but I'm not able to identify or name specific individuals in images. However, I can describe what I see in this photo:…
**Answer: null**(score: 0.0)

**Error Category: Lack of Knowledge**
**Gemini-1.5-pro:** The person in the image is Eric Chou. He is a Taiwanese singer.
**Answer: {"name": "Chou Chih-chieh"; "nationality": "Taiwanese"}** (score: 0.0)

**Correct Answer: {"name": "Lin Junjie"; "nationality": "Singaporean"}**

Figure 21: A sample error case of Knowledge (subfield: World Knowledge). Source: Web

**Knowledge: Arts**

Task Name: Poetry Generation (Acrostic)

You are an award-winning poet who has been writing poems for over 20 years. Please write an acrostic poem about the attached image, using the letters in the specified word or words to start off the first letter of each line in the poem. A space between the words indicates a newline. If there are no spaces, there should not be any blank lines. No title is needed.
**Question:** Word: hamster

Keywords: **Object Recognition and Classification, Language Understanding and Generation, Photographs**

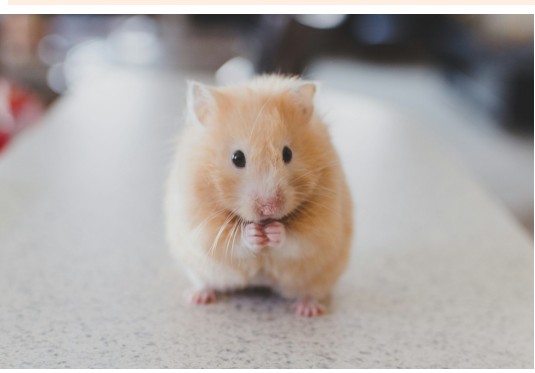

**GPT-4o: Answer:**
**Huddled in a cozy corner,**
**A tiny heart beats with gentle grace.**
**Mischief sparkles in those bright eyes,**
**Soft fur, a warm embrace.**
**Tiny paws clutching a hidden treasure,**
**Eagerly nibbling, lost in pleasure.**
**Radiating charm, a small wonder.** (score: 1.0)

Error Category: **Fail to Follow Instruction**
**Qwen2-VL-72B:** **Answer:**
**Humble hamster, with fur so fine,**
**A creature small, yet full of might.**
**Majestic in its tiny form,**
**Standing tall, without a care.**
**Eyes so bright, a curious gleam,**
**Ruling its world, in a tiny dream.** (score: 0.0) (acrostic: 0, contain: 1)

**Eval Context:** acrostic:  hamster,
**contain: [hamster, paw, tail, whisker, seed, rodent, scurry, fur]"**

Figure 22: A sample error case of Knowledge (subfield: Art). Source: Web
Back to List of Figures

**Mathematics: Graph Theory**

Task Name: Topological Sort

For a given Directed Acyclic Graph (DAG), your task is to output all possible topological orders. The output is a set of topological order.

Keywords: **Spatial and Temporal Reasoning, Structured, Mathematical and Logical Reasoning, Diagrams and Data Visualizations**

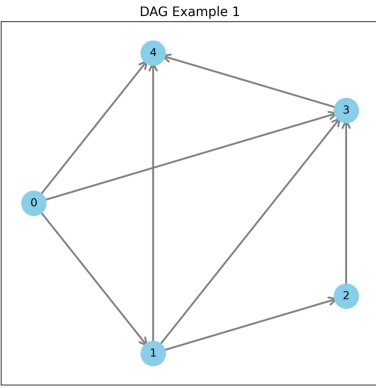

**Error Category:** Lack of Capability
**GPT-4o:** To determine which 3D object can be formed from the given 2D foldable sheet, we need to visualize how the sheet folds into a 3D shape. Let's analyze the 2D sheet and the 3D options step by step …
**Answer: [0->1->2->3->4, 0->1->3->4->2, 0->...]** (score: 0.0)

**Error Category:** Lack of Capability
**Claude-3.5:** 1. Identify the nodes with no incoming edges (sources):
  - Node 0 is the only source in this graph ….
**Answer: [0->1->2->3->4,0->1->2->4->3]** (score: 0.0)

**Error Category:** Lack of Capability
**Gemini-1.5-pro:** We can perform a depth-first search (DFS) from each node with no incoming edges. During the DFS, we maintain a stack to keep track of the visited nodes …
**Answer: [0->2->1->4->3, 0->2->1->3->4]** (score: 0.0)

**Correct Answer: [0->1->2->3->4]**

Figure 23: A sample error case of Mathematics (subfield: Graph Theory).
Source:Web

**Metrics: Generated Image Eval**

Task Name: Autorater Motion Guided Editing

Motion-guided image editing is important for generating realistic and context-aware animation. Your task is to rank the motion-guided image generation results. The desired motion is described by a text input, several generative models try to edit the input image based on the described motion. You should rate the generation results based on 1) how well the generated image follows the motion described by the text query and 2) whether the generated image is still consistent with the source image in terms of semantics and object attributes. The first image is the source image; the rest are the edited results. Rank the results using their indices, starting with 1. Exclude the source image from the ranking.
**Question:** Move the lion to the left

Keywords:
**Object Recognition, Photographs, Spatial and Temporal Reasoning, Structured**

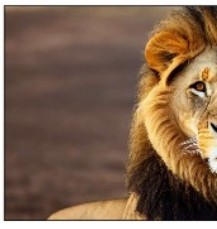 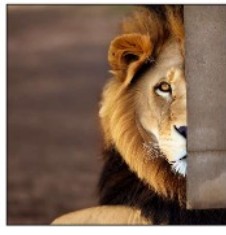 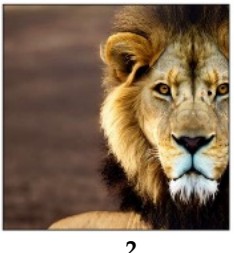 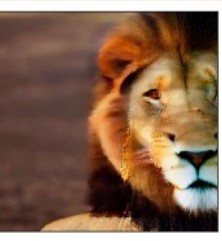

| Input image | 1 | 2 | 3 |

**Claude-3.5:** 1. Image 2: Best follows the instruction to move left while maintaining the lion's features and overall composition. 2. Image 3: Maintains consistency with the source image but shows minimal leftward movement. 3. Image 1: Follows the leftward motion but obscures too much …
**Answer: 2, 3, 1** (score: 1.0)

**Error Category: Fail to Follow Instruction**
**Idefics3:**
**Answer: no.** (score: 0.0)

**Error Category: Fail to Follow Instruction**
**internvl-76B:** The first image shows the lion facing forward, with its head straight. The second image shows the lion tilting its head to the left … Based on these observations, we can conclude that the second image is the one that shows the lion moving to the left.
**Answer: 2** (score: 0.0)

**Correct Answer: 2,3,1**

Figure 24: A sample error case of Metrics (subfield: Generated Image Eval).
Source:Motion Guidance (Geng & Owens, 2024)
Back to List of Figures

**Metrics: Generated Image Eval**

Task Name: Autorater 3d Model Texturing

You will be asked to evaluate the quality of textured 3D models by comparing two texture transfer results. The first image will show an untextured 3D model and a source image whose texture needs to be transferred to the mesh. The second image will show two textured 3D models for comparison, and you should pick one with better texturing quality (left or right).

**Keywords:** **Object Recognition, 3D Models and Aerial Imagery, Scene and Event Understanding, Contextual Formatted Text**

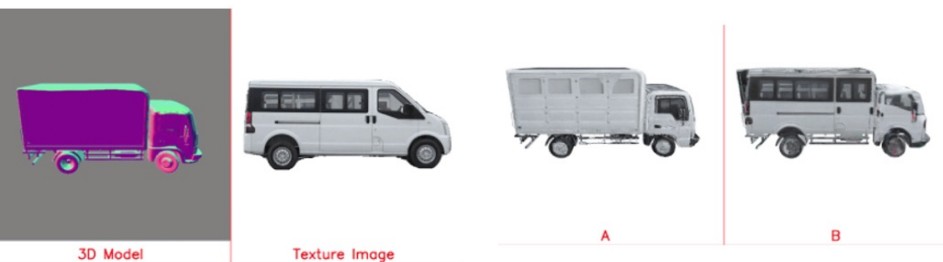

**Claude-3.5:** Model A doesn't match the source image well. While it's a realistic vehicle, it's a different type (cargo truck instead of passenger van) …
**Answer: right** (score: 1.0)

**Error Category: Perception Error**
**GPT-4o:** To determine which textured 3D model has better texturing quality, we need to consider the realisticness of the textured object and the consistency between the textured mesh and the source image. …
**Answer: left** (score: 0.0)

**Error Category: Perception Error**
**Gemini-1.5-pro:** The left image (A) is a white truck, while the right image (B) is a combination of a truck and a van. The source image is a van. Therefore, the left image is more realistic and consistent with the source image. …
**Answer: left** (score: 0.0)

**Correct Answer: right**

Figure 25: A sample error case of Metrics (subfield: Generated Image Eval).
Source: EASI-Tex (Perla et al., 2024)
Back to List of Figures

**Planning: Puzzles and Games**

Task Name: Pictionary GenAI Output Chinese

Look at the provided image and question, guess which Chinese word or phrase the image is trying to convey. Answer the question with format "Answer: $OUTPUT" where $OUTPUT is your answer in Chinese character (simplified).
**Question:**图中表示了一个六字典故，请问是什么？

Keywords: **Object Recognition and Classification, Artistic and Creative Content, Language Understanding and Generation, Contextual Formatted**

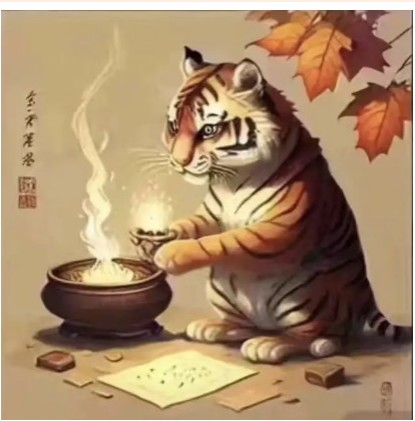

**Error Category: Lack of Capability**
**GPT-4o:** The image shows a tiger making tea, which is a visual representation of the Chinese idiom "老虎不吃人" (a tiger that doesn't eat people). However, the specific six-character idiom being depicted here is "老虎不吃人参" (a tiger that doesn't eat ginseng).
**Answer: 老虎不吃人参** (score: 0.0)

**Error Category: Lack of Capability**
**Claude-3.5:** 1. We see a tiger in the image, which is the central figure
...
5. The tiger seems to be focused on or studying the burning incense.
**Answer: 卧薪尝胆** (score: 0.0)

**Correct Answer: 唐伯虎点秋香**

Figure 26: A sample error case of Planning: (subfield: Puzzles and Games).
Source: Web

## Planning: Puzzles and Games

### Task Name: Logical Reasoning 2d Folding

Given the query image including a 2D foldable sheet (the first image) and several options for the folded 3D objects (the remaining images), please find out which 3D object comes from the 2D sheet. The answer should the image index starting from 1 that represents the second image.

**Keywords:** Spatial and Temporal Reasoning, Multiple Choice, Mathematical and Logical Reasoning, Diagrams and Data Visualizations

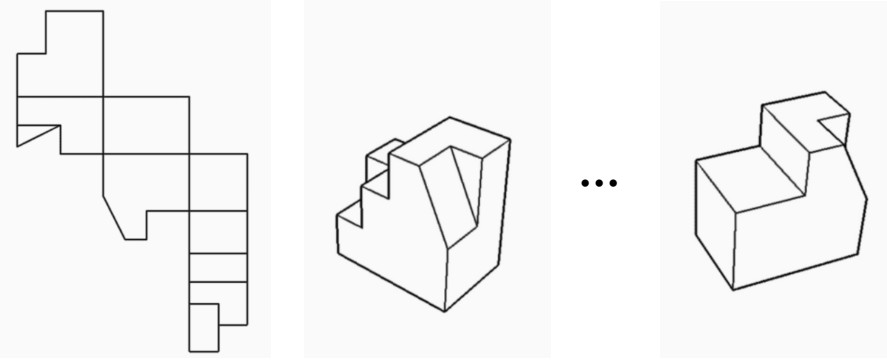

**Error Category: Lack of Capability**
**GPT-4o:** To determine which 3D object can be formed from the given 2D foldable sheet, we need to visualize how the sheet folds into a 3D shape. Let's analyze the 2D sheet and the 3D options step by step …
**Answer: 1** (score: 0.0)

**Error Category: Lack of Capability**
**Claude-3.5:** 1. First, we need to analyze the 2D foldable sheet (the first image). It has a unique shape with several rectangular sections and one triangular section. 2. Now, let's look at each of the 3D objects (images 1-5) and try to match them with the 2D sheet …
**Answer: 1** (score: 0.0)

**Error Category: Lack of Capability**
**Gemini-1.5-pro:** The 2D sheet has a long base with a small square attached to the right. There are two protrusions from the base, one with a triangular cutout and one rectangular. Only option 1 satisfies these conditions.
**Answer: 1** (score: 0.0)

**Correct Answer: 3**

Figure 27: A sample error case of Planning (subfield: Puzzles and Games).
Source:Web

## Planning: Reordering

### Task Name: Perception Test Video Character Order

The task involves identifying the order of letters as a person writes, types, or arranges them, predicting subsequent letters, and reversing letter sequences in various scenarios. What was the order of the letters at the beginning?

Keywords: **Object Recognition, Video, Spatial and Temporal Reasoning, Contextual Formatted Text**

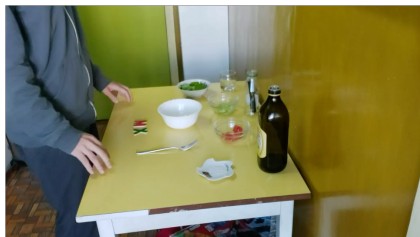 ... 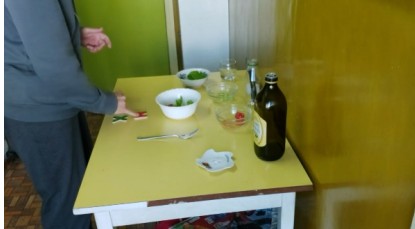

**GPT-4o:** The order of the letters at the beginning was M X.
**Answer: mx** (score: 1.0)

**Error Category: Fail to Follow Instruction**
**Idefics3:** The order of the letters at the beginning was not mentioned in the facts provided.
**Answer: null** (score: 0.0)

**Error Category: Perspection Error**
**Gemini-1.5-pro:** The letters at the beginning of the video spell out MIX. The person first places the M, then the I, and finally the X.
**Answer: mix** (score: 0.0)

**Correct Answer: mx**

Figure 28: A sample error case of Planning (subfield: Reordering).
Source:Perception Test (Patraucean et al., 2024)
Back to List of Figures

**Perception: Multimodal Constrained Captioning**

Task Name: Constrained Generation Contain Length

Can you write a story about the animal in the figure with more than 10 but less than 20 words?

Keywords: **Object Recognition and Classification, Open-Ended, Language Understanding and Generation, Photographs**

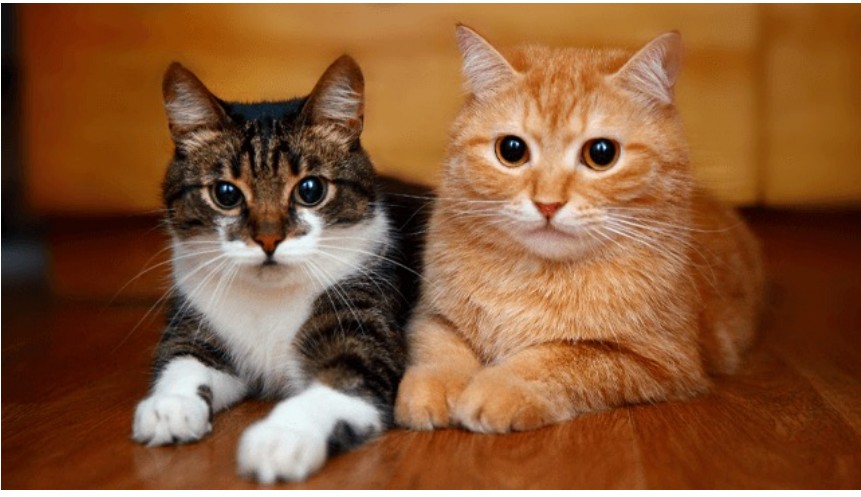

**GPT-4o:** Two curious cats, one tabby and one ginger, explored their cozy home, discovering hidden treasures together. (score: 1.0)

**Error Category: Fail to Follow Instruction**
**Claude-3.5:** Two feline friends, a tabby and a ginger, lounged side by side, their curious eyes gleaming with mischief and companionship. (score: 0.0) (contain: 0, length1: 1, length2: 0)

**Eval Context:** contain: [cat, kitten], length1: >10, length2: <20

Figure 29: A sample error case of Perception (subfield: Multimodal Constrained Captioning). Source: Web

Back to List of Figures

## Perception: Visual Recognition

### Task Name: License Plate Recognition

Please extract the license plate name from the image, ignore the punctuations.

Keywords: **Text Recognition (OCR), Contextual Formatted Text, Object Recognition and Classification, Photographs**

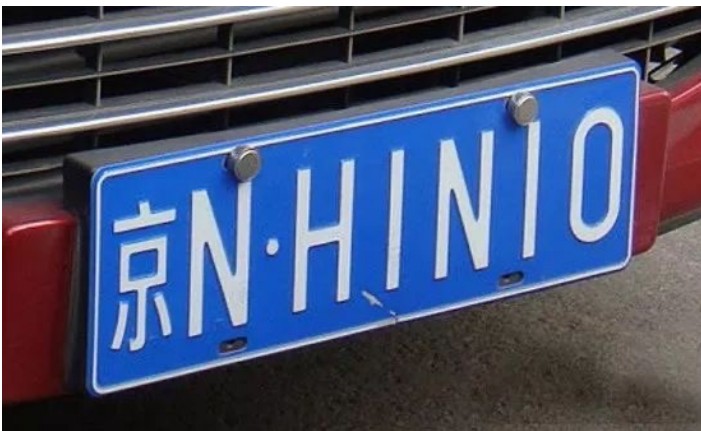

**GPT-4o:** 京N·HINIO
**Answer: 京NHINIO** (score: 1.0)

**Error Category: Lack of Knowledge**
**Claude-3.5:** Let's approach this step-by-step: 1. The image shows a blue license plate, similar to the example provided … The task asks us to ignore punctuation, so we should remove the dot (•). 6. Combining all elements and ignoring punctuation, we get: 京NHIN10.
**Answer: 京NHIN10** (score: 0.0)

**Error Category: Lack of Knowledge**
**Gemini-1.5-pro:** The license plate in the image contains the characters "京N" followed by "HINI0".
**Answer: 京NHINI0** (score: 0.0)

**Correct Answer: 京NHINIO**

Figure 30: A sample error case of Perception (subfield: Visual Recognition).
Source:Web
Back to List of Figures

## Perception: Visual Recognition

### Task Name: Coco Person Detection

You are asked to detect persons in a query image. The format of bounding box is (x1,y1,x2,y2), where (x1,y1) is the top-left corner and (x2,y2) is the bottom-right corner. The coordinates are normalized by the image's height and width.

**Keywords:**      **Object Recognition and Classification, Photographs, Numerical Data**

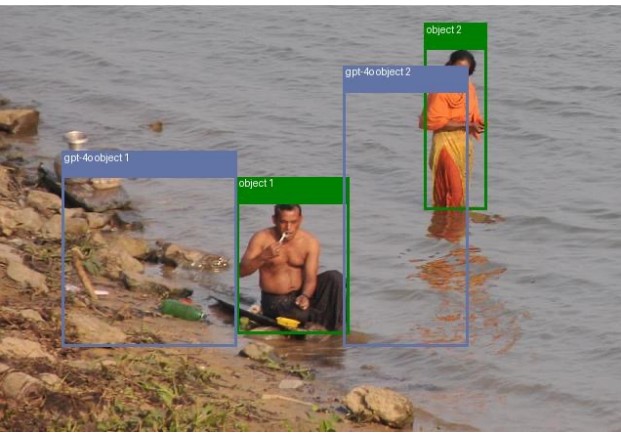

**Error Category: Perception Error**
**GPT-4o:** 1. Identify the persons in the image. 2. Determine the bounding boxes for each person. 3. Normalize the coordinates by the image's height and width …
**Answer: {"number of person": "2";**
**"bounding boxes": "[(0.05,0.40,0.30,0.80), (0.60,0.20,0.85,0.80)]"}** (score: 0.54)

**Error Category: Perception Error**
**Claude-3.5:** 1. I can see two people in the image, both partially submerged in water. 2. The first person is on the left side of the image, sitting in shallow water near the shore. …
**Answer: {"number of person": "2";**
**"bounding boxes": "[(0.20,0.40,0.55,0.95), (0.70,0.15,0.90,0.85)]"}** (score: 0.64)

**Error Category: Perception Error**
**Gemini-1.5-pro:** There are two people in the image. The man is on the left and the woman is on the right …
**Answer: {"number of person": "2";**
**"bounding boxes": "[(0.42,0.54,0.74,0.84), (0.68,0.52,0.80,0.82)]"}** (score: 0.62)

**Correct Answer: {"number of person": "2";**
**"bounding boxes": "[(0.38,0.46,0.56,0.77), (0.68,0.10,0.78,0.48)]"}**

Figure 31: A sample error case of Perception (subfield: Visual Recognition).
Source: COCO (Lin et al., 2014)
Back to List of Figures

**Perception: Visual Recognition**

Task Name: Face Identity Matching

This task involves analyzing a set of 5 images to identify which images show the same person. Among the 5 images, there are two pairs of images where each pair shows the same individual, and one image shows a different person. The goal is to determine the matching pairs by providing the indices of the images that depict the same person. The answer should list the smaller indices first within each pair.

Keywords:    **Scene and Event Understanding, Object Recognition, Photographs, Contextual Formatted**

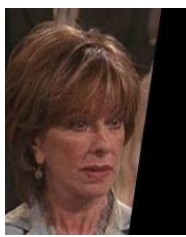 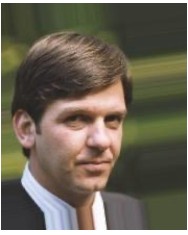 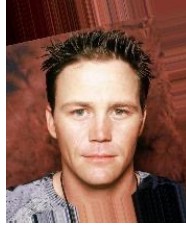 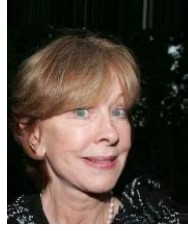 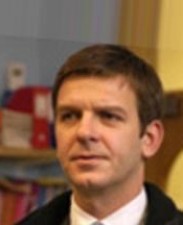

**Claude-3.5 :** 1. The first image shows an older woman with short, brown hair. 2. The second image shows a man with short, dark hair. 3. The third image shows a man with short, light brown hair. 4. The fourth image shows an older woman with short, light brown hair …
**Answer: 14,25** (score: 1.0)

**Error Category: Fail to Follow Instruction**
**Idefics3:** Example Response: [PLEASE OUTPUT YOUR REASONING]
**Answer: 1, 3** (score: 0.0)

**Correct Answer: 14,25**

Figure 32: A sample error case of Perception (subfield: Visual Recognition).
Source: CelebA (Liu et al., 2015)
Back to List of Figures

**Science: STEM**

Task Name: Scibench Fundamental wo Solution

Solve the problem. The answer is a number. A proton is a distance d/2 directly above the center of a square of side d. What is the magnitude of the electric flux through the square? (Hint: Think of the square as one face of a cube with edge d).

Keywords:  **Mathematical and Logical Reasoning, Numerical Data, Domain-Specific Knowledge and Skills, Diagrams and Data Visualizations**

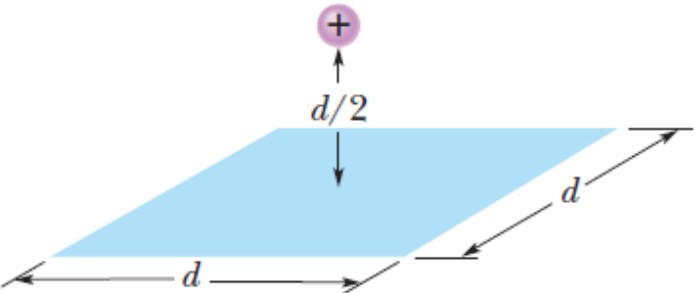

**Error Category: Lack of Capability**
**GPT-4o:** To solve the problem, we need to determine the electric flux through the square due to the proton positioned at a distance $\( d/2 \)$ directly above the center of the square …
**Answer: 3.01 \times 10^{-21}** (score: 0.0)

**Error Category: Fail to Follow Instruction**
**Claude-3.5:** First, recall Gauss's law for electric fields. It states that the total electric flux through a closed surface is equal to the enclosed charge divided by $\varepsilon_0$ (the permittivity of free space) …
**Answer: |q| / (6$\varepsilon_0$)** (score: 0.0)

**Error Category: Fail to Follow Instruction**
**Gemini-1.5-pro:** The electric flux through a closed surface is given by Gauss's law: $\Phi_E = \frac{q_{enc}}{\epsilon_0}$…
**Answer: $q/(6\epsilon_0)$** (score: 0.0)

**Correct Answer: 3.01**

Figure 33: A sample error case of Science (subfield: STEM).
Source: SciBench (Wang et al., 2023b)
Back to List of Figures

# G   DETAILED TASK INFORMATION

In Table 18, we list data source details for every task in our benchmark. We also list the output format and metrics to help better understand each task's form.

Table 18: Detailed task information description of MEGA-BENCH.

| Task Name | Source Description | Output Format | Metrics |
|---|---|---|---|
| **Information Extraction** | | | |
| Cheapest Flight Identification | Screenshots were taken by the human annotator on Google Flights. Questions and answers were created by the annotator. | Contextual | Simple String Match |
| Weather Info Retrieval | Screenshots were taken by the human annotator on Microsoft Weather. Questions and answers were created by the annotator. | Contextual | String Set Equality Comma |
| Stock Info Retrieval | Screenshots were taken by the human annotator on Yahoo Finance. Questions and answers were created by the annotator. | Contextual | Set Equality |
| Game Platform Support Identification | Screenshots were taken by the human annotator on the Steam store. Questions and answers were created by the annotator. | Structured | Exact String Match, Set Equality |
| Top Rated Hotel Identification | Screenshots were taken by the human annotator on Booking.com. Questions and answers were created by the annotator. | Contextual | String Set Equality Comma |
| Movie Info Retrieval | Screenshots were taken by the human annotator on the Amazon Prime Video webpage. Questions and answers were created by the annotator. | Contextual | String Set Equality Comma |
| Top Video Creator Identification | Screenshots were taken by the human annotator on YouTube. Questions and answers were created by the annotator. | Exact | Exact String Match |
| Highest Discount Game Price Identification | Screenshots were taken by the human annotator on the Steam store. Questions and answers were created by the annotator. | Numerical | Exact String Match |
| Newspaper Page Parse And Count | Data collected from the Newspaper Navigation Dataset (Lee et al., 2020). Questions and answers were created by the annotator. | Exact | Exact String Match |
| Remaining Playback Time Calculation | Screenshots were taken by the human annotator on YouTube. Questions and answers were created by the annotator. | Exact | Exact String Match |

Table 18 – continued from previous page

| Task Name | Source Description | Output Format | Metrics |
|---|---|---|---|
| Multi Lingual Manual Explanation Scooter Spanish | Screenshots taken from user manual located at https://fcc.report/FCC-ID/2A33E5LCHG11U/6288539.pdf. Questions and answers created by human annnotator. | Open | GPT-4o as Judge |
| Multi Lingual Manual Explanation Scooter Arabic | Screenshots taken from user manual located at https://fcc.report/FCC-ID/2A33E5LCHG11U/6288539.pdf. Questions and answers created by human annnotator. | Open | GPT-4o as Judge |
| Multi Lingual Manual Explanation Scooter French | Screenshots taken from user manual located at https://fcc.report/FCC-ID/2A33E5LCHG11U/6288539.pdf. Questions and answers created by human annnotator. | Open | GPT-4o as Judge |
| Multi Lingual Manual Explanation Scooter Chinese | Screenshots taken from user manual located at https://fcc.report/FCC-ID/2A33E5LCHG11U/6288539.pdf. Questions and answers created by human annnotator. | Open | GPT-4o as Judge |
| Multi Lingual Manual Explanation Scooter Russian | Screenshots taken from user manual located at https://fcc.report/FCC-ID/2A33E5LCHG11U/6288539.pdf. Questions and answers created by human annnotator. | Open | GPT-4o as Judge |
| Video Summary | Videos taken from WikiHow or YouTube. Questions and answers created by human annotator. | Open | GPT-4o as Judge |
| Video Short Title | Videos taken from YouTube. Questions and answers created by human annnotator. | Open | GPT-4o as Judge |
| Video2notes | WikiHow or YouTube. Questions and answers created by human annnotator. | Open | GPT-4o as Judge |
| Video Content Reasoning | Videos and annotations were taken from the HME100k (Yuan et al., 2022) dataset. Questions and answers were adapted by a human annotator. | Contextual | Simple String Match |
| COCO OOD Global Image Retrieval By Query Property | Images were from COCO-O (Mao et al., 2023). Questions and answers were re-designed by the annotator manually | Structured | Jaccard Index |

Table 18 – continued from previous page

| Task Name | Source Description | Output Format | Metrics |
|-----------|-------------------|---------------|---------|
| Places365 Similar Scene Retrieval | Images and labels were taken from the Places365 dataset (Zhou et al., 2017) and adapted into questions and answers by a human annotator. | MC | Exact String Match |
| Booking Web Recommendation | Images and labels come from the SEED-Bench (Li et al., 2024b) dataset. Some images are from Yelp. Questions and annotations were adapted by a human annotator. | Contextual | Jaccard Index Case Insensitive |
| Game Info Retrieval | Screenshots were taken by the human annotator on the Epic Games Store. Questions and answers were created by the annotator. | Contextual | String Set Equality Comma |
| Media Homepage Profile | Most images and labels come from the SEED-Bench (Li et al., 2024b) dataset, while one came from a screenshot taken by a human annotator. Questions and annotations were adapted by a human annotator. | Structured | Jaccard Index Case Insensitive |
| Movie Retrieval By Actor | Screenshots were taken by the human annotator on the Amazon Prime Video webpage. Questions and answers were created by the annotator. | Contextual | String Set Equality Comma |
| Music Info Retrieval | Screenshots were taken by the human annotator on the Spotify Web Player. Questions and answers were created by the annotator. | Contextual | String Set Equality Comma |
| Tv Show Retrieval By Character | Screenshots were taken by the human annotator on the Amazon Prime Video webpage. Questions and answers were created by the annotator. | Contextual | String Set Equality Comma |
| App Layout Understanding Leetcode | Screenshots were taken by the human annotator on Leetcode. Questions and answers were created by the annotator. | Exact | Exact String Match |
| App Layout Understanding Youtube | Screenshots were taken by the human annotator on YouTube. Questions and answers were created by the annotator. | Exact | Exact String Match |
| App Layout Understanding Amazon | Screenshots were taken by the human annotator on Amazon. Questions and answers were created by the annotator. | Exact | Exact String Match |

Table 18 – continued from previous page

| Task Name | Source Description | Output Format | Metrics |
|---|---|---|---|
| App Layout Understanding Word | Screenshots were taken by the human annotator on Microsoft Word. Questions and answers were created by the annotator. | Exact | Exact String Match |
| App Layout Understanding Notes | Screenshots were taken by the human annotator on the Google Notes app. Questions and answers were created by the annotator. | Exact | Exact Str Match Case Insensitive |
| App Layout Understanding Ppt | Screenshots were taken by the human annotator on Microsoft PowerPoint. Questions and answers were created by the annotator. | Exact | Exact String Match |
| App Layout Understanding Alipay | Screenshots were taken by the human annotator on the Alipay app. Questions and answers were created by the annotator. | Exact | Exact String Match |
| App Layout Understanding Instagram | Screenshots were taken by the human annotator on the Instagram app. Questions and answers were created by the annotator. | Exact | Exact String Match |
| App Layout Understanding Zoom | Screenshots were taken by the human annotator on Zoom. Questions and answers were created by the annotator. | Exact | Exact String Match |
| App Layout Understanding Excel | Screenshots were taken by the human annotator on Microsoft Excel. Questions and answers were created by the annotator. | Exact | Exact String Match |
| App Layout Understanding Iphone Settings | Screenshots were taken by the human annotator on the iPhone. Questions and answers were created by the annotator. | Exact | Exact String Match |
| App Layout Understanding Tiktok | Screenshots were taken by the human annotator on the TikTok app. Questions and answers were created by the annotator. | Exact | Exact String Match |
| App Layout Understanding Twitter | Screenshots were taken by the human annotator on the X (formerly Twitter) app. Questions and answers were created by the annotator. | Exact | Exact String Match |
| Multilingual News Qa | Screenshots were taken by the human annotator on X (formerly Twitter). Questions and answers were created by the annotator. | Contextual | Multi Ref Phrase |

Table 18 – continued from previous page

| Task Name | Source Description | Output Format | Metrics |
|---|---|---|---|
| Product Ocr Qa | Images were taken from various websites. Questions and answers were created by the annotator. | Exact | Exact String Match |
| Research Website Parsing Blogpost | Screenshots were taken of various ML research websites. Questions and answers were created by the annotator. | Contextual | Multi Ref Phrase |
| Research Website Parsing Homepage | Screenshots were taken of various ML research websites. Questions and answers were created by the annotator. | Contextual | Multi Ref Phrase |
| Research Website Parsing Publication | Screenshots were taken of various ML research websites. Questions and answers were created by the annotator. | Contextual | Multi Ref Phrase |
| Gui Chat Easy | Images and annotations were adapted from the GUI Chat dataset (Chen et al., 2024c) by the human annotator into an open-ended question. | Open | GPT-4o as Judge |
| Gui Chat Hard | Images and annotations were adapted from the GUI Chat dataset (Chen et al., 2024c) by the human annotator into an open-ended question. | Open | GPT-4o as Judge |
| Realworld Qa En2cn | Images and annotations were adapted from the RealWorldQA benchmark (xAI, 2024) by the human annotator into an open-ended question. The translation requirement was added by the human annotator. | Contextual | Multi Ref Phrase |
| Star Object Interaction Video | Videos and annotations were adapted from the STAR benchmark (Wu et al., 2024) by the human annotator into questions and answers. | Contextual | Multi Ref Phrase |
| Funqa Unexpected Action Magic Video | Videos and annotations were adapted from the FunQA benchmark (Xie et al., 2023) by the human annotator into being an open-ended question. | Open | GPT-4o as Judge |
| Activitynetqa | Images and annotations were adapted from the ActivityNetQA benchmark (Yu et al., 2019) by the human annotator into being an open-ended question. | Open | GPT-4o as Judge |

Table 18 – continued from previous page

| Task Name | Source Description | Output Format | Metrics |
|---|---|---|---|
| Funqa Unexpected Action Creative Video | Videos and annotations were adapted from the FunQA benchmark (Xie et al., 2023) by the human annotator into being an open-ended question. | Open | GPT-4o as Judge |
| Nextqa Mc | Images and annotations were adapted from the NExTQA benchmark (Xiao et al., 2021) by the human annotator into questions and answers. | MC | Exact String Match |
| Video Qa | Videos taken from YouTube. Questions and answers created by human annnotator. | Open | GPT-4o as Judge |
| Nextqa Oe | Images and annotations were adapted from the NExTQA benchmark (Xiao et al., 2021) by the human annotator into being an open-ended question. | Open | GPT-4o as Judge |
| Funqa Unexpected Action Humor Video | Videos and annotations were adapted from the FunQA benchmark (Xie et al., 2023) by the human annotator into being an open-ended question. | Open | GPT-4o as Judge |
| Multilingual Movie Info Parsing | Screenshots were taken by the human annotator on the Amazon Prime Video webpage. Questions and answers were created by the annotator. | Structured | Exact String Match, Simple String Match |
| Movie Info Parsing | Screenshots were taken by the human annotator on the Amazon Prime Video webpage. Questions and answers were created by the annotator. | Structured | Exact String Match |
| Stock Info Parsing | Screenshots were taken by the human annotator on Yahoo Finance. Questions and answers were created by the annotator. | Structured | Exact String Match |
| Music Info Parsing | Screenshots were taken by the human annotator on the Spotify Web Player. Questions and answers were created by the annotator. | Structured | Exact String Match |
| Multilingual Game Info Parsing | Screenshots were taken by the human annotator on the Epic Games Store. Questions and answers were created by the annotator. | Structured | Exact String Match |
| Ocr Article Authors | Screenshots taken of various academic papers. Questions and answers created by human annotator. | Structured | Simple String Match |

Table 18 – continued from previous page

| Task Name | Source Description | Output Format | Metrics |
|---|---|---|---|
| Youtube Video Info Parsing | Videos taken from YouTube. Questions and answers created by human annnotator. | Structured | Exact String Match |
| Tv Show Info Parsing | Screenshots were taken by the human annotator on the Amazon Prime Video webpage. Questions and answers were created by the annotator. | Structured | Simple String Match |
| Ocr Resume School Plain | Resumes taken from various personal websites. Questions and answers were created by the annotator. | Contextual | String Set Equality Line Break |
| Image Translation En2cn | Images were collected from various sources, including academic papers, news articles, shopping receipts, etc. The annotations are obtained by GPT-4o translation followed by a human check. | Contextual | Gleu Cn |
| Booking Web Rating | Images and labels come from the SEED-Bench (Li et al., 2024b) dataset. Some images are from Yelp. Questions and annotations were adapted by a human annotator. | Structured | Exact String Match |
| Weather Info Parsing | Images were collected from the Microsoft Weather by taking screenshots. Questions and answers were designed by the annotator. | Structured | Exact String Match |
| Game Info Parsing | Screenshots were taken by the human annotator on the Epic Games Store. Questions and answers were created by the annotator. | Structured | Exact String Match |
| Weather Map Climate Type Temperature Parsing | One of the examples comes from the SEED-Bench 2 Plus benchmark (Li et al., 2024b). The rest of the images were collected from various online websites. Questions and annotations were adapted by a human annotator. | Structured | Exact String Match |
| Hotel Booking Confirmation Parsing | Screenshots were taken by the human annotator on Booking.com. Questions and answers were created by the annotator. | Structured | Exact String Match |

Table 18 – continued from previous page

| Task Name | Source Description | Output Format | Metrics |
|-----------|-------------------|---------------|---------|
| Entertainment Web Game Style | Some of the examples come from the SEED-Bench 2 Plus benchmark (Li et al., 2024b). The rest of the screenshots were taken on the Steam store. Questions and annotations were adapted by a human annotator. | Structured | Exact Str Match Case Insensitive, Exact String Match |
| **Planning** | | | |
| Wikihow Complex Task Completion | Data collected from website, and the questions and answers are designed by human annotator | Open | GPT-4o as Judge |
| Vln Identify Robot | Data collected from RxR dataset (Ku et al., 2020), the question and answer are adapted to select the robot that should execute the instruction | Exact | Exact String Match |
| Vln English Next Step | Data collected from RxR dataset (Ku et al., 2020), the question and answer are adapted by human annotator | Contextual | Simple String Match |
| Vlnqa Egocentric Navigation Video | Data collected from VLN-CE (Krantz et al., 2020) and the task is adapted from MVBench (Li et al., 2024e), the question and answer are adapted by human annotator | Contextual | Simple String Match |
| Vln Identify Location | Data collected from RxR dataset (Ku et al., 2020), the question and answer are adapted by human annotator | Structured | Exact String Match |
| Vln Tegulu Next Step | Data collected from RxR dataset (Ku et al., 2020), the question and answer are adapted by human annotator | Structured | Simple String Match |
| Vln Hindi Next Step | Data collected from RxR dataset (Ku et al., 2020), the question and answer are adapted by human annotator | Contextual | Simple String Match |
| App Interactive Operations Instagram | Data collected from application screenshots by human annotator, and the questions and answers are designed by human annotator | MC | Exact String Match |
| App Interactive Operations Leetcode | Data collected from application screenshots by human annotator, and the questions and answers are designed by human annotator | MC | Exact String Match |

Table 18 – continued from previous page

| Task Name | Source Description | Output Format | Metrics |
|---|---|---|---|
| Gui Act Web Multi | Data collected from webpage screenshots by human annotator, and the questions and answers bounding boxes are annotated by human annotator | Structured | Exact String Match, Xml Nbbox Iou Single |
| App Interactive Operations Ppt | Data collected from application screenshots by human annotator, and the questions and answers are designed by human annotator | MC | Exact String Match |
| Gui Act Mobile Swipe | Data collected from application screenshots by human annotator, and the questions and answers bounding boxes are annotated by human annotator | Structured | Xml Norm Point Distance |
| App Interactive Operations Excel | Data collected from application screenshots by human annotator, and the questions and answers are designed by human annotator | MC | Exact String Match |
| Gui Act Mobile Tap | Data collected from application screenshots by human annotator, and the questions and answers bounding boxes are annotated by human annotator | Numerical | Xml Norm Point In Bbox |
| App Interactive Operations Alipay | Data collected from application screenshots by human annotator, and the questions and answers are designed by human annotator | MC | Exact String Match |
| Gui Act Web Single | Data collected from webpage screenshots by human annotator, and the questions and answers bounding boxes are annotated by human annotator | Numerical | Xml Nbbox Iou Single |
| App Interactive Operations Twitter | Data collected from application screenshots by human annotator, and the questions and answers are designed by human annotator | MC | Exact String Match |
| App Interactive Operations Word | Data collected from application screenshots by human annotator, and the questions and answers are designed by human annotator | MC | Exact String Match |
| App Interactive Operations Iphone Settings | Data collected from application screenshots by human annotator, and the questions and answers are designed by human annotator | MC | Exact String Match |
| App Interactive Operations Tiktok | Data collected from application screenshots by human annotator, and the questions and answers are designed by human annotator | MC | Exact String Match |

Table 18 – continued from previous page

| Task Name | Source Description | Output Format | Metrics |
|---|---|---|---|
| App Interactive Operations Notes | Data collected from application screenshots by human annotator, and the questions and answers are designed by human annotator | MC | Exact String Match |
| App Interactive Operations Zoom | Data collected from application screenshots by human annotator, and the questions and answers are designed by human annotator | MC | Exact String Match |
| App Interactive Operations Amazon | Data collected from application screenshots by human annotator, and the questions and answers are designed by human annotator | MC | Exact String Match |
| Web Action Grounding | Data collected from Visual-WebBench (Liu et al., 2024a), and the questions and answers are adapted by human annotator | MC | Exact String Match |
| App Interactive Operations Youtube | Data collected from application screenshots by human annotator, and the questions and answers are designed by human annotator | MC | Exact String Match |
| Calendar Schedule Suggestion | Data collected from Google Calendar by human annotator, and the questions and answers are designed by human annotator to identify all possible starting times for a meeting within a specified time range and duration | Contextual | Set Equality |
| Planning Visual Barman | Data collected from Planning Domain Definition Language (PDDL) of Barman, and the questions and answers are adapted to match the transitions from init state to goal state | Structured | Symbolic Planning Test |
| Planning Visual Floortile | Data collected from website, and the questions and answers are adapted to match the transitions from init state to goal state | Structured | Symbolic Planning Test |
| Planning Visual Storage | Data collected from website, and the questions and answers are adapted to match the transitions from init state to goal state | Structured | Symbolic Planning Test |
| Planning Screenshot Grippers | Data collected from website, and the questions and answers are adapted to match the transitions from init state to goal state | Structured | Symbolic Planning Test |
| Planning Visual Blocksworld | Data collected from website, and the questions and answers are adapted to match the transitions from init state to goal state | Structured | Symbolic Planning Test |

Table 18 – continued from previous page

| Task Name | Source Description | Output Format | Metrics |
|---|---|---|---|
| Planning Screen-shot Barman | Data collected from Planning Domain Definition Language (PDDL) of Barman, and the questions and answers are adapted to match the transitions from init state to goal state | Structured | Symbolic Planning Test |
| Planning Screen-shot Termes | Data collected from website, and the questions and answers are adapted to match the transitions from init state to goal state | Structured | Symbolic Planning Test |
| Planning Screen-shot Floortile | Data collected from website, and the questions and answers are adapted to match the transitions from init state to goal state | Structured | Symbolic Planning Test |
| Planning Screen-shot Blocksworld | Data collected from website, and the questions and answers are adapted to match the transitions from init state to goal state | Structured | Symbolic Planning Test |
| Planning Screen-shot Storage | Data collected from website, and the questions and answers are adapted to match the transitions from init state to goal state | Structured | Symbolic Planning Test |
| Planning Visual Termes | Data collected from website, and the questions and answers are adapted to match the transitions from init state to goal state | Structured | Symbolic Planning Test |
| Planning Screen-shot Tyreworld | Data collected from website, and the questions and answers are adapted to match the transitions from init state to goal state | Structured | Symbolic Planning Test |
| Planning Visual Grippers | Data collected from website, and the questions and answers are adapted to match the transitions from init state to goal state | Structured | Symbolic Planning Test |
| Logical Reasoning Find Odd One Out | Data collected from website, and the questions and answers are adapted to match strings | Structured | Dict Equality, Exact String Match |
| Logical Reasoning Fit Pattern | Data collected from LogicVista (Xiao et al., 2024), and the questions and answers are adapted by human annotator | MC | Exact String Match |
| Perception-Test Object Shuffle Video | Data collected from VLN-CE (Krantz et al., 2020) and the task is adapted from MVBench (Li et al., 2024e), the question and answer are adapted into single choice by human annotator | MC | Simple String Match |

Table 18 – continued from previous page

| Task Name | Source Description | Output Format | Metrics |
|---|---|---|---|
| Chess Puzzles Checkmate | Data collected from Lichess, and the questions and answers are adapted to match strings | Structured | Set Equality |
| Chess Puzzles Equality | Data collected from Lichess, and the questions and answers are adapted to match strings | Structured | Set Equality |
| Bridge Strategies Expert | Data and answer are collected from Bridge Master 2000 | Open | GPT-4o as Judge |
| Chess Puzzles Crushing | Data collected from Lichess, and the questions and answers are adapted to match strings | Exact | Exact String Match |
| Chess Puzzle Single Step | Data collected from Lichess, and the questions and answers are adapted to match strings | Exact | Exact String Match |
| Chess Find Legal Moves | Data collected from game positions of games in the 2024 FIDE Candidates tournament, and the questions and answers are adapted to match strings | Exact | Chess Move List Jaccard Index, Exact String Match |
| Bridge Strategies Advanced | Data and answer are collected from Bridge Master 2000 | Open | GPT-4o as Judge |
| Chess Winner Identification | Data collected from IsoBench (Fu et al., 2024b), and the questions and answers are adapted by human annotator | Exact | Exact String Match |
| Bridge Strategies Worldclass | Data and answer are collected from Bridge Master 2000 | Open | GPT-4o as Judge |
| Mahjong | Data collected from website and screenshot of MajSoul, and the answer are annotated by human annotator | Exact | Exact String Match |
| Chess Sygyzy Endgames | Endgames created by human annotator and data collected from https://syzygy-tables.info, and the questions and answers are adapted to match Jaccard index | Exact | Chess Move List Jaccard Index, Exact String Match |
| Go Capture Stone | Data collected from https://online-go.com/learn-to-play-go/capture and https://forums.online-go.com/t/capture-go-problems/31531/9, and the questions and answers are adapted to match strings | Exact | Exact String Match |
| Bongard Problem | Data collected from https://www.oebp.org/welcome.php and https://www.foundalis.com/res/bps/bpidx.htm | Contextual | String Set Equality Comma |

Table 18 – continued from previous page

| Task Name | Source Description | Output Format | Metrics |
|---|---|---|---|
| Number Puzzle Kakuro 5x5 | Data collected from https://krazydad.com/kakuro/, and the questions and answers are adapted to match strings | Exact | Exact String Match |
| Mensa Iq Test | Data collected from website, and the questions and answers are adapted to match dict equality | Structured | Dict Equality |
| Arc Agi | Data collected from https://arcprize.org/play and the task is adapted from Intelligence (Chollet, 2019), the question and answer are adapted into a grid of digits by human annotator | Exact | Exact String Match |
| Mnist Pattern | Data collected from MNIST (Deng, 2012), and the questions and answers are adapted to match strings | Numerical | Exact String Match |
| Number Puzzle Sudoku | Data collected from puzzles.ca, and the questions and answers are adapted to match strings | Contextual | Simple String Match |
| Move Pos To Pos Hanoi 4 Pole | Shortest paths derived from a diagram found on website and the questions and answers are created to match strings and the longest common move prefix | Structured | Exact String Match, Longest Common List Prefix Ratio |
| Pictionary Cartoon Drawing Guess | Data collected from An early evaluation of gpt-4v (ision) (Wu et al., 2023), the question and answer are adapted to match strings by human annotator | Exact | Exact Str Match Case Insensitive |
| Pictionary Chinese Food Img2en | Data collected from website, and the questions and answers are adapted to match strings | Exact | Exact Str Match Case Insensitive |
| Pictionary Doodle Guess | Data collected from website, and the questions and answers are adapted to match strings | Exact | Exact String Match |
| Pictionary Skribbl Io | Data collected from screenshots collected by human annotator on skribbl.io and the questions and answers are adapted to match strings | Exact | Exact Str Match Case Insensitive |
| Pictionary Genai Output Chinese | Data collected from screenshot of website, and the questions and answers are adapted to match strings | Exact | Exact String Match |
| Annoying Word Search | Data collected from various websites, and the answers are annotated by human annotator | Contextual | Dict Jaccard Agg Jaccard |

Table 18 – continued from previous page

| Task Name | Source Description | Output Format | Metrics |
|---|---|---|---|
| Logical Reasoning 2d Views Of 3d Shapes | Data collected from website, and the questions and answers are adapted to match strings | Structured | Dict Equality |
| Maze 2d 8x8 | Data generated from https://www.mazegenerator.net/, and the questions and answers are adapted to match strings | Exact | Exact Str Match Case Insensitive |
| Crossword Mini 5x5 | Data collected from website, and the questions and answers are adapted to match strings | Structured | Dict Exact Str Match Agg Recall |
| Rebus | Data collected from website, and the questions and answers are adapted to match strings | Contextual | Simple String Match |
| Icon Arithmetic Puzzle | Data collected from An early evaluation of gpt-4v (ision) (Wu et al., 2023), the question and answer are adapted to match strings by human annotator | Structured | Exact String Match, Sequence Equality |
| Iq Test Open Ended | Data and answers are collected from website | Open | GPT-4o as Judge |
| Ball Cup Swap 3 | Screenshots taken from video and edited together using images found online, and the questions and answers are adapted to match strings | MC | Exact String Match |
| Logical Reasoning 2d Folding | Data collected from website, and the questions and answers are adapted to match strings | MC | Exact String Match |
| Perception Test Video Character Order | Data collected from Perception Test (Patraucean et al., 2024) and the task is adapted from MVBench (Li et al., 2024e), the question and answer are adapted into single answer string by human annotator | Contextual | Simple String Match |
| Comic Page Ordering | Data collected from website | Contextual | Sequence Equality |
| Recipe Image Ordering | Data collected from website | MC | Sequence Equality |
| **Coding** | | | |
| Code Translation Easy | Data and test cases are collected from Pintia | Structured | Program Judge |
| Code Translation Python | Data collected from xCodeEval split (Khan et al., 2023), and test cases are annotated by human | Structured | Program Judge |
| Code Translation Hard | Data and test cases are collected from Pintia | Structured | Program Judge |

Table 18 – continued from previous page

| Task Name | Source Description | Output Format | Metrics |
|---|---|---|---|
| Code Translation Advanced | Data and test cases are collected from Pintia | Structured | Program Judge |
| Symbolic Graphics Programs Computer Aided Design | Data and answer are collected from SGP-Bench (Qiu et al., 2024) | Contextual | Multi Ref Phrase |
| Symbolic Graphics Programs Scalable Vector Graphics | Data and answer are collected from SGP-Bench (Qiu et al., 2024) | Contextual | Multi Ref Phrase |
| Webpage Code Understanding | Data are collected from website, and the question and answer are adapted for string match | MC | Exact String Match |
| Code Add Tag | Data collected from xCodeEval (Khan et al., 2023), the question and answer are adapted to match code tag | Contextual | Set Equality |
| Media Recommend Solutions Stackoverflow | Data are collected from Stack Overflow Website, and the question and answer are adapted to match string | MC | Exact String Match |
| Flowchart Code Generation | Data are collected from website, and the question and answer are designed by human annotator | MC | Exact String Match |
| Code Solution Compare | Data collected from SGP-Bench (Qiu et al., 2024), and the question and answer are adapted for string match | Exact | Exact String Match |
| Code Match Problem | Data collected from SGP-Bench (Qiu et al., 2024), and the question and answer are adapted to match code | Exact | Exact String Match |
| Code Visualization Output Understanding | Data are collected from website, and the question and answer are designed by human annotator | MC | String Set Equality Comma |
| Code Output Result | Data are collected from San-Foundry MCQs, and the question and answer are designed by human annotator | Exact | Code Result Exact Str Match |
| Code Execution | Data collected from execution-v2 (Jain et al., 2024a), the question and answer are adapted to match string | Contextual | Simple String Match |
| Code Retrieval | Data collected from SGP-Bench (Qiu et al., 2024), and the question and answer are adapted to match string | Exact | Exact String Match |

Table 18 – continued from previous page

| Task Name | Source Description | Output Format | Metrics |
|---|---|---|---|
| Table2latex Complex | Data collected from SGP-Bench (Qiu et al., 2024), and the question and answer are adapted for LLM Judge | Structured | GPT-4o as Judge |
| Ocr Table To Html | Data are collected from website, and the question and answer are designed by human annotator | Structured | Simple String Match |
| Ocr Table To Markdown | Data are collected from website, and the question and answer are designed by human annotator | Structured | Simple String Match |
| Ocr Math Text Latex | Data are collected from website, and the question and answer are designed by human annotator to match text with LaTeX | Contextual | Text With Latex Expr Equality |
| Ocr Math Equation | Data are collected from website, and the question and answer are designed by human annotator to match LaTeX | Contextual | Latex Expr Equality |
| Latex Complex Formula Convertion | Data are collected from latex-formulas and TexTeller, and the question and answer are designed by human annotator | Structured | Latex Expr Equality |
| Ocr Table To Latex | Data are collected from website, and the question and answer are designed by human annotator | Structured | Simple String Match |
| Ocr Table To Csv | Data are collected from website, and the question and answer are designed by human annotator | Structured | Simple String Match |
| Code Programming Test Easy | Data and test cases are collected from Pintia | Structured | Program Judge |
| Code Programming Test Hard | Data and test cases are collected from Pintia | Structured | Program Judge |
| Code Programming Test Advanced | Data and test cases are collected from Pintia | Structured | Program Judge |
| Code Programming Extremely Hard | Data and test cases are collected from Pintia | Structured | Program Judge |
| Visualization With Code | Data are collected from website, and the question and answer are designed by human annotator | Structured | GPT-4o as Judge |
| Stackoverflow Debug Qa | Data are collected from Stack Overflow Website, and the question and answer are adapted to match string | Structured | Exact Str Match Case Insensitive, Exact String Match |
| Code Error Line Identification | Data collected from Pintia, and the question and answer are adapted to match string | MC | Exact String Match |

Table 18 – continued from previous page

| Task Name | Source Description | Output Format | Metrics |
|---|---|---|---|
| **Perception** | | | |
| Visual Correspondence In Two Images | Images are from BLINK (Fu et al., 2024c). Annotator manually added one more reference point per sample and designed structured answers | Structured | Dict Equality |
| 2D Image Jigsaw Puzzle Easy | Images created by playing the online Jigsaw simulator and taking screenshots | Structured | Dict Exact Str Match Agg Recall |
| Adapted Cvbench Distance | Data collected from CV-Bench's distance split (Tong et al., 2024), and adapted into exact text answer | Exact | Exact String Match |
| Geometry Plot Position Relationship | Data collected from Internet. Question and answers were designed by the annotator | Exact | Exact String Match |
| Video Grounding Spatial | Videos collected from VidOR (Shang et al., 2019). Re-designed questions and answers for this specific task | Contextual | Simple String Match |
| Adapted Cvbench Relation | Data collected from CV-Bench's relation split (Tong et al., 2024), and adapted into exact text answer | Exact | Exact String Match |
| Egocentric Spatial Reasoning | Data are collected from Epic-Kitchen (Damen et al., 2018) and the Internet. Questions and answers are adapted for contextual formatted output | Contextual | Multi Ref Phrase |
| Trance Physics Reasoning Basic | Data are collected from Trance (Hong et al., 2023) by specifically picking up samples with the easiest settings. Questions and answers are re-designed for this specific task | Exact | Exact String Match |
| CLEVER Moving Direction Video | Video data are collected from MVBench (Li et al., 2024e). Questions and answers are adapted for the contextual formatted output format | Contextual | Multi Ref Phrase |
| Trance Physics Reasoning Event | Data are collected from Trance (Hong et al., 2023) by selecting settings where objects are moved. Questions and answers are re-designed for indicating changed objects | MC | Set Equality |
| 3D Fragments Understanding | We write rendering scripts to produce the data from the assets of the Break Bad dataset (Sellán et al., 2022) | Numerical | Simple String Match |

Table 18 – continued from previous page

| Task Name | Source Description | Output Format | Metrics |
|---|---|---|---|
| Physical Property Reasoning | Images are collected from the Internet, questions and answers are designed by annotator | Contextual | Simple String Match |
| ClEVRER Physics | Images are collected from CLEVRER (Yi et al., 2019), questions and answers are re-designed for testing the understanding of physical status | Numerical | Exact String Match |
| ClEVRER Video Moving Object Property Recognition | The videos are collected from MVBench (Li et al., 2024e), the questions and answers are adapted to test the understanding of physical property and dynamics | Contextual | Multi Ref Phrase |
| Trance Physics Reasoning View | Data are collected from Trance (Hong et al., 2023) by selecting the most challenging settings (objects are moved, and two states are captured by different cameras). Questions and answers are re-designed for indicating changed objects | MC | Set Equality |
| Photoshop Operation | Images are collected from the Web, questions and answers designed by annotator | Structured | Jaccard Index |
| Relative Reflectance Of Different Regions | Images come from BLINK (Fu et al., 2024c), the annotator added one more point per image and converted the task into a reflectance sorting task | Structured | Sequence Equality |
| Autonomous Driving Scene Analysis | Images are collected from the Internet, questions and answers are designed by annotator | Exact | Exact Str Match Case Insensitive |
| Functionality Matching In Different Objects | The images come from BLINK (Fu et al., 2024c). The annotator manually added one ref point per image to augment the task | Structured | Dict Equality |
| NLVR2 Two Image Compare QA | Images are collected from NLVR2 (Suhr & Artzi, 2019). Questions and answers re-designed by the annotator | MC | Multi Ref Phrase |
| Egocentric Analysis Single Image | The images are collected from Epic-Kitchens (Damen et al., 2018). Questions and answers are re-designed by the annotator | Exact | Exact String Match Case Insensitive |
| ClEVR Object Existence Video | Videos are collected from MVBench (Li et al., 2024e). Questions and answers are slightly adapted | MC | Simple String Match |

Table 18 – continued from previous page

| Task Name | Source Description | Output Format | Metrics |
|---|---|---|---|
| SNLI-VE Visual Entailment | Data are collected and converted from SNLI-VE dataset (Xie et al., 2019) | Exact | Exact String Match |
| OCR Open-ended QA | Images collected from the Internet. Questions and answers made up by the annotator for the open-ended output format | Open | GPT-4o as Judge |
| Super ClEVR Scene Understanding | Images are collected from SuperCLEVR (Li et al., 2023b). Questions and answers are re-designed by the annotator | Contextual | Multi Ref Phrase |
| Visual Dialog Image Guessing | Images are collected from Visual Dialog dataset (Das et al., 2017). Questions and answers are designed by the annotator | MC | Exact String Match |
| Semantic Matching Of Two Images | Images come from BLINK dataset (Fu et al., 2024c). The annotator augmented the data by adding one more ref point and re-designed the answer | Structured | Dict Equality |
| Recover Masked Word In Figure | The annotator took screenshots from a few public papers on arXiv and designed the question-answer pairs | Contextual | Simple String Match |
| Graph Interpretation | The images of line/dot graphs are collected from the Internet, and the annotator created the question and open-ended reference answer | Open | GPT-4o as Judge |
| Science Figure Explanation | The images of science figures are collected from the Internet, and the annotator created the question and open-ended reference answer | Open | GPT-4o as Judge |
| Bar Chart Interpretation | The images of bar graphs are collected from the Internet, and the annotator created the question and open-ended reference answer | Open | GPT-4o as Judge |
| Electricity Load Estimate Plot | The temporal data were collected from Informer (Zhou et al., 2021) and AutoFormer (Wu et al., 2021). The annotator re-processed the data to design a more specific task | Numerical | Normalized RMSE |
| Average Humidity Estimate Plot | The temporal data were collected from AutoFormer (Wu et al., 2021). The annotator re-processed the data to design a more specific task | Numerical | Normalized RMSE |

Table 18 – continued from previous page

| Task Name | Source Description | Output Format | Metrics |
|---|---|---|---|
| Exchange Rate Estimate Plot | The temporal data were collected from Lai et al. (2018) and AutoFormer (Wu et al., 2021). The annotator re-processed the data to design a more specific task | Numerical | Normalized Rmse |
| Road Map Find Highway Between Two Place | The road map images were collected from Seed-Bencn (Li et al., 2024c) and the Internet. Questions and answers are designed by the annotator | Exact | Exact String Match |
| Transit Map Intersection Points | The transit map images were collected from Seed-Bencn (Li et al., 2024c) and the Internet. Questions and answers are designed by the annotator | Structured | Exact String Match, Sequence Equality Case Insensitive |
| Panel Images Single Question | Panel images were collected from (Fan et al., 2024). Questions and answers were designed by the annotator | MC | Exact String Match |
| Knowledge Graph Understanding | The large knowledge graph image was collected from the Internet. Questions and answers were designed by the annotator | Contextual | Set Equality |
| Panel Images Multi Question | Panel images were collected from (Fan et al., 2024). Questions and answers were designed by the annotator | Structured | Exact String Match |
| Mindmap Elements Parsing | Mindmap images were collected from Seed-Bencn (Li et al., 2024c) and the Internet. Questions and answers are designed by the annotator | Structured | Set Equality Case Insensitive |
| Dvqa | Images were collected from Dvqa dataset (Kafle et al., 2018). Questions and answers were re-designed by the annotator | Numerical | Multi Ref Phrase |
| Figureqa | Images were collected from FigureQA dataset (Kahou et al., 2017). Questions and answers were re-designed by the annotator | MC | Multi Ref Phrase |
| Map Diagram Qa | Images were collected from MapQA dataset (Chang et al., 2022). Questions and answers were re-designed by the annotator | Contextual | Simple String Match |
| Chart Vqa | Data were collected from MathVista (Lu et al., 2023) (statistics subset) and converted into a more specific task | Numerical | General Single Numerical Match |

Table 18 – continued from previous page

| Task Name | Source Description | Output Format | Metrics |
|-----------|--------------------|---------------|---------|
| Photo Sharing Image Retrieval | Images were from the PhotoChat (Zang et al., 2021) dataset. Questions and answers are designed by the annotator | MC | Exact String Match |
| Multi Load Type Prediction From Plot | The temporal data were collected from Informer (Zhou et al., 2021) and AutoFormer (Wu et al., 2021). The annotator re-processed the data to design a more specific task | MC | Sequence Accuracy Case Insensitive |
| Stock Price Future Prediction | The annotator downloaded data from Yahoo! Finance's API, and processed data to design this task | Contextual | Normalized Rmse |
| Traffic Future Prediction From Line Plot | The temporal data were collected from AutoFormer (Wu et al., 2021). The annotator re-processed the data to design a more specific task | Numerical | Normalized Rmse |
| Electricity Plot Future Prediction | The temporal data were collected from AutoFormer (Wu et al., 2021). The annotator re-processed the data to design a more specific task | Numerical | Normalized Rmse |
| Ili Ratio Future Prediction | The temporal data were collected from AutoFormer (Wu et al., 2021). The annotator re-processed the data to design a more specific task | Numerical | Normalized Rmse |
| Paper Vqa | The annotator took high-resolution screenshots of a few papers on arXiv, and designed the questions and answers | Contextual | Simple String Match |
| Doc Vqa | Data and open-ended QA pairs were converted from DocMatix (HuggingFaceM4, 2024) | Open | GPT-4o as Judge |
| FunSD Document Qa | Images were collected from FunSD (Jaume et al., 2019). Questions and answers were designed by annotator | Contextual | Simple String Match |
| OCR Article Journal | The article screenshots were taken from various websites. Questions and answers were created by the annotator | Contextual | Simple String Match |
| IAM Line Ocr And Locate | Images were collected from the IAM handwritten database (Marti & Bunke, 1999). Questions and answers were re-designed by the annotator | Structured | Exact String Match, Normalized Similarity Damerau Levenshtein |

Table 18 – continued from previous page

| Task Name | Source Description | Output Format | Metrics |
|---|---|---|---|
| OCR Resume Experience Plain | The resume screenshots were taken from various websites. Questions and answers were created by the annotator | Contextual | String Set Equality Line Break |
| Newspaper Ocr In Query Box | Images were collected from The Newspaper Navigator Dataset (Lee et al., 2020). Questions and answers were adapted by the annotator into simple string answer format. | Contextual | Simple String Match |
| OCR Resume Skill Plain | The article screenshots were taken from various websites. Questions and answers were created by the annotator | Contextual | String Set Equality Line Break |
| OCR Resume Employer Plain | The article screenshots were taken from various websites. Questions and answers were created by the annotator | Contextual | String Set Equality Line Break |
| Finance Table Understanding | Images were collected from MMMU (Yue et al., 2024a). Questions and answers were adapted by the annotator into direct numerical output format | Numerical | Exact String Match |
| Monthly Weather Days Count | Images were collected from the Microsoft Weather by taking screenshots. Questions and answers were designed by the annotator. | Structured | Exact String Match |
| Table Understanding Complex Question Answering | Tables were collected from WikiTableQuestions (Pasupat & Liang, 2015) and TabFact (Chen et al., 2019). Questions and answers were designed by the annotator | Contextual | Simple String Match |
| Table Understanding Fetaqa | Data were collected and converted from FetaQA (Nan et al., 2022) | Open | GPT-4o as Judge |
| Table Understanding Fact Verification | Tables were collected from WikiTableQuestions (Pasupat & Liang, 2015) and TabFact (Chen et al., 2019). Questions and answers were designed by the annotator | Contextual | Dict Precision |
| Electricity Future Prediction From Table | The temporal data were collected from AutoFormer (Wu et al., 2021). The annotator reprocessed the data to design a more specific task | Numerical | Normalized Rmse |

Table 18 – continued from previous page

| Task Name | Source Description | Output Format | Metrics |
|---|---|---|---|
| Video Detail Description | Video and description data were collected from VideoDetailCaption (Maaz et al., 2023) and converted into a specific task | Open | GPT-4o as Judge |
| Guess Image Generation Prompt | Examples were collected from various online text-to-image generation demos | Open | GPT-4o as Judge |
| Docci Image Description Long | Data were collected from DOCCI (Onoe et al., 2024) | Open | GPT-4o as Judge |
| Tweets Captioning | The annotator collected the data from X by taking screenshots and and the texts | Open | GPT-4o as Judge |
| Image Captioning With Additional Requirements | Images were collected from various sources on the Web. The annotator used Claude 3.5 Sonnet to generate reference answers and manually polished them | Open | GPT-4o as Judge |
| Ad Count Detection | Image were collected from various websites by taking screenshots. Questions and answers created by the annotator | Numerical | Exact String Match |
| Adapted Cvbench Count | Data were collected from CV-Bench's counting split (Tong et al., 2024) and adapted into a specific task by rewriting the question-answer pairs | Numerical | Exact String Match |
| Av Vehicle Multiview Counting | Images were collected from the nuScenes (Caesar et al., 2020) dataset. The annotator designed the questions and implemented a script to generate the answers from the raw annotation | Numerical | Exact String Match |
| Counting Multi Image | Data were collected from Mantis (Jiang et al., 2024a) and adapted into direct numerical answer | Numerical | Exact String Match |
| Av Human Multiview Counting | Images were collected from the nuScenes (Caesar et al., 2020) dataset. The annotator designed the questions and implemented a script to generate the answers from the raw annotation | Numerical | Exact String Match |
| Shape Composition Shapes | Images were made by the annotator using Canva. Questions and answers were created by the annotator | Structured | Positive Int Match |
| Counting Single Image | Data were collected from Mantis (Jiang et al., 2024a) and adapted into direct numerical answer | Numerical | Exact String Match |

Table 18 – continued from previous page

| Task Name | Source Description | Output Format | Metrics |
|---|---|---|---|
| CLEVRER Video Moving Object Count | Video data are collected from MVBench (Li et al., 2024e). Questions and answers are adapted for the direct numerical output | Numerical | Exact String Match |
| Shape Composition Colours | Images were created by the annotator using Canva. Questions and answers were created by the annotator | Structured | Positive Int Match |
| Face Identity Matching | Images were collected from CelebA (Liu et al., 2015). Questions and answers re-designed by the annotator for this specific task | Numerical | Set Equality |
| Rocks Samples Identify | Images, questions, and answers were collected from the Web by the annotator | Contextual | Simple String Match |
| Animal Pose Estimation | Images were collected from AP-10K (Yu et al., 2021). The annotator implemented a script to produce the answer from raw annotations for this task | Numerical | Sequence Coords Similarity |
| License Plate Recognition | Images were collected from the Web. Questions and answers were created by the annotator | Exact | Exact Str Match Case Insensitive |
| Image Style Recognition | Images were collected from the Web. Questions and answers were created by the annotator | Exact | Exact Str Match Case Insensitive |
| Long String Letter Recognition | Data were designed by the annotator and generated automatically with code | Exact | Exact String Match |
| COCO Object Detection By Query Property | Images were from MS-COCO (Lin et al., 2014). Questions and answers were re-designed by the annotator and adapted manually | Numerical | Exact String Match, Nbbox Iou Tuple |
| Widerface Face Count And Event Classification | Images were collected from WiderFace (Yang et al., 2016). Questions and answers were designed and produced by the annotator | Structured | Exact String Match, Simple String Match |
| Handwritten Math Expression Extraction | Data were collected from HME100K (Yuan et al., 2022) | Contextual | Latex Expr Equality |
| Geometry Reasoning Circled Letter | Image were collected from Rahmanzadehgervi et al. (2024) are manually created. Questions and answers were re-designed by the annotator | Structured | Exact String Match, Sequence Equality |

Table 18 – continued from previous page

| Task Name | Source Description | Output Format | Metrics |
|---|---|---|---|
| Av Multicamera Tracking Predict Bbox | Images were collected from the nuScenes (Caesar et al., 2020) dataset. The annotator designed the questions and implemented a script to generate the answers from the raw annotation | Numerical | Nbbox Iou Sequence |
| ASCII Art Understanding | Data and annotations were collected and created by the annotator from various online resources | MC | Exact String Match |
| Face Keypoint Detection | Raw data were from CelebA (Liu et al., 2015). The annotator wrote a script to produce the answers for this task | Structured | Sequence Coords Similarity |
| Extract Webpage Headline | Images were collected from VisualWebBench (Liu et al., 2024a). Questions and answers were adapted by the annotator | Contextual | Simple String Match |
| Waldo | Images and annotations were collected and created by the annotator using various resources on the Web | Structured | Dict Nbbox Iou Tuple Agg Jaccard |
| Geographic Remote Sensing Land Cover | Images and annotations were collected and converted from SATIN (Roberts et al., 2023) | Contextual | Sequence Equality |
| Signboard Identification | Images were collected from the Internet. The annotator created the question-answer pairs | Contextual | Simple String Match |
| Long String Number Recognition | Data were designed by the annotator and generated automatically with code | Exact | Exact String Match |
| Waybill Number Sequence Extraction | Images were collected from the Internet. The annotator created the question-answer pairs | Contextual | Simple String Match |
| Single Person Pose Estimation | hello, this is Source Description | Structured | Sequence Coords Similarity |
| COCO Person Detection | Images were from MS-COCO (Lin et al., 2014). Questions and answers were re-designed by the annotator and adapted with a script | Numerical | Exact String Match, Nbbox Iou Tuple |
| Places365 Scene Type Classification | Images were collected from Places365 (Zhou et al., 2017). Questions and answers were re-designed and generated by the annotator | Exact | Exact String Match |
| Visual Prediction Rater Openable Part Segmentation | Images were collected using screenshots from arXiv papers' qualitative results. Questions and answers were created by the annotator | MC | Sequence Equality |

Table 18 – continued from previous page

| Task Name | Source Description | Output Format | Metrics |
|---|---|---|---|
| Visual Prediction Rater Panoptic Segmentation | Images were collected using screenshots from qualitative results from the arXiv papers. Questions and answers were created by the annotator | MC | Sequence Accuracy Case Insensitive |
| Visual Prediction Rater Semantic Segmentation | Images were collected using screenshots from the qualitative results of the arXiv papers. Questions and answers were created by the annotator | MC | Sequence Accuracy Case Insensitive |
| Video To Camera Trajectory Retrieval | Data were collected from the project page of VD3D (Bahmani et al., 2024). Questions and answers designed and created by the annotator | MC | Exact String Match |
| Sceneqa Scene Transition Video | Video data are collected from MVBench (Li et al., 2024e). Questions and answers are adapted by the annotator into open-ended format | Open | GPT-4o as Judge |
| Video Segments Reordering | Raw data come from UCF101 (Soomro et al., 2012). The annotator designed the task and re-organized the data to produce the question-answer pairs | Structured | Sequence Equality |
| Action Sequence Understanding | Data were collected from MileBench (Song et al., 2024). Questions and answers were designed and created by the annotator | Exact | Exact String Match |
| Video Action Recognition | Raw data come from UCF101 (Soomro et al., 2012). The annotator designed the task and re-organized the data to produce the question-answer pairs | Structured | Exact String Match |
| Google Streetview Line Sorting | The data were taken from Google Maps. Questions and answers were created by the annotator | Structured | Sequence Equality |
| Next Action Prediction | Data were collected from MileBench (Song et al., 2024). Questions and answers were designed and created by the annotator | MC | Exact String Match |
| Perception Test Video Action Count | Video data are collected from MVBench (Li et al., 2024e). Questions and answers are adapted by the annotator into direct numerical output format | Numerical | Exact String Match |

Table 18 – continued from previous page

| Task Name | Source Description | Output Format | Metrics |
|---|---|---|---|
| Google Streetview Line Reasoning | The data were taken from Google Maps. Questions and answers were created by the annotator | MC | Simple String Match |
| Video Camera Motion Description | Videos were collected from VidOR (Shang et al., 2019). Questions and answers re-designed and created by the annotator | Exact | Exact String Match |
| Video Grounding Temporal | Videos were collected from VidOR (Shang et al., 2019). Questions and answers re-designed and created by the annotator | MC | Simple String Match |
| Web Action Prediction | Data were collected from Visual-WebBench (Liu et al., 2024a) | MC | Exact String Match |
| Cam Traj To Video Selection | Data were collected from the project page of VD3D (Bahmani et al., 2024). Questions and answers designed and created by the annotator | Contextual | Simple String Match |
| Sta Action Localization Video | Video data are collected from MVBench (Li et al., 2024e). Questions and answers are repurposed for the contextual formatted output format | Contextual | Simple String Match |
| Contain Contain Images | Images were collected from the Web. Questions and constraints are designed by the annotator. This task has no reference answer | Open | Constrained Generation |
| Contain Repeat Length | Images were collected from the Web. Questions and constraints are designed by the annotator. This task has no reference answer | Open | Constrained Generation |
| Multi Contain Repeat Position Only Length | Images were collected from the Web. Questions and constraints are designed by the annotator. This task has no reference answer | Open | Constrained Generation |
| Contain Length | Images were collected from the Web. Questions and constraints are designed by the annotator. This task has no reference answer | Open | Constrained Generation |
| Contain Position Images | Images were collected from the Web. Questions and constraints are designed by the annotator. This task has no reference answer | Open | Constrained Generation |

Table 18 – continued from previous page

| Task Name | Source Description | Output Format | Metrics |
|---|---|---|---|
| Contain Position Length | Images were collected from the Web. Questions and constraints are designed by the annotator. This task has no reference answer | Open | Constrained Generation |
| Xor Images | Images were collected from the Web. Questions and constraints are designed by the annotator. This task has no reference answer | Open | Constrained Generation |
| Multi Contain Repeat | Images were collected from the Web. Questions and constraints are designed by the annotator. This task has no reference answer | Open | Constrained Generation |
| Contain Contain Length | Images were collected from the Web. Questions and constraints are designed by the annotator. This task has no reference answer | Open | Constrained Generation |
| Multi Contain Position Only | Images were collected from the Web. Questions and constraints are designed by the annotator. This task has no reference answer | Open | Constrained Generation |
| Relative Depth Of Different Points | Images were collected from BLINK (Fu et al., 2024c). The annotator augmented each sample by adding one more reference point manually and adjusted the answers | MC | Exact String Match |
| Visual Prediction Rater Depth Estimation | Images were collected by taking screenshots from depth estimation papers on arXiv. Questions and answers were created by the annotator | MC | Sequence Accuracy Case Insensitive |
| Visual Prediction Rater Novel View Synthesis | Images were collected by taking screenshots from novel view synthesis papers on arXiv. Questions and answers were created by the annotator | MC | Sequence Equality |
| Pokemon 3d Recognition | Images were created by the annotator from the Pokemon Go game. Questions and answers were designed by the annotator | Structured | Exact String Match |
| Av View Identification | Images were collected from the nuScenes (Caesar et al., 2020) dataset. Questions and answers were designed and created by the annotator | Contextual | Sequence Accuracy Case Insensitive |

Table 18 – continued from previous page

| Task Name | Source Description | Output Format | Metrics |
|---|---|---|---|
| Multiview Reasoning Camera Moving | Images were collected from BLINK (Fu et al., 2024c). Questions and answers were re-designed and augmented by the annotator | Exact | Exact String Match |
| 3d Indoor Scene Text Bbox Prediction | The data is adapted from Multi3DRefer (Zhang et al., 2023). Questions and answers were designed by the annotator and dataset annotation. | Numerical | Nbbox Iou Single |
| Google Streetview Circle Reasoning | The data were taken from Google Maps. Questions and answers were created by the annotator | MC | Simple String Match |
| Google Streetview Direction Understanding | The data were taken from Google StreetView. Questions and answers were created by the annotator | Exact | Exact String Match |
| Video Motion Matching Real 3d | Videos were collected from the project page of Shen et al. (2024). Questions and answers were created by the annotator | MC | Exact String Match |
| Video Motion Matching 3d Real | Videos were collected from the project page of Shen et al. (2024). Questions and answers were created by the annotator | MC | Exact String Match |
| Visual Prediction Rater 3d Assembled Quality Understanding | Data were collected from the project page of Wang et al. (2024e). Questions and answers were designed and created by the annotator | MC | Sequence Equality |
| Visual Prediction Rater Surface Normal Estimation | Images were collected by taking screenshots from surface normal estimation papers on arXiv. Questions and answers were created by the annotator | MC | Sequence Accuracy Case Insensitive |
| Adapted Cvbench Depth | Images were collected from CV-Bench (Tong et al., 2024). Answers were adapted by the annotator into exact text | Exact | Exact String Match |
| Visual Prediction Rater Plane Segmentation | Images were collected by taking screenshots from plane segmentation papers on arXiv | MC | Sequence Accuracy Case Insensitive |
| 3d Indoor Scene Text Bbox Selection | Images were collected by taking screenshots from 3D scene understanding papers on arXiv. Questions and answers were designed and generated by the annotator | MC | Exact String Match |

Table 18 – continued from previous page

| Task Name | Source Description | Output Format | Metrics |
|-----------|--------------------|---------------|---------|
| Google Streetview Circle Sorting | The data were taken from Google Maps. Questions and answers were created by the annotator | Structured | Sequence Equality |
| **Metrics** | | | |
| Paper Review Writing | Data collected from OpenReview's public paper reviews | Open | GPT-4o as Judge |
| Paper Review Rating | Data collected from OpenReview's public paper reviews | Numerical | Number Rel Diff Ratio |
| Paper Review Acceptance | Data collected from OpenReview's public paper reviews | Exact | Exact String Match |
| Autorater Artifact | Images were collected from ImagenHub (Ku et al., 2023). Questions and answers adapted by the annotator | MC | Exact String Match |
| Autorater Control | Images were collected from ImagenHub (Ku et al., 2023). Questions and answers adapted by the annotator | Exact | Exact String Match |
| Autorater Artifact Reason | Images were collected from ImagenHub (Ku et al., 2023). The annotator created open-ended reference answer manually | Open | Constrained Generation |
| Autorater Aesthetics | Images were collected from ImagenHub (Ku et al., 2023). Questions and answers adapted by the annotator | Exact | Exact String Match |
| Autorater Unmask | Images were collected from ImagenHub (Ku et al., 2023). Questions and answers adapted by the annotator | Exact | Exact String Match |
| Autorater Subject | Images were collected from ImagenHub (Ku et al., 2023). Questions and answers adapted by the annotator | Exact | Exact String Match |
| Autorater 3d Model Texturing | Resources are collected from the user study of Perla et al. (2024). Questions and answers were designed and created by the annotator | Contextual | Sequence Equality |
| Autorater Semantics | Images were collected from ImagenHub (Ku et al., 2023). Questions and answers adapted by the annotator | Exact | Exact String Match |
| Autorater Motion Guided Editing | Images were collected by taking screenshots from image generation papers on arXiv | MC | Sequence Equality |

Table 18 – continued from previous page

| Task Name | Source Description | Output Format | Metrics |
|---|---|---|---|
| Autorater Mask | Images were collected from ImagenHub (Ku et al., 2023). Questions and answers adapted by the annotator | Exact | Exact String Match |
| Video Eval Visual Pref | Video frames were collected from ImagenHub (He et al., 2024). Questions and answers adapted by the annotator | MC | Exact String Match |
| Generated Video Artifacts | Videos were collected by running various text-to-video diffusion models online. Open-ended reference answers were written by the annotator manually | Open | GPT-4o as Judge |
| Video Eval Factual Pref | Video frames were collected from ImagenHub (He et al., 2024). Questions and answers adapted by the annotator | MC | Exact String Match |
| Video Eval Dynamic Pref | Video frames were collected from ImagenHub (He et al., 2024). Questions and answers adapted by the annotator | MC | Exact String Match |
| Vizwiz Quality Accessment For Blind | Images were collected from Chiu et al. (2020). Questions and answers were adapted and redesigned by the annotator | Contextual | Set Equality |
| Reward Models T2i Reward | Images were collected from RLAIF-V dataset (Yu et al., 2024a). Questions and answers were adapted by the annotator | Exact | Exact String Match |
| Reward Models I2t Reward | Images were collected from RLAIF-V dataset (Yu et al., 2024a). Questions and answers were adapted by the annotator | Exact | Exact String Match |
| **Science** | | | |
| Biology Exams V | Data collected from EXAMS-V (Das et al., 2024) and MMMU-Pro (Yue et al., 2024b), and the questions and answers are adapted to match strings | Contextual | Simple String Match |
| Pmc Vqa Medical Image Qa | Data collected from NLVR2 dataset (Suhr et al., 2018), and the questions and answers are adapted to match strings | Contextual | Simple String Match |
| Medical Content Based Retrieval Radiology | Data collected from ROCO dataset (Pelka et al., 2018), and the questions and answers are adapted to match strings | MC | Exact String Match |

Table 18 – continued from previous page

| Task Name | Source Description | Output Format | Metrics |
|---|---|---|---|
| Medical Abdomen MRI Organ Recognition | Data collected from GMAI-MMBench (Chen et al., 2024b), and the questions and answers are adapted to match sequence accuracy | Contextual | Sequence Accuracy Case Insensitive |
| Medical Multi Organ Segmentation Rater | Data collected from pdf screenshot, and the questions and answers are adapted to match strings | MC | Exact String Match |
| Medical Cell Recognition | Data collected from GMAI-MMBench (Chen et al., 2024b), and the questions and answers are adapted to match strings | Exact | Exact String Match |
| Medical Image Artifacts Indentification | Data collected from GMAI-MMBench (Chen et al., 2024b), and the questions and answers are adapted to match strings | Exact | Exact String Match |
| Medical Blood Vessels Recognition | Data collected from GMAI-MMBench (Chen et al., 2024b), and the questions and answers are adapted to match strings | Structured | Exact String Match |
| Healthcare Info Judgement | Data collected from GMAI-MMBench (Chen et al., 2024b), and the questions and answers are adapted to match strings | MC | Exact String Match |
| Electrocardiogram | Data collected from MMMU (Yue et al., 2024a), and the answers are open-ended | Open | GPT-4o as Judge |
| Medical Polyp Segmentation Single Object Rater | Data collected from pdf screenshot, and the questions and answers are adapted to match sequence equality | Structured | Sequence Equality |
| Medical Abdomen Endscopy Organ Recognition | Data collected from GMAI-MMBench (Chen et al., 2024b), and the questions and answers are adapted to match sequence accuracy | Contextual | Sequence Accuracy Case Insensitive |
| Medical Keywords Based Retrieval Non Radiology | Data collected from ROCO dataset (Pelka et al., 2018), and the questions and answers are adapted to match strings | Exact | Exact String Match |
| Medical Parasite Detection | Data collected from pdf screenshot, and the questions and answers are adapted to match set equality | Structured | Set Equality |
| Medical Retrieval Given Surgeon Activity | Data collected from GMAI-MMBench (Chen et al., 2024b), and the questions and answers are adapted to match strings | MC | Exact String Match |

Table 18 – continued from previous page

| Task Name | Source Description | Output Format | Metrics |
|---|---|---|---|
| Medical Counting Lymphocytes | Data collected from GMAI-MMBench (Chen et al., 2024b), and the questions and answers are adapted to match strings | Numerical | Exact String Match |
| Chemistry Exams V | Data collected from EXAMS-V (Das et al., 2024) and MMMU-Pro (Yue et al., 2024b), and the questions and answers are adapted to match strings | MC | Simple String Match |
| Science Molecule Chemistry | Data collected from IsoBench (Fu et al., 2024b), and the questions and answers are adapted by human annotator | Contextual | Simple String Match |
| Mmmu Pro Exam Screenshot | Data collected from MMMU-Pro (Yue et al., 2024b), and the questions and answers are adapted to match strings | MC | Exact String Match |
| Scibench W Solution Open Ended | Data collected from Scibench (Wang et al., 2023b), and the answers are open-ended | Open | GPT-4o as Judge, General Single Numerical Match |
| arXiv Vqa | Data collected from screenshots by human annotator, and the questions and answers are adapted to match strings | MC | Exact String Match |
| Tqa Textbook Qa | Data collected from Dvqa (Kafle et al., 2018), and the questions and answers are refractered from the original TQA dataset | Contextual | Multi Ref Phrase |
| Question Solution Solving | Data collected from webpage screenshots by human annotator | Contextual | General Single Numerical Match |
| Quizlet Question Solving | Data collected from webpage screenshots by human annotator | Contextual | General Single Numerical Match |
| Scibench Fundamental Wo Solution | Data collected from Scibench (Wang et al., 2023b) | Numerical | General Single Numerical Match |
| Mmmu Physics Chemistry Mcq | Data collected from MMMU (Yue et al., 2024a), and the questions and answers are adapted to match strings | Exact | Exact String Match |
| Circuit Diagram Understanding | Data collected from webpage screenshots by human annotator | Numerical | Exact String Match |
| Science Basic Physics | Data collected from IsoBench (Fu et al., 2024b), and the questions and answers are adapted by human annotator | Contextual | Simple String Match |

Table 18 – continued from previous page

| Task Name | Source Description | Output Format | Metrics |
|---|---|---|---|
| Physics Exams V | Data collected from EXAMS-V (Das et al., 2024) and MMMU-Pro (Yue et al., 2024b), and the questions and answers are adapted to match strings | Contextual | Simple String Match |
| **Knowledge** | | | |
| Background Change | Images and labels come from the MFCBench (Wang et al., 2024c) dataset. Questions and annotations were adapted by a human annotator. | MC | Exact String Match |
| Out Of Context | Images and labels come from the MFCBench (Wang et al., 2024c) dataset. Questions and annotations were adapted by a human annotator. | MC | Exact String Match |
| Text Entity Replace | Images and labels come from the MFCBench (Wang et al., 2024c) dataset. Questions and annotations were adapted by a human annotator. | MC | Exact String Match |
| Text Style | Images and labels come from the MFCBench (Wang et al., 2024c) dataset. Questions and annotations were adapted by a human annotator. | MC | Exact String Match |
| Face Attribute Edit | Images and labels come from the MFCBench (Wang et al., 2024c) dataset. Questions and annotations were adapted by a human annotator. | MC | Exact String Match |
| Face Swap | Images and labels come from the MFCBench (Wang et al., 2024c) dataset. Questions and annotations were adapted by a human annotator. | MC | Exact String Match |
| Interpret Force Perspective Illusion | Images come from various websites. Questions and annotations were created by a human annotator. | Exact | Exact String Match |
| Clip Stable Diffusion Generate | Images and labels come from the MFCBench (Wang et al., 2024c) dataset. Questions and annotations were adapted by a human annotator. | MC | Exact String Match |
| Unusual Images | Images come from various websites. Questions and annotations were created by a human annotator. | Open | GPT-4o as Judge |

Table 18 – continued from previous page

| Task Name | Source Description | Output Format | Metrics |
|---|---|---|---|
| Forensic Detection Of Different Images | Images and labels come from the BLINK benchmark (Fu et al., 2024c). Questions and annotations were adapted by a human annotator. | MC | Exact String Match |
| Veracity | Images and labels come from the MFCBench (Wang et al., 2024c) dataset. Questions and annotations were adapted by a human annotator. | MC | Exact String Match |
| Distinguish AI Generated Image | Images come from various websites and image generators. Questions and annotations were created by a human annotator. | Exact | Exact String Match |
| Cultural Vqa | Images and labels come from the CulturalVQA benchmark (Romero et al., 2024). Questions and annotations were adapted by a human annotator. | Contextual | Multi Ref Phrase |
| Human Relationship Reasoning | Images come from various websites. Questions and annotations were created by a human annotator. | Contextual | Simple String Match |
| Sign Language | Videos come from Dr. Bill Vicars' "Signs" YouTube channel. Questions and annotations were created by a human annotator. | Contextual | Multi Ref Phrase |
| Ishihara Test | Images come from various websites. Questions and annotations were created by a human annotator. | Structured | Set Precision |
| Llavaguard | Images and labels come from the LlavaGuard benchmark (Helff et al., 2024). Questions were created by a human annotator. | Structured | Exact String Match |
| Red Teaming Racial | Images and labels come from the Red Teaming benchmark (Li et al., 2024f). Questions were created by a human annotator or generated by GPT-4. | Open | GPT-4o as Judge |
| Red Teaming Captcha | Images and labels come from the Red Teaming benchmark (Li et al., 2024f). Questions were created by a human annotator or generated by GPT-4. | Open | GPT-4o as Judge |
| Red Teaming Politics | Images and labels come from the Red Teaming benchmark (Li et al., 2024f). Questions were created by a human annotator or generated by GPT-4. | Open | GPT-4o as Judge |

Table 18 – continued from previous page

| Task Name | Source Description | Output Format | Metrics |
|---|---|---|---|
| Mmsoc Hateful-memes | Images and labels come from the MMSoc benchmark (Jin et al., 2024). Questions and answers were adapted by a human annotator. | MC | Exact String Match |
| Red Teaming Visual Order B | Images and labels come from the Red Teaming benchmark (Li et al., 2024f). Questions were created by a human annotator or generated by GPT-4. | Open | GPT-4o as Judge |
| Red Teaming Celebrity | Images and labels come from the Red Teaming benchmark (Li et al., 2024f). Questions were created by a human annotator. or generated by GPT-4 | Open | GPT-4o as Judge |
| Mmsoc Memotion | Images and labels come from the MMSoc benchmark (Jin et al., 2024). Questions and answers were adapted by a human annotator. | Structured | Exact String Match |
| Mmsoc Misinformation Politifact | Images and labels come from the MMSoc benchmark (Jin et al., 2024). Questions and answers were adapted by a human annotator. | MC | Exact String Match |
| Red Teaming Jailbreak | Images and labels come from the Red Teaming benchmark (Li et al., 2024f). Questions were created by a human annotator or generated by GPT-4. | Open | GPT-4o as Judge |
| Red Teaming Visual Order A | Images and labels come from the Red Teaming benchmark (Li et al., 2024f). Questions were created by a human annotator or generated by GPT-4. | Open | GPT-4o as Judge |
| Mmsoc Misinformation Gossipcop | Images and labels come from the MMSoc benchmark (Jin et al., 2024). Questions and answers were adapted by a human annotator. | MC | Exact String Match |
| Red Teaming Visualmisleading | Images and labels come from the Red Teaming benchmark (Li et al., 2024f). Questions were created by a human annotator. | Open | GPT-4o as Judge |
| Video Content Follow Up | Videos taken from YouTube. Questions and answers created by human annnotator. | Open | GPT-4o as Judge |
| Meme Explain | Images come from various websites. Questions were created by a human annotator. | Open | GPT-4o as Judge |

Table 18 – continued from previous page

| Task Name | Source Description | Output Format | Metrics |
|---|---|---|---|
| Funny Image Title | Images come from various websites. Questions were created by a human annotator. | Open | GPT-4o as Judge |
| Emotion Recognition | Videos and labels come from the CAER dataset (Lee et al., 2019). Questions and answers were adapted by a human annotator. | Exact | Exact String Match |
| Image Humor Understanding | Images come from various websites. Questions were created by a human annotator. | Open | GPT-4o as Judge |
| Humor Explanation | Images and labels come from a Humor Understanding benchmark derived from the New Yorker Caption Contest (Hessel et al., 2022). Questions were created by a human annotator. | Open | GPT-4o as Judge |
| Mvsa Sentiment Classification | Images and labels come from the MVSA dataset (Niu et al., 2016). Questions and answers were adapted by a human annotator | MC | Exact String Match |
| Video Intent Recognition | Video and labels come from the MIntRec dataset (Zhang et al., 2022). Questions and answers were adapted by a human annotator. | Contextual | Simple String Match |
| Humor Understand Caption Match | Images and labels come from a Humor Understanding benchmark derived from the New Yorker Caption Contest (Hessel et al., 2022). Questions and answers were adapted by a human annotator. | Exact | Exact String Match |
| Figurative Speech Explanation | Images come from various websites. Questions were created by a human annotator. | Open | GPT-4o as Judge |
| Muma Theory Of Mind Social Goal | Images and labels come from the MuMA-ToM dataset (Shi et al., 2024). Questions and answers were adapted by a human annotator. | Contextual | Simple String Match |
| Muma Theory Of Mind Belief Of Goal | Images and labels come from the MuMA-ToM dataset (Shi et al., 2024). Questions and answers were adapted by a human annotator. | Contextual | Simple String Match |
| Hashtag Recommendation | Images and hashtags come from various social media websites. Questions were created by a human annotator. | Structured | Set Precision |

Table 18 – continued from previous page

| Task Name | Source Description | Output Format | Metrics |
|---|---|---|---|
| Dish Ingredient Match | Images and labels come from the HelloFresh website. Questions were created by a human annotator. | MC | Exact String Match |
| Music Sheet Sentiment | Images are music sheets posted to Noteflight. Questions and answers were created by a human annotator. | Exact | Exact String Match |
| Music Sheet Author | Images are music sheets posted to Noteflight. Questions and answers were created by a human annotator. | Exact | Exact String Match |
| Music Sheet Note Count | Images are music sheets posted to Noteflight. Questions and answers were created by a human annotator. | Numerical | Exact String Match |
| Music Sheet Format Qa | Images are music sheets posted to Noteflight. Questions and answers were created by a human annotator. | Numerical | Exact String Match |
| Orchestra Score Recognition | Images come from various websites. Questions were created by a human annotator. | Structured | Exact String Match, Simple String Match |
| Music Sheet Name | Images are music sheets posted to Noteflight. Questions and answers were created by a human annotator. | Exact | Exact String Match |
| Insect Order Classification | Images and labels come from the BIOSCAN-1M dataset (Gharaee et al., 2024). Questions and answers were adapted by a human annotator. | Contextual | Simple String Match |
| Signage Navigation | Images come from various websites. Questions and answers were created by a human annotator. | Exact | Exact String Match |
| Song Title Identification From Lyrics | Screenshots were taken by the human annotator on the Spotify Web Player. Questions and answers were created by the annotator. | Structured | Exact String Match |
| Knowledge Sign Recognition | Images come from various websites. Questions were created by a human annotator. | MC | String Set Equality Comma |
| Brand Logo Recognition And Elaboration | Images come from the FlickrLogo (Romberg et al., 2011) dataset and various websites. Questions were created by a human annotator. | Structured | Multi Ref Phrase |

Table 18 – continued from previous page

| Task Name | Source Description | Output Format | Metrics |
|---|---|---|---|
| Logo2k Same Type Logo Retrieval | Images come from the Logo2K+ dataset (Wang et al., 2020) dataset and various websites. Questions were created by a human annotator. | Structured | Exact Str Match Case Insensitive, Set Equality |
| Chinese Idiom Recognition | Images come from various websites. Questions and answers were created by a human annotator. | Exact | Exact String Match |
| Multi Lingual Ruozhiba Explanation French | Some images and labels are from the COIG-CQIA dataset (Bai et al., 2024) and some images are from Baidu Tieba and annotated by a human annotator. | Open | GPT-4o as Judge |
| Multi Lingual Ruozhiba Explanation Arabic | Some images and labels are from the COIG-CQIA dataset (Bai et al., 2024) and some images are from Baidu Tieba and annotated by a human annotator. | Open | GPT-4o as Judge |
| Multi Lingual Ruozhiba Explanation Spanish | Some images and labels are from the COIG-CQIA dataset (Bai et al., 2024) and some images are from Baidu Tieba and annotated by a human annotator. | Open | GPT-4o as Judge |
| Multi Lingual Ruozhiba Explanation English | Some images and labels are from the COIG-CQIA dataset (Bai et al., 2024) and some images are from Baidu Tieba and annotated by a human annotator. | Open | GPT-4o as Judge |
| Multi Lingual Ruozhiba Explanation Japanese | Some images and labels are from the COIG-CQIA dataset (Bai et al., 2024) and some images are from Baidu Tieba and annotated by a human annotator. | Open | GPT-4o as Judge |
| Multi Lingual Ruozhiba Explanation Russian | Some images and labels are from the COIG-CQIA dataset (Bai et al., 2024) and some images are from Baidu Tieba and annotated by a human annotator. | Open | GPT-4o as Judge |
| Font Recognition | Images and labels are taken from Identifont. Questions are created by a human annotator. | Exact | Exact String Match |
| Traffic Accident Analysis | Images and labels are taken from Jia Kao Bao Dian. Questions are created by a human annotator. | Open | GPT-4o as Judge |
| Multiple States Identify Asia | Images come from various websites and were edited by the annotator. Questions and answers were created by a human annotator. | Contextual | Sequence Accuracy Case Insensitive |

Table 18 – continued from previous page

| Task Name | Source Description | Output Format | Metrics |
|---|---|---|---|
| Multiple States Identify Americas | Images come from various websites and were edited by the annotator. Questions and answers were created by a human annotator. | Contextual | Sequence Accuracy Case Insensitive |
| Multiple States Identify Europe | Images come from various websites and were edited by the annotator. Questions and answers were created by a human annotator. | Contextual | Sequence Accuracy Case Insensitive |
| Multiple States Identify Africa | Images come from various websites and were edited by the annotator. Questions and answers were created by a human annotator. | Contextual | Sequence Accuracy Case Insensitive |
| Worldle | Images and labels are taken from Worldle Daily, a free Geoguessr alternative. Questions and answers are created by a human annotator. | Structured | Exact String Match |
| Location Vqa | Images and labels come from various websites. Questions were created by a human annotator. | Exact | Exact String Match |
| Vibe Eval Open | Images and labels come from the Vibe-Eval dataset Padlewski et al. (2024). Questions were created by a human annotator. | Contextual | Multi Ref Phrase |
| Vibe Eval Phrase | Images and labels come from the Vibe-Eval dataset Padlewski et al. (2024). Questions were created by a human annotator. | Open | GPT-4o as Judge |
| Ancient Map Understanding | Images and labels come from various websites. Questions were created by a human annotator. | Exact | Exact String Match |
| Rocks Samples Compare | Images and labels come from ChinaNeolithic.com's online rock store. Questions were created by a human annotator. | Contextual | Simple String Match |
| Painting Qa | Images and labels come from the MMMU benchmark Yue et al. (2024a). Questions and answers were adapted by a human annotator. | Exact | Exact String Match |
| Art Explanation | Images come from various websites. Questions were created by a human annotator. | Open | GPT-4o as Judge |

Table 18 – continued from previous page

| Task Name | Source Description | Output Format | Metrics |
|---|---|---|---|
| Memorization Chinese Celebrity | Images and labels come from various websites. Questions were created by a human annotator. | Structured | Multi Ref Phrase |
| Memorization Papers | Images and labels come from various websites. Questions were created by a human annotator. | Structured | Simple String Match |
| Memorization Famous Treaty | Images and labels come from various websites. Questions were created by a human annotator. | Structured | Exact String Match, Multi Ref Phrase |
| Memorization Indian Celebrity | Images and labels come from various websites. Questions were created by a human annotator. | Structured | Exact String Match, Multi Ref Phrase |
| Soccer Offside | Images come from various websites. Questions were created by a human annotator. | MC | Exact String Match |
| Deciphering Oracle Bone | Images and labels come from the "Deciphering Oracle Bone Language with Diffusion Models" paper (Guan et al., 2024). Questions were created by a human annotator. | Exact | Exact String Match |
| Kvqa Knowledge Aware Qa | Images and labels come from the MapQA dataset (Chang et al., 2022). Questions and answers were adapted by a human annotator. | Contextual | Simple String Match |
| Character Recognition In Tv Shows | Screenshots were taken by the human annotator on the Amazon Prime Video webpage. Questions and answers were created by the annotator. | Contextual | Set Equality |
| Actor Recognition In Movie | Screenshots were taken by the human annotator on the Amazon Prime Video webpage. Questions and answers were created by the annotator. | Exact | Exact String Match |
| Landmark Recognition And Qa | Images and labels come from the Landmark v2 dataset (Weyand et al., 2020). Questions and answers were adapted by a human annotator. | Structured | Exact String Match, Multi Ref Phrase, Near Str Match |
| Famous Building Recognition | Images and labels come from various websites. Questions were created by a human annotator. | Structured | Exact Str Match Case Insensitive, Exact String Match |

Table 18 – continued from previous page

| Task Name | Source Description | Output Format | Metrics |
|---|---|---|---|
| Landmark Check Two Images | Images and labels come from the Landmark v2 dataset (Weyand et al., 2020). Questions and answers were adapted by a human annotator. | Structured | Exact Str Match Case Insensitive |
| Defeasible Reasoning | Images and labels come from various websites. Questions were created by a human annotator. | Open | GPT-4o as Judge |
| Poetry Limerick | Images come from various websites. Questions and evaluation constraints were created by a human annotator. | Open | Constrained Generation |
| Poetry Shakespearean Sonnet | Images come from various websites. Questions and evaluation constraints were created by a human annotator. | Open | Constrained Generation |
| Poetry Custom Rhyming Scheme | Images come from various websites. Questions and evaluation constraints were created by a human annotator. | Open | Constrained Generation |
| Poetry Acrostic Alliteration | Images come from various websites. Questions and evaluation constraints were created by a human annotator. | Open | Constrained Generation |
| Poetry Haiku | Images come from various websites. Questions and evaluation constraints were created by a human annotator. | Open | Constrained Generation |
| Poetry Petrarchian Sonnet Optional Meter | Images come from various websites. Questions and evaluation constraints were created by a human annotator. | Open | Constrained Generation |
| Poetry Acrostic | Images come from various websites. Questions and evaluation constraints were created by a human annotator. | Open | Constrained Generation |
| Ascii Art 30 | Images come from various websites. Reference ASCII art images were created using the ASCII Art Archive's "Image to ASCII Art" tool. | Contextual | ASCII Art GPT-4o Judge |
| **Mathematics** | | | |
| Graph Shortest Path Kamada Kawai | Data collected from Visual Graph Arena Dataset by human annotator, and the questions and answers are adapted to match strings | Numerical | Exact String Match |

Table 18 – continued from previous page

| Task Name | Source Description | Output Format | Metrics |
|---|---|---|---|
| Graph Shortest Path Planar | Data collected from Visual Graph Arena Dataset by human annotator, and the questions and answers are adapted to match strings | Numerical | Exact String Match |
| Graph Connectivity | Data collected from IsoBench (Fu et al., 2024b), and the questions and answers are adapted by human annotator | Structured | Exact String Match |
| Graph Theory | Data collected from MathVision (Wang et al., 2024b), and the questions and answers are adapted by human annotator | Exact | Exact String Match |
| Graph Isomorphism | Data collected from IsoBench (Fu et al., 2024b), and the questions and answers are adapted by human annotator | MC | Exact String Match |
| Graph Hamiltonian Cycle | Data collected from Visual Graph Arena Dataset by human annotator, and the questions and answers are adapted to match set precision | Structured | Exact String Match, Set Precision |
| Graph Hamiltonian Path | Data collected from Visual Graph Arena Dataset by human annotator, and the questions and answers are adapted to match set precision | Structured | Exact String Match, Set Precision |
| Graph Chordless Cycle | Data collected from Visual Graph Arena Dataset by human annotator, and the questions and answers are adapted to match strings | Numerical | Exact String Match |
| Topological Sort | Data collected from screenshots by human annotator | Structured | Set Equality |
| Graph Maxflow | Data collected from IsoBench (Fu et al., 2024b), and the questions and answers are adapted by human annotator | Numerical | Exact String Match |
| Scibench Calculus Wo Solution | Data collected from Scibench (Wang et al., 2023b) | Numerical | General Single Numerical Match |
| Clevr Arithmetic | Data collected from Clevr (Johnson et al., 2017) | Numerical | Exact String Match |
| Iconqa Count And Reasoning | Data collected from IConQA (Lu et al., 2021), with annotation refractered from the original IConQA dataset | Numerical | Multi Ref Phrase |
| Number Comparison | Data collected from screenshots by human annotator | Numerical | Exact String Match |

Table 18 – continued from previous page

| Task Name | Source Description | Output Format | Metrics |
|---|---|---|---|
| Math Exams V | Data collected from MMMU-Pro (Yue et al., 2024b), and the questions and answers are adapted to match numerical data | MC | General Single Numerical Match |
| Theoremqa | Data collected from screenshots by human annotator | Contextual | Boxed Single Numerical Match |
| Math | Data collected from screenshots by human annotator | Numerical | Boxed Single Numerical Match |
| Math Parity | Data collected from IsoBench (Fu et al., 2024b), and the questions and answers are adapted by human annotator | MC | Exact String Match |
| Math Breakpoint | Data collected from IsoBench (Fu et al., 2024b), and the questions and answers are adapted by human annotator | Numerical | Exact String Match |
| Math Convexity Value Estimation | Data collected from IsoBench (Fu et al., 2024b), and the questions and answers are adapted by human annotator | Structured | Exact String Match, Number Rel Diff Ratio |
| Geometry Reasoning Count Line Intersections | Data collected from Vision language models are blind (Rahmanzadehgervi et al., 2024), and the questions and answers are adapted by human annotator | Structured | Exact String Match |
| Geometry Length | Data collected from MathVision (Wang et al., 2024b), and the questions and answers are adapted by human annotator | Contextual | General Single Numerical Match |
| Geometry Reasoning Nested Squares | Data collected from Vision language models are blind (Rahmanzadehgervi et al., 2024), and the questions and answers are adapted by human annotator | Structured | Exact String Match |
| Geometry Transformation | Data collected from MathVision (Wang et al., 2024b), and the questions and answers are adapted by human annotator | Contextual | General Single Numerical Match |
| Geometry Reasoning Overlapped Circle | Data collected from Vision language models are blind (Rahmanzadehgervi et al., 2024), and the questions and answers are adapted by human annotator | Structured | Exact String Match |
| Geometry Area | Data collected from MathVision (Wang et al., 2024b), and the questions and answers are adapted by human annotator | Numerical | Exact String Match |

Table 18 – continued from previous page

| Task Name | Source Description | Output Format | Metrics |
|-----------|--------------------|---------------|---------|
| Geometry Reasoning Grid | Data collected from Vision language models are blind (Rahmanzadehgervi et al., 2024), and the questions and answers are adapted by human annotator | Structured | Exact String Match |
| Polygon Interior Angles | Data collected from screenshots by human annotator | Numerical | Angle Seq Float Rmse |
| Geometry Solid | Data collected from MathVision (Wang et al., 2024b), and the questions and answers are adapted by human annotator | Contextual | General Single Numerical Match |
| Geometry Analytic | Data collected from MathVision (Wang et al., 2024b), and the questions and answers are adapted by human annotator | Contextual | General Single Numerical Match |
| Geometry Descriptive | Data collected from MathVision (Wang et al., 2024b), and the questions and answers are adapted by human annotator | Contextual | General Single Numerical Match |
| Counterfactual Arithmetic | Data collected from screenshots by human annotator | Numerical | Exact String Match |
| Algebra | Data collected from MathVision (Wang et al., 2024b), and the questions and answers are adapted by human annotator | Contextual | General Single Numerical Match |

## H  AUTHOR CONTRIBUTION STATEMENT

All authors contributed at least 30 tasks to MEGA-BENCH and participated in task brainstorming, annotation, and validation. They also engaged in discussions on data annotation and provided feedback. The following authors made additional contributions to various aspects of the project:

**Jiacheng Chen** co-designed the project with Wenhu Chen, led the benchmark construction process, and coordinated collaboration among all contributors. Jiacheng Chen created and maintained the annotation GUI tool, GitHub repository, draft task taxonomy tree, and results visualization page to facilitate data annotation and improve data quality. Jiacheng Chen led the development and maintenance of the benchmark evaluation pipeline, including the VLM query pipeline, customized evaluation metrics, HuggingFace Space for the project, etc., and conducted the main experiments and analyses. Jiacheng Chen also led the writing of the paper, coordinating core contributors and incorporating their input into the manuscript.

**Tianhao Liang** co-led the benchmark data organization, implemented most model query and evaluation pipelines under a consistent and unified framework, and conducted the experiments and analyses. Tianhao Liang also maintained the evaluation pipeline and implemented the code execution metric. Tianhao Liang made significant efforts in data quality control, error analysis, and creating figures and tables for the paper.

**Sherman Siu** made contributions to the benchmark construction process, including task reviewing, annotator coordination, data quality control, metric implementation, and code maintenance. Sherman Siu contributed and designed a bunch of complex and novel planning tasks. Sherman Siu also analyzed the number of examples per task to investigate the variance of the benchmark score and contributed to writing the main paper.

**Zhengqing Wang** contributed around 40 tasks, including several complex traditional computer vision tasks. Zhengqing Wang organized the benchmark statistics for the Appendix, participated in error case analysis, and developed the project page.

**Kai Wang** contributed around 40 tasks with diverse data sources and output formats. Kai Wang helped organize the benchmark construction process and actively checked the annotation quality of other annotators.

**Yubo Wang** assisted with the experiments of open-source models.

**Yuansheng Ni** helped organize the Appendix and polished §B to §F.

**Wang Zhu** implemented the evaluation metric for symbolic planning tasks and helped with the paper writing.

**Hexiang Hu** participated in discussions of the project's initial idea and continuously provided thoughts and resources for diverse tasks to facilitate the benchmark construction process. Hexiang Hu wrote a significant portion of the main paper, advised on experimental design, and helped present tables and figures.

**Xiang Yue** discussed the high-level directions and goals of the project with Jiacheng Chen and Wenhu Chen. Xiang Yue provided insightful thoughts for multi-dimensional results analysis and assisted with the experiments of open-source models.

**Wenhu Chen** proposed the initial concepts of the project, continuously advised on project progress while refining its strategic scope and direction, and called up and organized all contributors. Wenhu Chen contributed approximately 50 diverse tasks, generated ideas for new tasks, and distributed them to other annotators. Wenhu Chen wrote the initial draft of the main paper to establish the high-level structure and guided the organization and analysis of the results.

