# OpenReview forum: "MEGA-Bench: Scaling Multimodal Evaluation to over 500 Real-World Tasks"
_ICLR.cc/2025/Conference — ICLR 2025 Poster_

### Official Review · Reviewer_9tqt · 2024-11-01

**Soundness:** 2
**Presentation:** 3
**Contribution:** 3
**Rating:** 6
**Confidence:** 4

**Summary:**

The paper introduces MEGA-BENCH, a comprehensive multimodal evaluation suite that encompasses over 500 real-world tasks, addressing the diverse daily use cases of end users. Its goal is to optimize for high-quality data samples that cover a wide range of multimodal tasks while facilitating cost-effective and accurate model evaluation. The authors have compiled 507 realistic tasks with over 8,000 samples from 16 expert annotators, embracing various output formats and developing over 40 metrics to accommodate these formats. MEGA-BENCH provides a fine-grained capability report across multiple dimensions, enabling in-depth interaction with and visualization of model capabilities. The paper also evaluates various state-of-the-art vision-language models using MEGA-BENCH, revealing significant performance variations among models that were previously thought to be similar.

**Strengths:**

The creation of MEGA-BENCH is an original contribution to the field of multimodal AI evaluation. It scales up the number of tasks to an unprecedented level, offering a comprehensive assessment of model capabilities across a vast array of real-world applications. The approach of embracing diverse output formats and developing over 40 metrics to accommodate these is innovative, moving beyond the limitations of traditional multi-choice question-based benchmarks.

The quality of the work is evident in the meticulous construction of the benchmark. With 507 realistic tasks and over 8,000 samples collected from 16 expert annotators, the dataset is both extensive and rich in diversity. The rigorous annotation process, including the development of an annotation GUI and a taxonomy tree, ensures high-quality data that is well-suited for evaluating multimodal models.

The paper is well-structured and clearly articulated. The figures and tables are effectively used to convey complex information in a digestible manner. The taxonomy tree and the breakdown of tasks across different dimensions are particularly clear, aiding the reader in understanding the scope and organization of MEGA-BENCH.

**Weaknesses:**

The paper presents a snapshot of model performance but does not address how these benchmarks might be used to track performance over training time. A good benchmark should be verified by scaling laws.

**Questions:**

Has the authors' team conducted any analysis on the environmental impact of the computational resources required for the benchmarking process? If so, could they share some insights?

Are there plans to release the annotation tools, pre-processing pipelines, and evaluation metrics as open-source to facilitate community-wide reproducibility and further development?

Could the authors discuss how the tasks in MEGA-BENCH map to real-world applications? Are there any tasks that are particularly relevant to current industry needs or future technological trends?

---

> ### Author Response · Authors · 2024-11-21
> **Response to Reviewer 9tqt**
>
> We thank the reviewer for the constructive questions and try to address the points in Weaknesses and Questions below:
>
> ---
>
> **W1:** The paper presents a snapshot of model performance but does not address how these benchmarks might be used to track performance over training time. A good benchmark should be verified by scaling laws.
>
> **Answer:** Thanks for the suggestion. We agree that tracking the performance of a VLM over training time is one important usage of general-purpose multimodal benchmarks. However, we do not have access to the intermediate checkpoints of the popular open-source models evaluated in Table 2.
>
> To get some reasonable data points for answering this question, we use the model checkpoints from our unpublished ongoing work. The model uses Qwen2-7B as the language model and SigLIP as the image encoder. The training roughly has two stages inspired by LLaVA-OneVision: the first stage is single-image training, and the second stage combines single-image, multi-image, and video. We get four checkpoints: 1) at the middle of the first stage, 2) at the end of the first stage, 3) at the middle of the second stage, and 4) at the end of the third stage. The results are shown in the table below, which reveals a reasonable trend as the training proceeds.
>
> |  | Core (w/o CoT) |
> |:---------------:|:--------------:|
> | Mid of Stage-1 | 13.58 |
> | End of Stage-1 | 17.21 |
> | Mid of Stage-2 | 25.81 |
> | End of Stage-2 | 26.46 |
>
> The model size is another perspective used to verify the benchmark with scaling laws. We can do this by comparing the models from the same family but different sizes. Table 2 shows that larger models from the same model family (Qwen2-VL, InternVL2, LlavaOneVision, etc.) consistently outperform the smaller ones. Detailed breakdown results are provided in Appendix. E.
>
> ----
>
> **Q1.** Has the authors' team conducted any analysis on the environmental impact of the computational resources required for the benchmarking process? If so, could they share some insights?
>
> **Answer:** We didn’t seriously consider the carbon emission of the evaluation process. The low inference cost of MEGA-Bench is in comparison to a suite of existing benchmarks/datasets used by existing evaluation practices to get detailed breakdown results. People usually needed to set up 10 or more different existing benchmarks to obtain detailed multi-dimensional results. A single benchmark can have thousands of test samples (e.g., the VQA-test has more than 100K test samples, and the MathVista-test has more than 1K samples for math tasks), while ours has ~8K samples in total. Therefore, we believe the computational costs of MEGA-Bench are lower than those of the traditional evaluation paradigm when aiming for a detailed breakdown analysis.
>
> ---
>
> **Q2.** Are there plans to release the annotation tools, pre-processing pipelines, and evaluation metrics as open-source to facilitate community-wide reproducibility and further development?
>
> **Answer:** Yes, we will release everything, including the tools used in our annotation process. We also plan to integrate our benchmark into those unified VLM evaluation frameworks (e.g., [lmms-eval](https://github.com/EvolvingLMMs-Lab/lmms-eval), [VLMEvalKit](https://github.com/open-compass/VLMEvalKit)) to make the evaluation pipeline more accessible.
>
> ---
>
> **Q3.** Could the authors discuss how the tasks in MEGA-BENCH map to real-world applications? Are there any tasks that are particularly relevant to current industry needs or future technological trends?
>
> **Answer:** We discuss two ways that tasks in MEGA-BENCH map to real-world applications:
>
> - Some of the tasks are exactly existing real-world multimodal application/use cases, like deciding which UI button to click on a website given the instruction, making calendar suggestions based on screenshots, playing board games, solving puzzles, etc.
> - Some other tasks are sub-tasks of a real-world application. For example, identifying the nearby cars and pedestrians given street-view images is a sub-task of autonomous driving, understanding the temporal dynamics of an event described by a video is a sub-task of an embodied agent, and understanding the human emotion from a video is a sub-task of chatbots.
>
> In the revised PDF, Table 18 provides more detailed information on each task. This table can help better understand the source of the task and how the annotator collected the data and could help understand the corresponding application of each task.

---

> ### Author Response · Authors · 2024-11-26
> **Rebuttal followup**
>
> Dear Reviewer 9tqt,
>
> We hope our response has addressed your concerns and questions. We would appreciate any additional feedback or suggestions for improving our paper. If you feel that our responses have resolved your questions, we would be grateful if you could consider reflecting this in your evaluation.  Please let us know if there are any remaining concerns or questions we can further clarify.
>
> Thank you again for your time and effort!
>
> Authors of Submission 851

---

### Official Review · Reviewer_8FfE · 2024-11-01

**Soundness:** 3
**Presentation:** 3
**Contribution:** 2
**Rating:** 8
**Confidence:** 3

**Summary:**

The paper presents Mega Bench, a comprehensive benchmark for evaluating multimodal models on over 500 tasks. Mega Bench features a wide range of output formats and uses multiple metrics for evaluation. It includes a detailed capability report for popular language and vision models. The benchmark's assessment of leading models reveals significant performance differences and the importance of task diversity.

**Strengths:**

S1: The proposed open-source benchmark includes a large number of diverse tasks for LLMs that can potentially address the limitations of existing benchmarks. It provides valuable resource for the community.

S2: The paper also provides an extensive experiment and analysis of popular LLMs using Mega Bench. It yields many interesting findings.

S3: This paper is well-written and easy to read.

**Weaknesses:**

### Major weaknesses

W1: The rationale behind the task taxonomy tree is not well-explained. Section 3.1 can be strengthened by discussing the design considerations for the draft taxonomy tree. For example, why do we want perception, planning, reasoning? Are these the limitations of existing benchmarks? How do we know this taxonomy is comprehensive and reflects the real usage of LLMs?

W2: The introduction highlights Mega Bench's contributions in multimodal tasks. However, there is limited information regarding non-text tasks in Section 3. I recommend adding a few non-text tasks in Figure 4 and discussing the image and video tasks included in Mega Bench in Section 3.

W3: It is unconvincing that Mega Bench makes significant contributions over existing benchmarks. In the introduction, the paper lists four limitations of existing benchmarks: (1) limited output diversity, (2) lack of task coverage, (3) expensive inference cost, and (4) unmanageable setups. Section 3 and 4 explain how Mega Bench address limitations (1) and (2), but (3) and (4) remain unaddressed in the paper. I recommend discussing what makes Mega Bench less expensive and easier to run compared to other popular benchmarks.

### Minor weaknesses

M1: Replace $ (L83).

M2: The claim that "many examples or tasks are highly similar in the capabilities that they assess" requires evidence to back it up (L83-83).

M3: The tasks in Mega Bench have a lot in common with those in Big Bench. A detailed comparison to Big Bench would be beneficial.

**Questions:**

Q1: There are many different input/output formats and metrics in Mega Bench. How does Mega Bench address the challenge of "Unmanageable Setups" mentioned in the introduction?

Q2: Are there any copyright / privacy concerns for the tasks in the bench mark?

---

> ### Author Response · Authors · 2024-11-21
> **Response to  Reviewer 8FfE (1/2)**
>
> We thank the reviewer for the insightful and detailed comments. We try to address the concerns and questions below:
>
> ---
>
> **W1:** The rationale behind the task taxonomy tree is not well-explained. Section 3.1 can be strengthened by discussing the design considerations for the draft taxonomy tree. For example, why do we want perception, planning, reasoning? Are these the limitations of existing benchmarks? How do we know this taxonomy is comprehensive and reflects the real usage of LLMs?
>
> **Answer:** We created the first two levels of our task taxonomy mainly with two considerations: 1) we thoroughly review previous multi-task or multi-discipline LLM/VLM benchmarks from the literature to understand how they categorize their test samples, and 2) we come up with a task organization that is suitable for organizing annotation efforts.
>
> Concretely, the task tagging with multi-dimensional keywords was inspired by BIG-bench. We also borrowed thoughts from MMBench for categorization based on skills and from MMMU for categorization based on input visual formats. We created the task taxonomy based on the application type mainly to organize the tasks in a way that minimizes the potential overlaps between annotators. The initial application types (first two levels) were summarized from the main applications of many existing benchmarks (e.g., those listed in Table 1) and how recent VLMs are evaluated (e.g., the posts or pages like Qwen2-VL, Pixtral, NVLM, Molmo, etc.), and then refined by hosting a brainstorming session of all annotators, as described in Sec.3.1 of the updated paper.
>
> Through the design process described above, the taxonomy is more comprehensive than the existing benchmarks we discussed in Table 1 and covers the major usages interested by the practitioners (i.e., the groups/companies who work on VLMs). Although we cannot guarantee that all real-world use cases can be covered, MEGA-BENCH can produce detailed breakdown analyses with a single benchmark, which is much less expensive than the combination of 10+ other existing benchmarks.
>
> ---
>
> **W2:** The introduction highlights Mega Bench's contributions in multimodal tasks. However, there is limited information regarding non-text tasks in Section 3. I recommend adding a few non-text tasks in Figure 4 and discussing the image and video tasks included in Mega Bench in Section 3.
>
> **Answer:** We are sorry for the confusion. This seems to be a misunderstanding about Figure 4. The figure aims to illustrate only the answer and the corresponding metrics, and the query inputs (task instruction and images) are omitted to save space. All the inputs of MEGA-BENCH tasks are multimodal, with at least one image. For example, the input of the “Symbolic Planning (Barman)” task contains the task-specific instruction and two images, one image for illustrating the initial state and the other for illustrating the goal stage; the input of the “LaTeX Complex Formula Conversion” task contains a screenshot of a complex equation. We have updated the caption for clarification. There will be an interactive navigation page to visualize all our tasks with concrete examples on our project page.
>
> ---
>
> **W3:** It is unconvincing that Mega Bench makes significant contributions over existing benchmarks. In the introduction, the paper lists four limitations of existing benchmarks: (1) limited output diversity, (2) lack of task coverage, (3) expensive inference cost, and (4) unmanageable setups. Section 3 and 4 explain how Mega Bench address limitations (1) and (2), but (3) and (4) remain unaddressed in the paper. I recommend discussing what makes Mega Bench less expensive and easier to run compared to other popular benchmarks.
>
> **Answer:** The comparison is against the suite of many existing benchmarks (like those used in the blog of [Qwen2-VL](https://github.com/QwenLM/Qwen2-VL?tab=readme-ov-file#performance)) rather than comparing to a single benchmark or dataset. A single existing benchmark can have thousands of test samples (e.g., VQA-test has more than 100K test samples, MathVista-test has more than 1K samples for math tasks, MMBench-test has more than 2K samples for perception-focused vision tasks), while ours have ~8K samples in total. We will defer the discussion of the “unmanageable setups” to the reply of Q1.

---

> ### Author Response · Authors · 2024-11-21
> **Response to Reviewer 8FfE (2/2)**
>
> **W4:** Replace $ (L83).
>
> **Answer:** Thanks for the suggestion. We modified the expression of API cost in the updated PDF.
>
> ---
>
> **W5:** The claim that "many examples or tasks are highly similar in the capabilities that they assess" requires evidence to back it up (L83-83).
>
> **Answer:** We added one example to back it up. More discussions about this point can be found in *W1 of Reviewer d5yn*.
>
> ---
>
> **W6:** The tasks in MEGA-Bench have a lot in common with those in Big Bench. A detailed comparison to Big Bench would be beneficial.
>
> **Answer:** We thank the reviewer for the reminder. BIG-bench indeed inspired us in our conceptualization stage. As discussed in *our reply to W1*, we borrowed BIG-bench's organization style of multi-dimensional keywords. BIG-bench’s diverse NLP tasks also inspired us when we brainstormed to add second-level nodes to the draft taxonomy tree. There are three prominent differences when comparing MEGA-BENCH and BIG-bench:
>
> 1. The tasks in BIG-bench are purely in text and focus on evaluating text-level capabilities, while all tasks in MEGA-BENCH have visual inputs (images or videos) and focus more on visual and multimodal capabilities.
> 2. MEGA-BENCH has much more diverse output formats than BIG-bench
> 3. MEGA-BENCH tasks only have a single round of QA, while some tasks in BIG-bench require multiple QA rounds or even two model instances to interact with each other.
>
> ---
>
> **Q1:** There are many different input/output formats and metrics in Mega Bench. How does Mega Bench address the challenge of "Unmanageable Setups" mentioned in the introduction
>
> **Answer:** MEGA-BENCH’s evaluation code indeed contains some third-party dependencies to support the highly customized metrics. However, users can follow our instructions to download the data and set up the environment with less than 10 shell commands. Compared to configuring and running 10+ benchmarks, as we mentioned in the *reply to W1*, we believe our setup is much more manageable.
>
> ---
>
> **Q2:** Are there any copyright / privacy concerns for the tasks in the benchmark?
>
> **Answer:** While almost all the text instructions and annotations are created or re-written by our annotators, we have been highly cautious about the copyright and privacy concerns associated with the images in our benchmark. The majority of the images originate from clearly licensed sources, such as those under Apache, MIT, or Creative Commons licenses. Additionally, a significant portion of the images were created by our annotators, either through screenshots or by capturing content using various software tools. Since our dataset is strictly for academic use only, we can ensure that no copyright issues arise.

---

> ### Author Response · Authors · 2024-11-25
> **Rebuttal followup**
>
> Dear Reviewer 8FfE,
>
> We would like to learn if our response addresses your concerns and questions, and we invite any additional feedback or thoughts for improving our paper. If you feel that our responses resolve the issues raised, we would be grateful if you could consider reflecting this in the evaluation. We would be happy to address any further concerns or questions. Thank you again for your time and effort!
>
> Authors of Submission 851

---

> > ### Comment · Reviewer_8FfE · 2024-11-26
> > **Thank you so much for the detailed response!**
> >
> > Thank you so much for the detailed response! The response has addressed all my concerns. I have changed my score from 6 to 8. Great work!

---

> > > ### Author Response · Authors · 2024-11-26
> > >
> > > We are delighted that our rebuttal addressed your concerns. Thank you so much for your kind words and for taking the time to review our response!

---

### Official Review · Reviewer_d5yn · 2024-11-03

**Soundness:** 4
**Presentation:** 4
**Contribution:** 3
**Rating:** 8
**Confidence:** 3

**Summary:**

The paper presents MEGA-BENCH, a comprehensive multimodal benchmark scaling up to over 500 real-world tasks, designed to assess the diverse capabilities of vision-language models. It offers a fine-grained analysis across various dimensions, including application, input type, output format, and skills, and provides customized metrics for different output formats. The benchmark reveals significant performance variations among state-of-the-art models, emphasizing the importance of task diversity over increasing examples per task for insightful model evaluation.

**Strengths:**

1. MEGA-BENCH has a large scale and coverage, containing over 500 diverse real-world tasks, which allows for an in-depth assessment of multimodal models across various applications and skills.

2. It offers a sophisticated, fine-grained analysis capability by categorizing tasks along multiple dimensions, providing a nuanced understanding of model performance in specific areas and revealing strengths and weaknesses that aggregate scores might obscure.

3. The benchmark's design emphasizes cost-effectiveness and efficiency, demonstrating that increasing task diversity is more valuable for gaining performance insights than simply adding more examples per task.

**Weaknesses:**

1. While MEGA-BENCH offers a vast array of tasks, its large scale may lead to increased computational costs and complexity in evaluation, potentially limiting its accessibility for further research and extensive exploration.
2. MEGA-BENCH's focus on breadth may result in some tasks being too specific or niche, which could limit the generalizability of the benchmark results to a broader range of multimodal problems and applications.

**Questions:**

1. Could you explain more about the background of your 16 annotators and how you make sure for all task instances, the instruction and solution align with each other?
2. For the open-ended tasks, you mentioned using an LLM-assisted metric. How do you handle the potential for bias in the evaluation process, given that the scoring is dependent on a proprietary LLM? If we use different LLM as judges, will their ratings differ a lot from each other?
3. What are the considerations and challenges you preview when scaling MEGA-BENCH even further? How do you plan to maintain the benchmark's relevance and diversity as new multimodal tasks emerge?

---

> ### Author Response · Authors · 2024-11-21
> **Response to Reviewer d5yn (1/2)**
>
> We thank the reviewer for the constructive and detailed feedback and try to address the concerns and questions below.
>
> ---
>
> **W1.** While MEGA-BENCH offers a vast array of tasks, its large scale may lead to increased computational costs and complexity in evaluation, potentially limiting its accessibility for further research and extensive exploration.
>
>  **Answer:**  One key advantage of MEGA-Bench is that we can derive very detailed multi-dimensional analyses with a single benchmark, while people would have needed to set up 10 or more different other benchmarks to obtain similar evaluation results (like in the pages of [QwenVL-2](https://github.com/QwenLM/Qwen2-VL?tab=readme-ov-file#performance), [Pixtral](https://mistral.ai/news/pixtral-12b/), [NVLM](https://nvlm-project.github.io/), [Molmo](https://molmo.allenai.org/blog), etc.). A single other benchmark can have thousands of test samples (e.g., VQA-test has more than 100K test samples, MathVista-test has more than 1K samples for math tasks, MMBench-test has more than 2K samples for perception-focused vision tasks), while ours have ~8K samples in total. Therefore, the computational costs and evaluation complexity of MEGA-Bench are clearly lower than those of the traditional evaluation paradigm when we aim for a detailed breakdown of the analyses of model performance.
>
>  In the updated PDF, we included a single-image setting with 315 tasks from the full MEGA-BENCH. Since each query only contains a single image in this setting, the evaluation cost/complexity becomes much lower, while we can still produce multi-dimensional analyses using the 315 tasks. This can serve as an option with lower evaluation cost, and we will release the data/code for this setting as well.
>
> ---
>
> **W2.** MEGA-BENCH's focus on breadth may result in some tasks being too specific or niche, which could limit the generalizability of the benchmark results to a broader range of multimodal problems and applications.
>
> **Answer:**  If we look at a single task from MEGA-Bench, it can indeed be specific/niche. For example, some puzzles/chess/planning tasks are directly from real-world use cases; some information extraction tasks have very detailed and specific instructions like picking up a restaurant that satisfies the given constraints. **This is why we need a large number of tasks** guided by a carefully designed taxonomy tree, which helps us ensure reasonable coverage of multimodal problems/applications.
>
> ---
>
> **Q1:**  Explain more about the background of your 16 annotators, and how you make sure for all task instances, the instruction and solution align with each other?
>
>  **Answer:** The main background of the 16 annotators is: 2 electrical engineering, 2 math, 1 finance, 1 statistics, 1 biostatistics, 1 communication engineering, and 8 computer science. All annotators have reasonable knowledge and rich user experience with LLMs (12 out of 16 annotators previously served as annotators/authors of LLM/VLM benchmark papers published on top-tier vision or machine learning conferences). We require the annotators to have a strong computer science background so that they can understand the annotation guidelines and adeptly use the annotation tools (our GUI annotation tool and GitHub repo for submitting/maintaining tasks), which is important for maintaining reliable data quality.
>
> There are three key steps to ensure the correctness of the solution and the alignment between the instruction and solution:
>
> **(1). Task review process.** As introduced in Sec.3.1 and Appendix.B , our annotators submit tasks via creating pull requests (PRs) in our private GitHub repository. Core contributors then carefully review each task, communicate with the annotator to fix any observed glitches, and finally merge accepted tasks into the main branch.
>
> **(2). Evaluation results visualization.** As mentioned in Sec.3.1 L224-228 and Appendix.B (Figure 10), we have a visualization page for annotators to check all existing tasks. We periodically update the evaluation results of several leading VLMs, so that annotators can better understand the task difficulty and catch potential mistakes in their annotation.
>
> **(3). Quality control.** As mentioned in Sec3.1 L234-L241, we leverage commercial VLMs to conduct quality control, during which we ask annotators to augment too-easy tasks and remove tasks with wrong/inconsistent annotations
>
> Note that these three steps do not guarantee perfect annotations due to the large annotation workload and high output complexity,  but they effectively fix most of the annotation glitches.
>
> (to be continued)

---

> ### Author Response · Authors · 2024-11-21
> **Response to Reviewer d5yn (2/2)**
>
> **Q2:** For the open-ended tasks, you mentioned using an LLM-assisted metric. **How do you handle the potential for bias in the evaluation process, given that the scoring is dependent on a proprietary LLM? If we use different LLM as judges, will their ratings differ a lot from each other?**
>
> **Answer:** We thank for the great question. We believe the ideal way of doing the LLM-as-judge evaluation should be using multiple leading proprietary LLMs as the judge, and computing the average score. We did not do this in the main paper and only used GPT-4o (0806) mainly to save the API expense of evaluation — we evaluated ~20 models, evaluating each model with multiple commercial VLMs is too expensive for us.
>
> To better understand the biases from different judge models, we conducted an experiment comparing the Open-ended evaluation scores using three judge models: GPT-4o (0806), Claude 3.5 Sonnet (0620), and Gemini 1.5 Pro (001). To make the API expense affordable, we only evaluated three leading models: GPT-4o (0513), Claude 3.5 Sonnet (1022), and Gemini 1.5 Pro (002). The table below presents the results:
>
> |  | GPT-4o (0806) | Claude 3.5 Sonnet (0620) | Gemini 1.5 Pro (001) | Average |
> | --- | --- | --- | --- | --- |
> | Gemini 1.5 Pro (002) | 58.58 | 64.84 | 61.37 | 61.60 |
> | GPT-4o (0513) | 64.78 (+10.58%) | 68.34 (+ 5.39%) | 64.18 (+4.58%) | 65.77 (+6.77%) |
> | Claude 3.5 Sonnet (1022) | 65.63 (+12.03%) | 71.38 (+ 10.08%) | 66.31 (+8.05%) | 67.77 (+10.02%) |
>
>
> Different VLMs indeed lead to different score distributions: Claude 3.5 Sonnet (0620) is the most generous judge, while GPT-4o (0806) is the most strict one. The key finding is that the overall comparison trends using the three judge models are consistent. By further checking the relative score gap, the judge model shows some tendency/bias to assign higher scores to the model from its own family (e.g., the gap between GPT-4o (0513) and Claude 3.5 Sonnet (1022) is the smallest when using GPT-4o (0806) as the judge model; the gaps between Gemini 1.5 Pro (002) and other two evaluated models are the smallest when using Gemini 1.5 Pro (001) as the judge model). However, the overall consistent comparison trends suggest that the minor bias does not hurt the evaluation validity, and we can safely use one specific judge model to save the evaluation API expense.
>
> ----
>
> **Q3:** What are the considerations and challenges you preview when scaling MEGA-BENCH even further? How do you plan to maintain the benchmark's relevance and diversity as new multimodal tasks emerge?
>
> **Answer:** We answer the two questions separately below
>
> We believe the main challenges of scaling MEGA-BENCH further are the annotation labor and task selection. MEGA-BENCH needs non-trivial and realistic multimodal tasks, together with a customized metric for properly evaluating the results, which requires considerable annotation efforts. Furthermore, given the current coverage of the benchmark, looking for a large amount of tasks that have minimal overlaps with existing tasks can be hard.
>
> As a new multimodal task emerges, we need to do three checks to determine if it can be added to the benchmark:
>
> (1). Examine if the new task can be covered by the level-2 nodes of the current task taxonomy tree and the multi-dimensional keywords. If not, we probably cannot extend the keywords/taxonomy structure for a single task unless there is a bunch of new tasks for the new keyword or taxonomy node
>
> (2). Make sure the task have reasonable difficulty (non-trivial yet solvable), and can be evaluated by either a rule-based metric or a LLM judge model.
>
> (3). Make sure the task does not have high overlaps with existing tasks in the benchmark.

---

> ### Comment · Reviewer_d5yn · 2024-11-26
> **Thank you for your response**
>
> Thank you for your detailed response and statistics! All my concerns have been addressed. I would recommend this paper to be accepted.

---

> > ### Author Response · Authors · 2024-11-26
> >
> > Thank you so much for your feedback and kind recommendation! We are glad our rebuttal addressed your concerns and truly appreciate the time and effort you dedicated to reviewing our paper.

---

### Official Review · Reviewer_cTcJ · 2024-11-04

**Soundness:** 3
**Presentation:** 3
**Contribution:** 3
**Rating:** 6
**Confidence:** 4

**Summary:**

This work presents a new benchmark for multimodal LLMs. The authors attempt to create a novel, diverse, comprehensive benchmark for vision-language reasoning using a several-stage process for designing the benchmark, refining the questions, and developing appropriate metrics. The authors conduct a comprehensive large-scale evaluation of current SOTA multimodal models using the benchmark.

**Strengths:**

# Overall assessment

This work presents an interesting contribution in a much-needed space (benchmarks for multimodal large models). To address the current scattershot approach to multimodal model benchmarking, the authors attempt to create a single, highly diverse, comprehensive benchmark for a variety of image-language tasks (including video). To construct the benchmark the authors develop and refine a task taxonomy, but some details around the taxonomy and its construction are unclear. I have concerns about how the benchmark would be used in practice related to the 40 different evaluation metrics, and the distribution over various attributes (number of images, task type, etc.) but am willing to increase my score based on discussion with authors and other reviewers.

# Major comments

* The quality of the benchmark ultimately relies on the authors' proposed taxonomy, as this forms the basis for all data collection. However, I found the description of the annotation process somewhat disappointing; it effectively amounts to "the Feynman Method" (write down the problem, think hard about it, write down the solution). Critically, the authors provide no discussion or framing around the "conceptualization stage" for how they identified the top levels of the taxonomy (perception, planning, reasoning), nor how the annotators were selected or why they are representative of the whole of relevant multimodal knowledge (the sample of annotators could also bias the coverage in various ways). Please provide a clear discussion of (a) what the levels of the taxonomy are (please give the full list) and (b) how these levels were identified and why they comprise a holistic benchmark and (c) the disciplines of the annotators (since the authors state they are graduate or above from diverse disciplines).

* The diversity of output formats is an interesting contribution. However, the diveristy of evaluation metrics (over 40 metrics?!) also makes this benchmark somewhat unwieldy, and raises concerns about usability. These issues arise even in the authors' main findings, stated at the end of Section 1. For example, it is very difficult to understand what it means that GPT-4o is 3.5% better than Claude 3.5? What makes this a "significant margin"? If Qwen2-VL is 10% better than other open source models, what does this mean? T

* It is not clear whether all tasks in the benchmark have a single, objective answer. This makes it difficult to assess models' capabilities (for example, failure to write a latex equation may simply be due to a difference in formatting; writing a story containing two animals hinges on many different criteria which are difficult to assess).

* The advantages of a single, diverse, high-coverage benchmark are outlined nicely in the introduction. However, the paper's contribution hinges on whether it does indeed  achieve strong coverage of a "diverse" suite of tasks. Ultimately, this is nearly impossible to assess, but I have some concerns about the "concepttualization" process above that make me unsure that this benchmark is as comprehensive as the authors claim. On the other hand, the existing benchmarks are also imperfect (and a direct comparison to existing benchmarks in terms of content and task design would make it easier to assess whther the benefits of the new benchmark outweigh the potential downsides and complexity).

* It is unclear why certain task distributions are set as the authors designed them in the benchmark. For example, why are only 4% of tasks 6-8 images, while 8% are 9+ images? Why are 16% of tasks open-ended while 22% are structured? These design decisions can have significant effects when averaging over benchmarks, as will likely occur with this benchmark.

* The empirical study is useful, appears comprehensive, and leads to some interesting conclusions.

* It seems unlikely that the benchmark will last very long by relying on GPT-4o as judge. Is it possible to substitute the LLM judge in the benchmark if a future best frontier model emerges?


# Minor comments

* Another relevant multimodal baseline the authors may want to reference: Bitton, Yonatan, et al. "Visit-bench: A dynamic benchmark for evaluating instruction-following vision-and-language models." Advances in Neural Information Processing Systems 36 (2023): 26898-26922.

# Typos etc

* "these models have shown great potential to solve any desired task with a well-designed prompt" - this is editorlaizing somewhat; please revise.

* L111-113: "comprehensive studies...have discovered" passive voice, consider revising

**Weaknesses:**

see above

**Questions:**

see above

---

> ### Author Response · Authors · 2024-11-21
> **Response to Reviewer cTcJ (1/3)**
>
> We thank the reviewer for the insightful comments and questions. We try to address the concerns and questions one by one below.
>
> ----
>
> **Q1.** Please provide a clear discussion of (a) what the levels of the taxonomy are (please give the full list)  and (b) how these levels were identified and why they comprise a holistic benchmark and (c) the disciplines of the annotators (since the authors state they are graduate or above from diverse disciplines).
>
> **Answer:**  We thank for the detailed comment and resolve the three sub-questions one by one
>
> (a). The original submission actually presented the taxonomy tree up to level 3 in Table 3 of the Appendix, but we acknowledge that the organization of the old Appendix is a bit unclear. In the updated PDF, we organized full details about the taxonomy tree and the statistics of each dimension (skill, application, etc.) in a stand-alone section (Appendix. C). Hope this can make it easier to understand the task structure of MEGA-Bench. We will also create a task visualization tool in our code release to enable easier navigation of the tasks.
>
> (b). We identified the first two levels of our task taxonomy by 1) thoroughly reviewing previous multi-task or multi-discipline LLM/VLM benchmarks from the literature to understand how they categorize their test samples and 2) coming up with a task organization that is suitable for organizing annotation efforts.
>
> Concretely, the task tagging with multi-dimensional keywords was inspired by BIG-bench. We also borrowed thoughts from MMBench for categorization based on skills and from MMMU for categorization based on input visual formats. To organize the tasks to minimize the potential overlaps between annotators, we created the task taxonomy based on the main application type. The initial application types (first two levels) were summarized from many existing benchmarks (listed in Table 1) and then refined/updated by hosting a brainstorming session of all annotators, as mentioned in Sec.3.1 of the updated paper.
>
> (c). The discipline distribution of the 16 annotators is: 2 electrical engineering, 2 math, 1 finance, 1 statistics, 1 biostatistics, 1 communication engineering, and 8 computer science. All annotators have reasonable knowledge and rich user experience with LLMs (12 out of 16 annotators previously served as annotators/authors of LLM/VLM benchmark papers published on top-tier vision or machine learning conferences). We require the annotators to have a strong computer science background so that they can understand the annotation guidelines and adeptly use the annotation tools (our GUI annotation tool, and GitHub repo for submitting/maintaining tasks).
>
> The annotator’s disciplines do not cover all areas (e.g., Humanities or Social Sciences), and we did not claim that “they are representative of the whole of relevant multimodal knowledge.” Instead, we asked the annotators to read relevant papers carefully and look for online resources/documents when designing or collecting tasks in their unfamiliar disciplines.
>
> ----
>
> **Q2.** Concerns about usability.  “For example, it is very difficult to understand what it means that GPT-4o is 3.5% better than Claude 3.5? What makes this a "significant margin"? If Qwen2-VL is 10% better than other open source models, what does this mean?”
>
> **Answer:** The overall score serves as a summarizing indicator for the general capability of different models. These scores provide a general trend among all the evaluated VLMs, while more in-depth analyses are enabled by our multi-dimensional breakdown analysis, as shown in Sec.4.2 (as mentioned in the penultimate point of your major comments) and in the updated Sec4.3 with error analysis. For clarity, we provide a concrete example of the breakdown analysis below:
>
> In the updated paper, we provide results of the new Claude 3.5 Sonnet (1022). Its overall performance slightly surpasses GPT-4o (0513) with a margin of ~0.1%. The comparison of the overall scores indeed provides limited information. However, the detailed breakdown (Figure 5 in the main paper and Table 8-17 in the Appendix) shows more useful insights. For example, (1) the new version of Claude 3.5 Sonnet (1022) outperforms the old version (0620) significantly in planning applications and tasks with UI/Infographics inputs; (2) Claude 3.5 Sonnet works better in keywords like math, planning, and structured outputs, while GPT-4o works better in information extraction and knowledge-intensive tasks. Using a traditional evaluation paradigm (such as in the model page or blog posts of [Qwen2-VL](https://github.com/QwenLM/Qwen2-VL?tab=readme-ov-file#performance), [Pixtral-12B]( [https://mistral.ai/news/pixtral-large/](https://mistral.ai/news/pixtral-12b/)), [NVLM](https://nvlm-project.github.io/), [Aria](https://huggingface.co/blog/RhymesAI/aria), etc.), people usually need to evaluate a suite of at least 10 existing benchmarks to get similar breakdown analyses.

---

> ### Author Response · Authors · 2024-11-21
> **Response to Reviewer cTcJ (2/3)**
>
> **Q3.** “Whether all tasks in the benchmark have a single, objective answer”.
>
> **Answer:** This is a great question. We put a lot of effort into ensuring the metrics (i.e., score functions) could reasonably evaluate the answers. We provide some concrete examples below:
>
> 1. For the output formats evaluated by string matching (`exact match`, `contextual formatted text`, and `multiple-choice`), all the possible options are either directly provided in the task instruction (like the index of images to be selected, the letter choice, or a word/phrase provided in the query context).
>
> 2. For `structured` outputs (latex, python code, etc.), the metrics are highly customized. Some concrete examples: 1) For latex, we implement a latex comparison function to normalize the output latex and compare it with the ground-truth answer using the latex parsing utils in the `sympy` library; 2) For python code, we implement a python code execution metric that runs the model-generated code with the test cases provided in our annotation, and checks whether the output aligns with the expected output (similar to how online judges like LeetCode verify if a solution passes or not); 3)  For the Planning Domain Definition Language in some symbolic planning tasks, we run a simulator to check if the model’s output leads to the desired final state.
>
> 3. For the `open-ended` format, there are two cases: 1) Constrained generation tasks are evaluated by checking if all the constrained are satisfied (e.g., if a generated poetry follows the specified rhythm and contains the subject depicted by the image or if a generated story strictly follows the requirements on length and number of subject occurrences) 2) completely open-ended tasks (from the open-ended subset) are evaluated using LLM as a judge because it is hard to implement rule-based metrics that consider all aspects.
>
> 4. For the `numerical` format, we either use task-specific metrics (e.g., mean IoU for object detection, MSE for temperature prediction, etc.) or use a general numerical matching metric (borrowed from the metric of [MAmmoTH](https://arxiv.org/abs/2309.05653)) that allows a small relative error when comparing the model's answer with the ground truth.
>
> To verify the correctness of our rule-based metrics, we conducted a sanity check for our metric implementations (as discussed in Sec.3.2 ) to make sure that all ground-truth reference answers can get full marks. We will incorporate these discussions into the Appendix.D.
>
> ----
>
> **Q4.  “……** I have some concerns about the "concepttualization" process above that make me unsure that this benchmark is as comprehensive as the authors claim ……  ****a direct comparison to existing benchmarks in terms of content and task design would make it easier to assess whether the benefits of the new benchmark outweigh the potential downsides and complexity”
>
> **Answer:** We actually do not aim to thoroughly cover all possible multimodal use cases in our benchmark, and the comprehensiveness is relative to existing benchmarks, as shown in Table 1 of the paper. To clarify, the “complexity” of MEGA-Bench should be compared to the entire evaluation suite of existing benchmarks because VLM practitioners usually evaluate their models on more than 10 existing benchmarks to obtain breakdown results for various applications, skills, or input formats (like we discussed in the *reply to Q2*).
>
> More concretely, a single existing benchmark can have thousands of test samples and complexity in the evaluation setup. For example, the VQA-test has more than 100K test samples and multiple test splits for perception-focused QA with photograph input format; the MathVista-test has more than 1K samples for math tasks with multiple-choice or integer output; the MMBench-test has more 2K samples for perception and reasoning-focused vision tasks with only multiple-choice output; the MMMU-test has over 10K samples for multi-discipline college-level problems with only multiple-choice output. On the contrary, ours has ~8K samples in total with high diversity in task applications, evaluated skills, input formats, output formats, and so on. Therefore, when considering the complexity of getting detailed multi-dimensional results, the overall evaluation complexity of MEGA-Bench is much lower than the combination of 10+ existing tasks.

---

> ### Author Response · Authors · 2024-11-21
> **Response to Reviewer cTcJ (3/3)**
>
> **Q5.** “It is unclear why certain task distributions are set as the authors designed them in the benchmark. For example, why are only 4% of tasks 6-8 images, while 8% are 9+ images? Why are 16% of tasks open-ended while 22% are structured? These design decisions can have significant effects when averaging over benchmarks, as will likely occur with this benchmark.”
>
> **Answer:** We explicitly control the distributions of application types, output formats, and the number of input images during the benchmark construction process. By ensuring diversity in application types, we naturally achieve diversity in skills and input formats. Below, we elaborate on how these distributions were determined for each of the three dimensions:
>
> - **Application types.** During the benchmark conceptualization stage, we created the first and second-level nodes of the draft taxonomy tree as described in the *reply to Q1*. The number of tasks under each high-level node was determined based on our empirical observations of how people use VLMs in daily scenarios. Greater emphasis was placed on perception, information extraction, knowledge, and planning, as we believe these are the most common multimodal applications. Mathematics and coding received relatively lower priority since most real-world scenarios in these areas involve pure text. For science, its use cases are limited to some specific user groups, such as students and scientific professionals using VLMs to assist with assignments or explore scientific knowledge. For metrics, this type is relatively niche, and we included it as there is an increasing trend in the LLM/VLM community to use large models for automatic evaluation or reward estimation — since we knew this type was a bit biased by the annotator background, we only assigned small budgets to it.
>
> - **Output formats.** We require annotators to design or adapt tasks for diverse outputs. The task reviewers actively monitored the distribution in the annotation process and communicated with the annotators to adjust the output format, so that each format has a reasonable number of tasks.
>
> - **Number of input images.** We had the following considerations when setting up the distribution of single-image, multi-image, and video tasks: 1) single-image tasks should be dominating because most single-round QA in real-world applications have a single image in the query, and some open-source models even only support single-image input; 2) we do not include many video tasks, because most existing models cannot process long video effectively, and video tasks cover a relatively small portion of multimodal applications and skills compared to images. Therefore, we set the distribution to be roughly 60%, 30%, and 10% for single-image, multi-image, and video. The concrete percentage of “6-8 images” and “9+ images” are the factual stats derived from the final tasks — we use the fine-grained groups instead of the general “multi-image” group in case people are interested in the detailed information of multi-image tasks.
>
> ---
>
> **Q6.** “It seems unlikely that the benchmark will last very long by relying on GPT-4o as judge. Is it possible to substitute the LLM judge in the benchmark if a future best frontier model emerges?”
>
> **Answer:** Yes, this is a very good point. In our prompt design (see Figure 13 in the Appendix for the prompt template structure of the LLM-assisted metric), the evaluation criterion for each Open-ended task is highly customized and disentangled with the type of judge LLM. To use other judge models, we only need to inherit the current GPT-4o judge and overwrite several functions following the APIs of the new model.
>
> Reviewer d5yn asked about the potential bias of different LLM judge models, and we followed the above implementation guidelines to extend the metric for Claude 3.5 Sonnet (0620) and Gemini 1.5 Pro (001) as the judge model. Please refer to *Q2 of reviewer d5yn* for the results. We will release all the evaluation codes together with the benchmark data.
>
> ---
>
> **Q7.** “Another relevant multimodal baseline the authors may want to reference: Bitton, Yonatan, et al. "Visit-bench: A dynamic benchmark for evaluating instruction-following vision-and-language models." Advances in Neural Information Processing Systems 36 (2023): 26898-26922.”
>
> **Answer:** Thank you for pointing out this paper. It is a relevant benchmark that we missed in the literature review. We have added it to Table 1 to compare MEGA-BENCH statistics with existing VLM benchmarks.
>
> ----
>
> **Q8.** Typos
>
> **Answer:** Thanks for the careful catch. We fixed these typos and grammatical issues in the updated PDF.

---

> > ### Comment · Reviewer_cTcJ · 2024-11-25
> >
> > Acknowledging that I received and have reviewed the author response. I will retain my original score.

---

> > > ### Author Response · Authors · 2024-11-25
> > >
> > > Thank you again for your time and detailed initial review! We appreciate your efforts. If you have any follow-up questions or thoughts during the remaining discussion period, please feel free to share them.

---

> ### Author Response · Authors · 2024-11-25
> **Rebuttal Followup**
>
> Dear Reviewer,
>
> As the discussion period deadline approaches, we kindly invite any additional feedback or thoughts on our rebuttal. Your insights are highly valued, and we would be happy to address any further concerns or questions. Thank you again for your time and effort!
>
> Authors of Submission 851

---

### Author Response · Authors · 2024-11-21
**General Response and Summary of Revision**

We thank the reviewers for the time and expertise devoted to reviewing our paper. And we are grateful for the constructive and overall positive feedback. To help better address the questions and concerns raised by the reviewers and further improve the paper's quality, we made the following updates to the paper PDF. These contents are used in our response to the reviewers (as noted in the parenthesis).

- **Results and discussions of recently released models (Reviewer cTcJ and Reviewer d5yn).** We evaluated several VLMs released after the initial submission deadline and added the results to the main results table (Table 2), including Claude Sonnet 3.5 (1022), NVLM, and Aria. Based on the new results, we updated the discussion and analysis in Sec.4.2. Some interesting observations could be derived from the new results: the new Claude Sonnet 3.5 (1022) slightly surpasses GPT-4o (0513) and clearly improves over the previous Claude Sonnet 3.5 (0620) on planning tasks and tasks with UI/Infographics inputs, which are consistent with the use case of computer agent as advocated in Anthropic’s blog post (https://www.anthropic.com/news/developing-computer-use).
- **Single-image setting (Reviewer d5yn) .** We added a single-image setting so that models with only single-image support (some open-source models only support one image per query) can also be properly evaluated while also providing an evaluation option with lower cost. The single-image setting contains a subset of 315 tasks from the full MEGA-Bench, with 273 Core tasks and 42 Open-ended tasks. The detailed evaluation setting and the results are organized in **Appendix. A** and **Table 3** of the updated PDF. We evaluated single-image models such as Molmo-72B and Molmo-7B.
- **Error analysis (Reviewer cTcJ).** To help better understand the performance of the currently leading VLMs, we conducted a detailed error analysis by manually checking the evaluation results of GPT-4o (0513) on a subset of 255 Core tasks randomly sampled from MEGA-Bench. We added a new figure with accompanying discussions in Sec.4.3 of the updated PDF. To make more space for the error analysis content, we compressed the analysis of the per-task number of examples.
- **Details about conceptualization and annotator background (Reviewer cTcJ and Reviewer d5yn).** As requested by reviewer cTcJ and reviewer d5yn, we updated Sec.3.1 of the paper to provide more information about the conceptualization stage (creating the draft taxonomy tree) and the background of annotators.
- **Thorough details of the benchmark (Reviewer d5yn and Reviewer 9tqt).** We added more detailed information about the annotation protocols in **Appendix.B**, and added detailed per-task information in **Appendix.G**
- **Task refinement.** Two tasks with a potentially sensitive topic or an unclear task definition are removed (”Same profession gender classification” and “Oil temperature prediction from load plots”), and we update all results accordingly.


We will then post in each reviewer’s thread to address the concrete concerns and questions one by one. We will refer to the updated PDF for more detailed information when necessary.

---

### Meta-Review · Area_Chair_tWRn · 2024-12-22

**Metareview:**

The paper introduces MEGA-BENCH, a comprehensive multimodal benchmark designed to evaluate the diverse capabilities of vision-language models across over 500 tasks. The major advantages of the benchmark are as follows: (i) scope and scale: it includes 507 realistic tasks with over 8,000 samples; (ii) metrics: the benchmark supports diverse output formats and include over 40 evaluation criteria; (iii) structured and interpretable design: it adopts a taxonomy tree to categorize tasks and ensures diversity across applications, input types, and output formats. While the reviewers initially have some concerns regarding the overlap with existing benchmarks, the complexity of evaluation and the practicality of the design, the authors carefully addressed most of them during the rebuttal discussion phase. The paper, at the end, received unanimous positive reviews. The ACs agreed with the reviewers. The ACs urge the authors to incorporate the feedbacks from the reviewers into their final camera ready version. The ACs recommend acceptance.

**Additional Comments On Reviewer Discussion:**

The authors addressed most concerns during the rebuttal phase. Good job!

---

### Decision · Program_Chairs · 2025-01-22

Accept (Poster)